# Over-exploitation of natural resources is followed by inevitable declines in economic growth and discount rate

Adam Lampert [1,2]

A major challenge in environmental policymaking is determining whether and how fast our society should adopt sustainable management methods. These decisions may have long-lasting effects on the environment, and therefore, they depend critically on the discount factor, which determines the relative values given to future environmental goods compared to present ones. The discount factor has been a major focus of debate in recent decades, and nevertheless, the potential effect of the environment and its management on the discount factor has been largely ignored. Here we show that to maximize social welfare, policymakers need to consider discount factors that depend on changes in natural resource harvest at the global scale. Particularly, the more our society over-harvests today, the more policy-makers should discount the near future, but the less they should discount the far future. This results in a novel discount formula that implies significantly higher values for future environmental goods.

[1] School of Human Evolution and Social Change, Arizona State University, Tempe, AZ 85287, USA. [2] Simon A. Levin Mathematical, Computational and Modeling Science Center, Arizona State University, Tempe, AZ 85287, USA. Correspondence and requests for materials should be addressed to A.L. (email: adam.lampert@asu.edu)

The exploitation of ecosystems by humans has long-lasting consequences for the future provision of natural resources and ecosystem services[1,2]. This may negatively affect the provision of food, increase health hazards and risks of natural disasters, and more. Degraded ecosystems may be slow to recover or may not recover naturally even after their exploitation stops[3–5]. Consequently, the availability of natural resources such as food, clean air, and other ecosystem services, may be adversely impacted for extended periods if the ecosystems providing these resources become degraded. For example, the emission of greenhouse gases may affect the global climate for centuries[6,7]; invasive species and diseases may irreversibly damage ecosystems[8,9]; and the non-sustainable harvest of fisheries and forests may leave these systems degraded for decades[2,4], or even lead to their irreversible and permanent degradation[3,10]. Since natural resources are limited, it has been widely recognized that a transition to sustainable harvest is necessary[11]. What the optimal pathway and speed are for this transition, however, constitute the focus of an ongoing debate. For example, it has been suggested that an abrupt transition may slow economic growth in developing countries and may negatively affect production[12], and that rapid emission cuts may create energy deficits before we manage to develop viable substitutes[13].

Determining the optimal strategy for the adoption of sustainable management over time requires cost-benefit analyses. A common approach is to consider a social planner whose objective is to maximize social welfare[14–16]. This is often formalized as maximizing a net present value,

$$\mathrm{NPV} = \int_0^\infty B(t)e^{-\Delta(t)}\mathrm{d}t, \qquad (1)$$

where $B(t)$ is the benefit minus the cost (in units of consumption) due to both the management and the environment at time $t$, and $\Delta(t)$ is the cumulative discount. In turn, the discount factor, $\exp(-\Delta(t))$, is the number of units of some good or currency needed at present to compensate for the lack of one unit at time $t$. The rationale behind discounting is that the objective of our society is to maximize welfare rather than net consumption. In turn, if society is going to be wealthier in the future, then one unit of consumed goods in the future may add less to welfare than the same unit today[14–16].

Accurate discounting is particularly important for environmental policies in which the resultant damages are long-term, such as policies concerning climate change and provision of natural resources[6,17,18]. Specifically, a small difference in the discount may lead to a large difference in estimates of long-term environmental cost. For example, consider no changes in prices and a constant annual discount rate, $\delta \equiv \mathrm{d}\Delta/\mathrm{d}t$. Then, if the cost due to losing some good today is \$1M, then the cost due to losing the exact same good (no depreciation) 100 years from now is ~\$50K if $\delta = 3\%$, and only ~\$2.5K if $\delta = 6\%$. Therefore, even the best estimates of environmental damages may lead to an inadequate policy if we are unable to accurately convert future costs to their present-equivalent dollar value.

The central role that discounting plays in the valuation of natural resources has led to extensive debates over the value that policymakers should use for the discount rate and over how this value varies over time. Specifically, the small values given to future environmental goods due to discounting may contradict our intuition that our society should sustain our planet's ecosystems for future generations. One major debate followed the publication of the Stern report[6], which used a discount rate that is smaller than those used in previous major assessments, and consequently, argued for radical emission cuts. The bulk of the criticism[19] has focused on which discount rate should policymakers use (not on the comprehensive cost assessments). Also,

several authors[16,20–23] proposed that policymakers should use a discount rate that declines over time, and they showed that this is justified if future economic growth is uncertain. Another mechanism that could affect the discount rate is a large perturbation that significantly affects social welfare[24,25], such as an environmental degradation that may occur due to climate change or over-harvesting[26–29]. Particularly, several authors showed that global changes in the provision of non-substitutable natural resources might affect their relative prices[30,31] and the discount rate[25,32]. Nevertheless, these authors considered the changes in the provision of natural resources as given, while the long-term consequences of harvesting on economic growth and discount rate remain largely unknown.

In this paper, we examine how the discount rate and factor are affected by large changes in the harvest methods used at the global scale, such as the transition from over-harvesting to harvesting sustainably. Specifically, the decline in the provision of natural resources due to the future transition might be so large that it will significantly affect social welfare and economic growth. In turn, since discount rates depend on welfare and growth, this means that the discount rate itself could be affected. Revealing harvest-induced changes in the discount will provide policymakers with better evaluations of long-term benefits and costs, thereby enabling them to improve long-term environmental policies. We focus on the harvest of renewable resources in a broad sense, where non-sustainable harvest suppresses the future provision of the resource or the ecosystem service. Examples include the over-harvesting of fish and timber that degrades fisheries and forests[10], and non-sustainable agriculture and land-use that make future land-use less effective[33,34]. We show that over-harvesting temporarily keeps the discount rate higher, but is followed by a period of lower discount rates during the same period in which society makes the transition to sustainable harvesting. Specifically, during the transition, the rates of economic growth and discount could be much lower than their rates before and after the transition. Therefore, the more our society over-harvests natural resources today, the more policymakers should discount the near future, but the less they should discount the far future. Furthermore, we prove a theorem implying that postponing or slowing the transition to sustainable harvesting cannot prevent the ultimate declines in the cumulative discount. Accordingly, we develop a discount formula that incorporates the changes in the harvest methods, which, in turn, dictates significantly higher net costs due to long-lasting environmental damages.

## Results

**Theoretical framework**. We consider a social welfare function, $U^T$, that depends on the provision of some natural resource at the global scale, $f(t)$, and on the consumption of the other goods, including manufactured goods, $c(t)$ (Methods, Eq. 4). In turn, the dynamics of the $c(t)$ and $f(t)$, together with $U^T$, determine the social rate of discount, $\delta(t)$, which specifies the rate at which goods should be discounted by a social planner whose objective is to maximize social welfare[15,16,35]. To define the social rate of discount (hereafter, the discount rate), we adopt a well-established framework[12,14,16,32,36,37] and we assume that it is given by the rate of decline in the marginal contribution of consumption to social welfare (consumption rate of discount). Specifically, we consider a given currency unit, a dollar, that enables the consumption of exactly $\mu\varepsilon$ units of the natural resource and $(1 - \mu)\varepsilon$ units of the other goods, where $0 \le \mu \le 1$ and $\varepsilon$ is very small. Accordingly, the discount factor at time $t$ is given by the number of dollars needed at present to compensate for a lack of one dollar at time $t$. Note that the choice of

$\mu$ does not affect the value given to future goods, and therefore, it does not affect the policy and/or the management decisions; rather, $\mu$ determines the units and it affects only the relative role of the discount factor and the prices in determining the value of future goods[36,37]. In turn, we show that this implies that the discount rate, $\delta(t)$, and the cumulative discount $\Delta(t) = \int_0^t \delta(t')\mathrm{d}t'$, are given by Eq. 5, and the prices of the natural resource and of the other goods are given by Eq. A10 (Methods and Supplementary Note 1). Specifically, the discount rate and the prices depend on the substitutability of the natural resource and the other goods, which is incorporated in the social welfare function. In Supplementary Note 2, we derive specific expressions for the discount rate and for the prices in two cases, one in which the natural resource and the other goods are non-substitutable (Eqs. B5, B9), and one in which they are partially substitutable (Eqs. B12, B15, B16).

In turn, the novel part of this study comes from endogenizing the dynamics of $c(t)$ and $f(t)$ by modeling how they depend on the harvest methods used globally (see Methods). This allows us to examine how the discount factor and the prices depend on changes in harvest methods. We assume that, if the harvest methods do not change, then $c(t)$ and $f(t)$ increase exponentially at fixed rates, $g_c$ and $g_f$, respectively, due to exogenous factors such as technological developments and exogenous environmental changes; however, changes in the patterns of harvest may affect $c(t)$ and $f(t)$, thereby affecting the discount rate over time (see Methods). This approach builds on and generalizes previous studies that considered $f(t)$ and $c(t)$ that grows exponentially irrespective of the harvest[32,37]. Specifically, note that $c(t)$ and $f(t)$ characterize the total provision of the goods at the global scale, and accordingly, we consider a large ecosystem that comprises a large number of distinct regions (Fig. 1). This ecosystem may be, for example, the entire planet's aquatic ecosystem, where each region is some local fishery providing fish; the forest area on a given continent, where each region is a single forest providing timber; or the area that can be used for agriculture worldwide, where each region is a local geographic area comprised of agricultural fields. We are interested in the long-lasting effects of harvesting on the provision of the natural resource, and therefore, we focus on irreversible degradations of the ecosystem, rather than on temporary fluctuations of the resource stock. These degradations may occur, for example, if some ecosystem services are permanently lost[5] or if the ecosystem that provides the renewable resource collapses or undergoes an irreversible regime shift in some of its regions, such as occurs in eutrophication and deforestation[3,4,10]. We assume that higher rates of non-sustainable harvest (higher $H_n$) result in a greater provision of the natural resource at the time of harvest but also result in a higher degradation of the ecosystem (Eq. 6, see Methods). Specifically, we assume that a given portion of the global ecosystem, $H(t)$, is being harvested in year $t$, while some portion of the ecosystem, $H_n(t)$, becomes degraded during that year due to non-sustainable harvest, and cannot be used for harvest thereafter (Fig. 1). For example, $H_n(t)$ may characterize the portion of the global fish or timber stock that is lost due to the collapse of fisheries or the irreversible degradation of forests worldwide in year $t$[38]. For another example, $H_n(t)$ may characterize the persistent reduction in the yield of crop caused by the degradation of vital ecosystem services and the increase in the persistence of pests[33,34]. In turn, $H(t)$ and $H_n(t)$ are determined by the various harvest methods used in the system (see Methods).

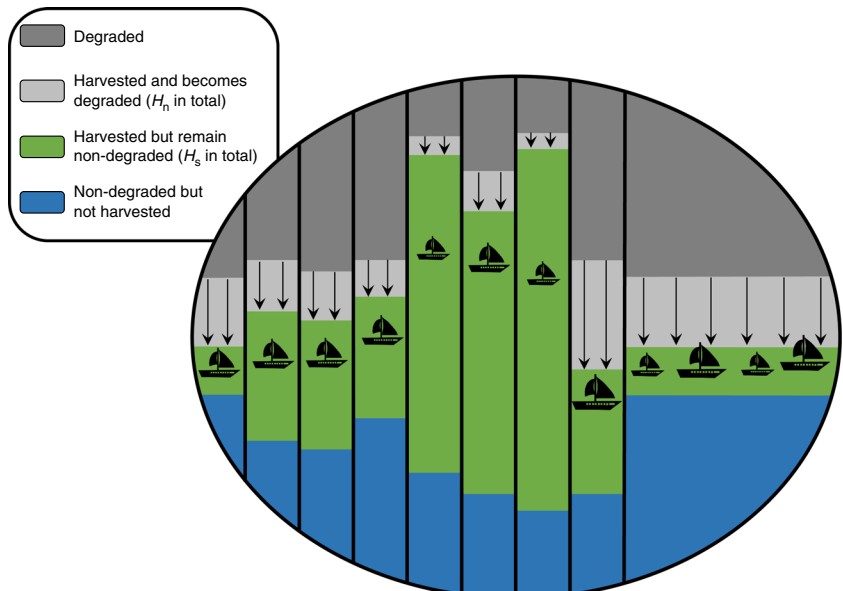

**Fig. 1** Schematic illustration of the model. Demonstrated is the state of the system at the global scale (e.g., the entire planet's marine area, forest area, or agricultural area) in a given year. The dark-gray area characterizes the part of the system that is degraded due to former non-sustainable harvesting. The light gray area with the arrows characterizes the part of the system that is being harvested non-sustainably and will be degraded starting next year (total dark-gray area is given by $H_n$). The green area with the fishing vessels characterizes the part of the system that is being harvested sustainably and will remain non-degraded next year (total green area is given by $H_s$). (Note that the total area under harvest, $H$, is given by the green and the light gray areas combined, $H = H_n + H_s$). The blue area characterizes the part of the system that is not degraded but is still not being harvested. We assume that the spatial scale of the system is very large, and therefore, the recovery of the degraded areas due to migrating biota from other regions is negligible and the total degraded area increases over time. Each year, $H_n$ and $H_s$ are determined by the aggregate management by all the managers. We assume that managers may be subject to different externalities in distinct regions, e.g., some regions are managed by a single manager that dictates the harvest method, while some regions are shared (open-access), and all managers are free to harvest in them (rightmost region). The variables $x_1$ and $x_2$ (Eqs. 7 and 8) characterize the total non-degraded areas (blue, green, and light gray) in the managed and in the shared regions, respectively

In the legend:
- Degraded
- Harvested and becomes degraded ($H_n$ in total)
- Harvested but remain non-degraded ($H_s$ in total)
- Non-degraded but not harvested

To examine the effect of over-harvesting on the natural resource and on the discount rate, we compare scenarios in which over-harvesting occurs to scenarios in which it does not. We consider two approaches. First, we consider a competitive market approach in which we compare the optimal solution that maximizes social welfare with the solution that emerges in a model of a perfectly competitive market with externalities (Figs. 2 and 3). Specifically, the competitive market includes managed regions that have a single manager (e.g., landowner, government), and shared regions in which multiple managers are free to harvest (e.g., open-access) (see Methods). Second, we consider a more general approach in which we compare the dynamics that emerge when the harvest is entirely sustainable with the dynamics that emerge following various ad hoc choices of non-sustainable harvest functions (Theorem and Fig. 4).

**Over-harvesting is followed by declines in the discount rate.** Following the optimal solution in which the harvest functions maximize social welfare, two phases emerge along the time axis (Fig. 2a, b). In the first phase ($t < t_0$), $c(t)$ is initially small,

and the harvest rates are limited due to the direct cost of harvesting (Methods, Eq. 9). Over time, as $c(t)$ increases, the direct cost plays a less significant role, and the harvest rates increase. Consequently, $f(t)$ increases at a rate that is greater than $g_f$, and the discount rate approximately follows Ramsey's formula. In the second phase ($t > t_0$), the entire ecosystem is under harvest (either sustainable or non-sustainable). Therefore, the society cannot increase $f$ via harvesting without increasing the non-sustainable harvest (i.e., increasing $H_n$), which would negatively affect the resource's future provision. Consequently, the non-sustainable harvest decreases exponentially and $c(t)$ and $f(t)$ increase at approximately the rates of their technological developments, namely, $\dot{c}/c \approx g_c$ and $\dot{f}/f \approx g_f$. This implies that, if $g_f < g_c$, the discount rate in the second phase is lower than it was in the first phase (Eqs. B6, B13, Supplementary Note 2). Note that the optimal solution comprises non-sustainable harvest ($H_n > 0$) because an increase in $f$ at a given time has a greater effect on welfare than the same increase at a later time; the lower the discount rate, the lower the rate of non-sustainable harvest.

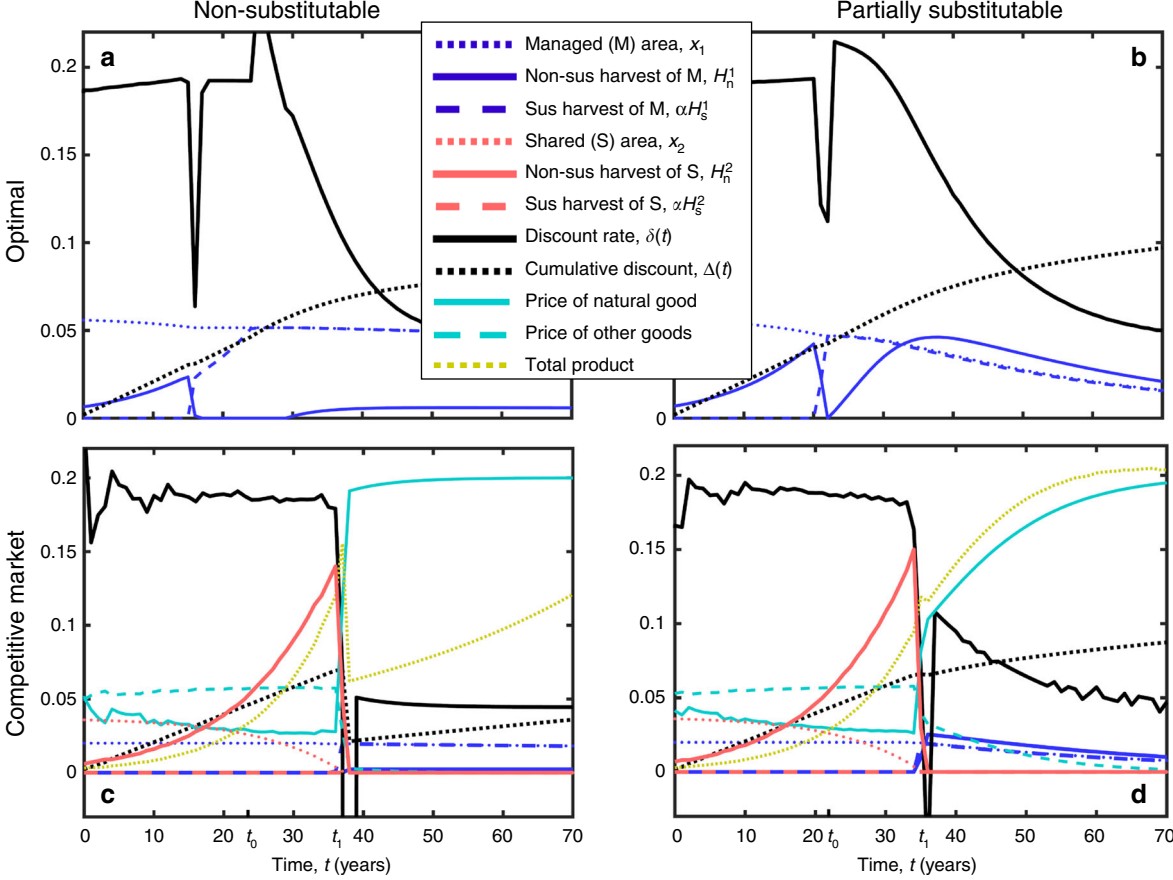

**Fig. 2** Over-harvesting extends the period during which the discount rate is high, but it is followed by sharp declines in the discount rate and the cumulative discount. Panels **a** and **b** demonstrate the optimal harvest of the natural resource from a social planner's perspective, where the natural resource and the other goods are either non-substitutable (**a**, Eq. B2) or partially substitutable (**b**, Eq. B10). In the early stages, harvesting activity increases exponentially and the discount rate is high. Approximately at time $t_0$, when harvesting is occurring in the whole system ($H_s + H_n = x_1 + x_2$), the total harvest stops increasing and the discount rate decreases. Next, panels **c** and **d** demonstrate harvesting in a competitive market in which some of the regions are shared. The parameters and utility functions used in panels **c** and **d** are identical to those used in panels **a** and **b**, respectively. The period during which the discount rate is high is extended until $t = t_1$ due to over-harvesting of the natural resource in the shared regions (compare panel **a** with panel **c**, and compare panel **b** with panel **d**). However, this period is followed by a rebound in which harvesting declines and the discount rate and the cumulative discount drop. In addition, around $t = t_1$, the price of the natural resource increases and the total product decreases. Note that, in accordance with the theorem, the cumulative discount approaches lower values if the harvest is determined by the market. Scaling: the harvest rates are given in (years)$^{-1}$, the total non-degraded areas are given in units showing the maximal annual sustainable yield ($ax_1$ and $ax_2$), and $\Delta$ is given by 100 times the value on the y-axis. The parameter values used are within their realistic ranges (Methods). Parameter values and Source data are provided as a Source Data file

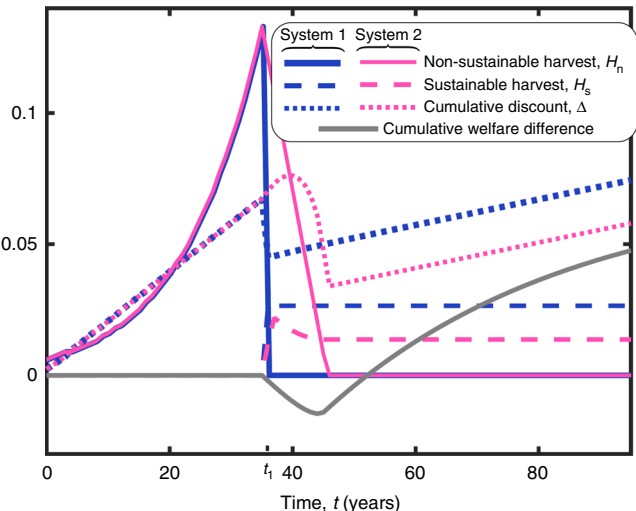

**Fig. 3** Social welfare and the cumulative discount are ultimately lower if the transition to sustainable harvest is more gradual. Demonstrated are the aggregate non-sustainable harvest, $H_n(t)$ (solid lines); the aggregate sustainable harvest, $H_s(t)$ (dashed lines); and the cumulative discount, $\Delta$ (dotted lines), for two systems. System 1 (blue) follows the market solution, in which society abruptly stops harvesting non-sustainably at $t = t_1$. System 2 (purple) follows the same dynamics until $t = t_1$, but then, society gradually shifts to sustainable harvest. The gradual transition postpones the decline in the cumulative discount, but ultimately, it declines to an even lower value than its value in system 1. Moreover, the cumulative welfare, $U^t$, in system 1 is initially smaller, but it ultimately becomes greater compared to system 2 (gray). Harvest rates are given in units of (years)$^{-1}$, and $\Delta$ is given by 100 times the value on the y-axis. The parameters are the same as in Fig. 2c (Parameter values and Source data are provided as a Source Data file)

In turn, in the competitive market solution (see Methods), the rate of non-sustainable harvest is higher than the socially optimal rate, namely, the solution exhibits over-harvesting (Fig. 2c, d). Specifically, the harvest is still primarily sustainable in the managed regions but is non-sustainable in the shared regions. The total area under (non-sustainable) harvest in the shared regions increases over time, and consequently, $f(t)$ continues to increase over an extended period of time, which postpones the decline in the discount rate. Eventually, however, at time $t = t_1$ (Fig. 2), the shared regions become entirely degraded and the total rate of non-sustainable harvest declines. In turn, the period during which managers over-harvest ($t < t_1$) is followed by declines in the discount rate, the cumulative discount ($\Delta$), total production (Eq. A11), and the price of manufactured goods (Eq. A10). These declines are greater if the magnitude and/or duration of the over-harvesting are greater (e.g., if more regions are shared), and also if the natural and the manufactured goods are non-substitutable. Note that the optimal solution exhibits no declines in economic growth or in $\Delta$ because the social planner plans for the forthcoming constraints on the harvest by avoiding over-harvesting in the early stages ($t < t_0$); in the market solution, managers also take into account the forthcoming decline in $f$ and avoid non-sustainable harvesting in the managed regions prior to time $t = t_1$, but they still over-harvest in the shared regions. Also note that, in both the optimal and the market solutions, the harvest functions, as well as $c(t)$ and $f(t)$, do not depend on $\mu$ (only the discount and the prices do).

**Decline in the cumulative discount is unavoidable (theorem).** More generally, the following theorem shows that over-harvesting

may result in an increase in $\Delta$ in the short run, but ultimately, $\Delta$ would return to a lower value than it would have had if managers used optimal harvesting or only sustainable harvesting (see proof in Supplementary Note 3 and demonstration in Figs. 3 and 4). Specifically, a more gradual transition to using sustainable harvest methods may result in a more gradual decline in $\Delta$, but the ultimate magnitude of the decline must exceed that of the incline in $\Delta$ that occurred formerly due to the over-harvesting (Figs. 3 and 4a). In particular, the theorem shows that the result is robust and does not depend on specific assumptions and parameters. It applies not only in the competitive market model but also in the more general case in which non-sustainable harvest is used instead of more sustainable harvest.

**Theorem**. *Assume that the social welfare, $U^T$, is given by Eq. 4, where $f(t)$ is given by Eq. 6, and $c(t)$ is given by Eq. 9 with $C_1 = C_2 = $ constant (Methods). Also, assume that $u(c,f)$ is monotonically increasing and twice differentiable with respect to both of c and f, and all of its second partial derivatives are non-positive (namely, an increase in c or f does not cause another increase to be more beneficial). In addition, we consider $g_f = 0$ and assume that as $c \to \infty$ while f remains fixed, $u_c/u_f \to 0$ (the price of c approaches 0), $u_{cc}/u_{ff} \to 0$ and $u_{cf}/u_{ff} \to 0$. (Alternatively, we consider $g_f > 0$ and assume that u satisfies the conditions of Lemmas 2B and 3). Finally, we assume that, for sufficiently large t, $cu_{fc}/u_f$ and $fu_{ff}/u_f$ are monotone with respect to t. (All these assumptions are satisfied if u is given by Eqs. B2, B10 with $\eta > 1$, or various other standard forms[32,37].)*

*Denote $\Delta_{opt}$ as the cumulative discount (Eq. 5) that emerges following the optimal harvest. Namely, the non-negative harvest functions maximize social welfare (max $U^T$ subject to Eqs. 6–9 where $T \to \infty$; see Methods). Next, denote $\Delta_{market}$ as the cumulative discount that emerges where the harvest functions are determined if each manager aims to maximize her/his own profit and the non-sustainable harvest may be higher than its socially optimal level (Methods). Then, there exists a time $t_c$ such that $\Delta_{market} \leq \Delta_{opt}$ for all $t \geq t_c$. Furthermore, denote $\Delta_{sus}$ as the cumulative discount that emerges following optimal harvest while excluding non-sustainable harvest ($H_n = 0$). Then, for any $\Delta$ that emerges if non-sustainable harvest occurs ($H_n(t) > 0$) between times $t_0$ and $t_1$, there exists $t_c > t_1$ such that $\Delta(t_c) \leq \Delta_{sus}(t_c)$.*

**A new discount formula**. The theorem shows that an upper bound on $\Delta(t)$ in the long run is given by $\Delta_{sus}(t)$, the cumulative discount that would have occurred if managers used only sustainable harvest, which increases at a rate given by $\delta_{sus}$ (Fig. 4 and Supplementary Note 2). Also, the present value of $\Delta_{sus}$ is below $\Delta$ because over-harvesting already has occurred prior to today. Specifically, $\phi_0 = \Delta(0) - \Delta_{sus}(0)$ reflects the negative shock to $\Delta$ that must occur during the transition to sustainable harvest methods due to the prior over-harvesting. It follows that, if $t$ is sufficiently large and $\delta_{sus}$ is constant, then

$$\Delta(t) \leq \delta_{sus} t - \phi_0. \tag{2}$$

Particularly, if the discount rate has been $\delta_{today} > \delta_{sus}$ due to non-sustainable harvest during the last $t_0$ years, and if $\delta_{today}$ and $\delta_{sus}$ have been constants, then $\phi_0 = (\delta_{today} - \delta_{sus})t_0$.

**The correction to the value of future goods is significant**. Next, we calculate the correction to the value of future natural goods as dictated from Eq. 2. Specifically, we compare the value dictated by the formula to the value dictated by a benchmark policy that assumes that the rate of increase in the provision of the natural resource would remain $g_c$ for the next $\tau$ years and decrease to $g_f$ thereafter[12]. Namely, this benchmark policy ignores the negative shock and simply uses a

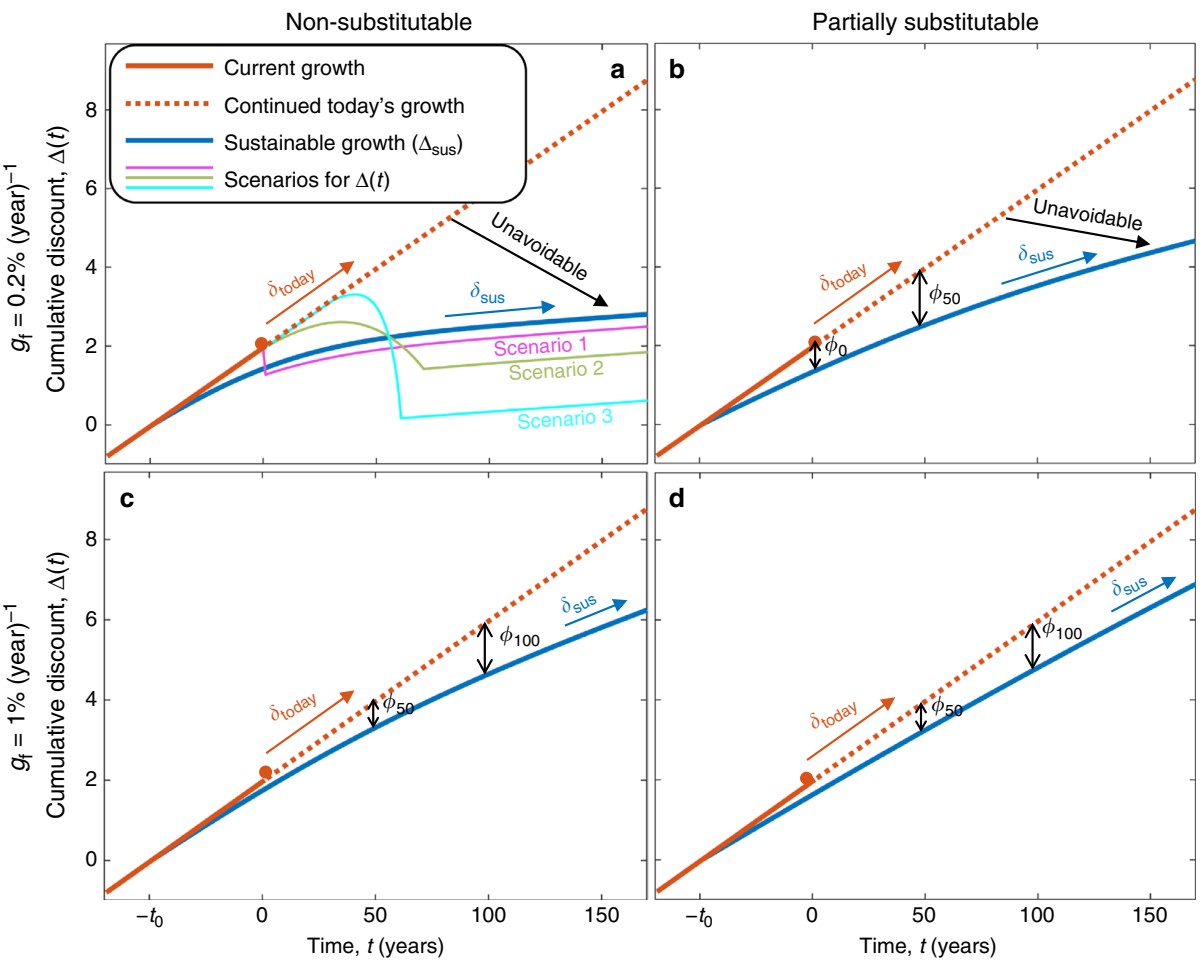

**Fig. 4** The decline in the cumulative discount is unavoidable (demonstration of the theorem). At some point in time, $-t_0$, some planetary boundaries for harvest have been approached, and the rate of discount that would have occurred if managers used only sustainable harvesting has decreased from $\delta_{today}$ to $\delta_{sus}$ (blue lines, $\Delta_{sus}$). Nevertheless, due to over-harvesting, the economy grew faster and the cumulative discount, $\Delta(t) = \int_0^t \delta(t')dt'$, continued to grow at a higher rate, $\delta_{today}$ (solid orange lines), at least until today ($t = 0$). In turn, the future value of $\Delta(t)$ depends on the future harvest patterns. If over-harvesting continues, the discount rate might remain close to $\delta_{today}$ for several years or decades (dotted orange lines). But in the longer run, according to the theorem, $\Delta$ has to decrease below the blue curve that characterizes $\Delta_{sus}$, regardless of how the resource is being harvested. This is also demonstrated for three scenarios in panel **a**: In scenario 1, the non-sustainable harvest stops today, while in scenarios 2 and 3, the non-sustainable harvest continues for a few decades and then declines gradually. Also, note that $\Delta_{sus}$ increases at a rate $\delta_{sus}$, so if one assumes that the discount rate remains $\delta_{today}$ for the next $\tau$ years and becomes $\delta_{sus}$ afterward, then he/she needs to subtract at least $\phi_\tau$ to obtain the correct $\Delta$ (Eqs. 2 and 3). (The value of $\phi_\tau$ is demonstrated in Fig. 5.) We assume that $u(c, f)$ is given by Eq. B5 (non-substitutable goods) in panels **a** and **c**, and by Eq. B12 (partially substitutable goods) in panels **b** and **d**. In turn, the scenarios are calculated for three different choices of $H_n(t)$, where the dynamics follow Eqs. 6–9 with $H(t) = x_1(t) + x_2(t)$ for all $t$. The parameter values used are within their realistic ranges (Methods). Parameter values and Source data are provided as a Source Data file

discount rate given by $\delta(t) = \delta_{today}$ if $t \le \tau$ and $\delta(t) = \delta_{sus}$ if $t > \tau$. In turn, we would like to calculate the correction to that policy due to the negative shock to $\Delta$. Note that the inevitable decline in the future value of the cumulative discount, $\phi_0$ (Eq. 2), is what policymakers need to incorporate due to the over-harvesting that has already occurred before $t = 0$. But if the discount rate remained $\delta_{today}$ for the next $\tau$ years, until $t = \tau$, then the lower bound on the negative shock, $\phi_\tau$, would be greater than $\phi_0$ and given by (Fig. 4)

$$\phi_\tau = \int_{-t_0}^{\tau} \left( \delta_{today}(t) - \delta_{sus}(t) \right) dt. \tag{3}$$

This greater shock would compensate for the $\tau$ years with the higher discount, such that, in the long run, $\Delta(t)$ would still satisfy Eq. 2. Note that the shock may be gradual and spread over many years, but this decline in $\Delta(t)$ eventually occurs (Theorem, Figs. 3 and 4).

Therefore, this shock implies that the correct discount factor should be greater by a factor of at least $\exp(\phi_\tau)$ compared to the one implied by the benchmark policy. Namely, ignoring this shock and simply considering the benchmark policy would result in under-estimating the value of future natural goods by a factor of at least $\exp(\phi_\tau)$ (Fig. 5). In turn, the magnitude of $\phi_\tau$ depends on the substitutability of the natural resource and the other goods, as well as on the exogenous growth rates, $g_c$ and $g_f$ (Supplementary Note 2). For example, if the natural resource is non-substitutable (Eq. B2), then $\delta_{sus}$ is given by Eq. B5 and $(\delta_{today} - \delta_{sus}) \to \eta(g_c - g_f)$ as $t \to \infty$ (Eq. B7). Expressions that result from other utility functions are given in Supplementary Note 2 and in the literature[32,37]. These expressions enable us to quantify $\exp(\phi_\tau)$ and examine how it depends on the parameters (Fig. 5). For example, if $g_f = 1\%$ year$^{-1}$, $g_c = 2\%$ year$^{-1}$ [36] and $\tau = 50$ years, then the value of future goods before the adjustment is underestimated by a factor greater than two ($\exp(\phi_\tau) > 2$), and this factor is greater if $g_f$ is smaller or if $\tau$ is larger.

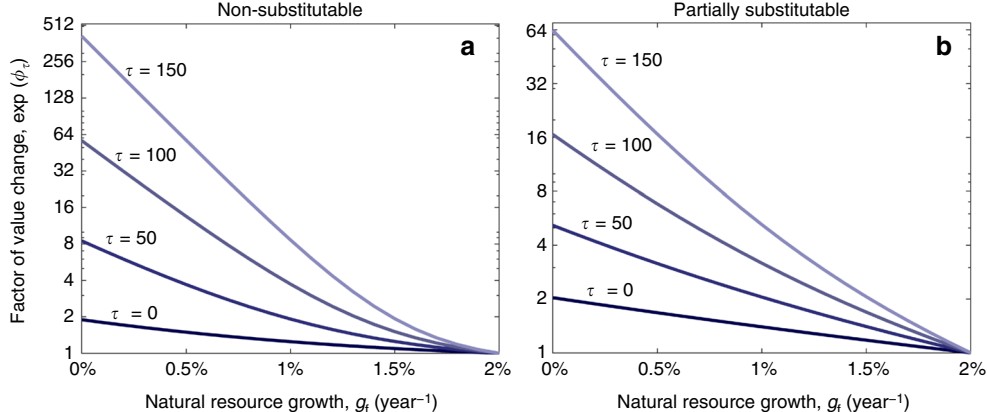

**Fig. 5** Endogenizing changes in harvest patterns implies a larger discount factor and higher values for future environmental goods. If a policymaker considers a gradual transition to sustainable harvest that would occur within $\tau$ years, then he/she may consider a sustainable discount rate, $\delta_{sus}$, starting from year $\tau$. In addition, however, he/she needs to add to $\Delta$ another factor, $\phi_{\tau}$, that accounts for the decline in the cumulative discount that will follow due to over-harvesting prior to time $\tau$ (Eqs. 2, 3 and Fig. 4). This factor may impose significantly higher values on future goods, e.g., over two times higher if $\tau = 50$ years and $g_f = 1\%$ year$^{-1}$ ($\exp(\phi_{\tau}) > 2$ in both panels **a** and **b**) and even significantly higher for higher values of $\tau$ or lower values of $g_f$. However, if the long-term provision of the natural resource continues to increase at the same rate as the other goods, i.e., $g_f = g_c = 2\%$ year$^{-1}$, then $\delta_{sus} = \delta_{today}$ and $\phi_{\tau} = 0$ (Eq. 3). The other parameter values are the same as in Fig. 4 (Parameter values and Source data are provided as a Source Data file)

## Discussion

After over-harvesting for decades, many societies around the world are beginning to transition to sustainable environmental management practices and sustainable harvest methods[11]. Our study shows that the transition to sustainable harvest methods after a period of over-harvesting is expected to result in a decline in social welfare, economic growth, and the discount rate. In particular, we show that the discount rate, or the social rate of discount, does not decline gradually to its sustainable asymptotic rate; rather, the transition to sustainable harvest may include a period during which the discount rate is far below its asymptotic level (Figs. 2–4 and Theorem). Note that several studies suggested that policymakers need to consider discount rates that decline gradually over time due to various mechanisms, including uncertainty in technological growth[16,20–23], slowdown in technological development due to environmental degradation[27,28], and declining production due to decline in the exploitation of natural resources[12]. In contrast, we showed here that the transition to sustainable harvest imposes a sharper, non-gradual decline in the cumulative discount (Figs. 2–4). The mechanism underlying this sharper decline is that the rate of increase in the provision of natural resources not only slows down, but must at some point become lower than it would be if over-harvesting had never occurred. In turn, social welfare depends on the provision of natural resources, and therefore, a decline in their provision implies a lower discount rate.

Our results also suggest that the calculations of the discount factor in the long run should not rely on simple extrapolations of the discount rates in the short run. Specifically, over-harvesting might continue for a couple of decades, which may keep the provision of natural resources high in the short run, but will ultimately result in an even lower provision of these resources. Therefore, continued over-harvesting may justify considering higher discount rates in the short run, but it also necessitates discounting the long run less (Fig. 4a). Ignoring the harvest-induced decline in the discount rate not only falsifies cost-benefit analyses, it also creates a bias: Over-harvesting increases the discount rate in the short run, which might unjustifiably bias the expectations of policymakers to anticipate higher future discount rates, which, in turn, is used to justify further exploitation. (This may also explain why policymakers should consider lower discount rates in the long run although there is no clear evidence

that the rate of return on capital will decline during the next 30–40 years[15].)

To correct for this bias and account for the future decline in the cumulative discount, we developed a new discount formula (Eqs. 2 and 3), which provides a simple way to estimate the increase in the present value of future goods due to the transition to sustainable harvest methods. Specifically, policymakers need to consider a cumulative discount, $\Delta(t)$ (Eq. 1), that is lower in the long run due to its decline during the transition to sustainable harvest. Although further over-harvesting may postpone the timing of the decline, we prove in the theorem that the decline eventually comes with a rebound as $\Delta(t)$ decreases even further: The more our society over-harvests, the lower $\Delta(t)$ ultimately becomes. Therefore, the expected decline in the cumulative discount must be at least as large as its former increase due to over-harvesting (Eq. 2, Fig. 4). In turn, this former increase is given by Eq. 3. The correction to discounting suggested by our formula is significant (Fig. 5), where adjustments of the order of magnitude implied by the formula suggest significant changes in climate policy, including significant emission cuts[6,36].

Note that the effect of harvest on discounting should be considered in addition to (not instead of) changes dictated by various other mechanisms and considerations. In particular, there is a controversy over the value of the rate of pure time preference, $\rho$, that should be used in environmental policies; some authors argue that policymakers should determine $\rho$ based on individual's preferences ($\rho \approx 3\%$ year$^{-1}$), but others argue that policymakers should use $\rho \approx 0$ based on considerations of intergenerational equity[6,16,19,39]. The value of $\phi_{\tau}$, however, does not depend on the value of $\rho$ and should be subtracted from $\Delta$ regardless of that choice. Similarly, uncertainty about technological development may imply that policymakers need to consider $\delta_{sus}$ that declines over time[16,20–23], which implies another decline in the cumulative discount on top of the one suggested here. Also, note that the future values of natural resources do not depend on the proportion given to their consumption in the currency unit, $\mu$. Specifically, their future values do not depend on whether they are accounted for as market or as non-market goods. Nevertheless, $\mu$ does affect the relative weights given to the discount factor and to the prices of natural resources in determining the resources' future values[32,36,37]. Specifically, ignoring the role of non-market natural resources in economic growth (considering a

small $\mu$) would imply that a change in the provision of these resources has a larger effect on their prices but a smaller effect on the discount factor (Supplementary Note 1). Therefore, focusing on the inevitable increase in the price of natural resources following their over-harvesting would result in the same conclusions and present an alternative approach to the one presented here. In particular, the adjustment $\exp(\phi_\tau)$ (Fig. 5) is due to the change in the discount factor, while the complementary change in the price (Fig. 2) introduces another adjustment to the future value of natural resources[36]. The total adjustment due to changes in both discount and prices does not depend on the choice of $\mu$, and would be $\geq \exp(\phi_\tau)$. The significant effect that the global transition to sustainable harvest has on the future value of natural resources suggests that climate policies should be determined jointly with other environmental policies.

## Methods

**Model overview.** We begin with describing a well-established framework[32,36,37] that specifies how social welfare and the discount rate depend on the provision of the natural resource over time, $f(t)$, and on the consumption of other goods over time, $c(t)$. Next, we specify how harvest at the global scale affects the dynamics of $f(t)$ and $c(t)$ (which would grow exponentially if the harvest functions are fixed). We complete the model by describing how the harvest strategies are determined by the various managers in a competitive market.

**Model of social welfare and the discount rate.** We consider a social welfare function that is given by the widely-used form[12,32,36,37]

$$U^T = \int_0^T u(c(t), f(t)) e^{-\rho t} dt,\qquad(4)$$

where $u(c, f)$ is the instantaneous utility that increases as $c$ and $f$ increase (Table 1), $\rho$ is a constant rate of pure time preference, and $T$ is a time horizon (we are

### Table 1 Table of symbols

*Objective function and discounting*

| Symbol | Meaning |
|---|---|
| $U^T$ | Social welfare |
| $u$ | Instantaneous utility |
| $\delta(t)$ | Discount rate (social rate of discount; consumption rate of discount) |
| $\rho$ | Rate of pure time preference |
| $\Delta(t)$ | Cumulative discount ($\delta(t) = d\Delta(t)/dt$) |
| $\mu$ | Portion of the natural resource in the curreny unit |

*Dynamical variables*

| Symbol | Meaning |
|---|---|
| $f(t)$ | Harvested natural resource for consumption |
| $c(t)$ | Consumed goods other than the natural resource |
| $x_1(t), x_2(t)$ | Total non-degraded areas of managed and of shared regions |

*Harvest functions (controls)*

| Symbol | Meaning |
|---|---|
| $H(t)$ | Portion of the system that is being harvested in year $t$ |
| $H_n(t)$ | Portion of the system that becomes degraded in year $t$ |
| $H_s(t)$ | Portion of the system that is harvested but remains non-degraded in year $t$ |
| $H_n^1(t), H_n^2(t)$ | Portion of the system that becomes degraded in managed and shared regions |
| $H_s^1(t), H_s^2(t)$ | Portion of the system that is harvested but remains non-degraded in managed and shared regions |

*Parameters of the dynamics*

| Symbol | Meaning |
|---|---|
| $\alpha$ | Fraction of the maximal amount of the resource that can be harvested via sustainable harvest ($0 < \alpha < 1$) |
| $g_f$ | Growth rate of natural resource yield per unit of non-degraded area |
| $g_c$ | Growth rate of production of the other goods |
| $C_1, C_2$ | Direct cost of harvest in managed and shared regions. |
| $\lambda$ | Ratio between the direct costs of sustainable and non-sustainable harvest |

interested in the limit $T \to \infty$). The distinction between the provision or consumption of the natural resource, $f(t)$, and that of the other goods, $c(t)$ is necessary here because, if the natural resource and the other goods are not entirely substitutable and the ratio between them varies over time, then social welfare depends on the ratio between $c$ and $f$ over time and cannot be written as a function of a single variable[29]. In turn, the substitutability is determined by the form of $u$[12,29,37]. For example, the goods may be non-substitutable, characterized by separable utility functions (Supplementary Note 2, Eq. B2), if one good cannot compensate for the lack of the other good (e.g., many cars cannot compensate for a lack of food). Alternatively, the goods may be partially substitutable (Eq. B10) if a sufficient amount of one good may compensate for the lack of the other good (e.g., many carrots can compensate for the lack of fish).

In turn, note that there are several candidates for quantifying the social rate of discount[15], including the consumption rate of discount and the social and private rates of return to investment. These three quantities are closely-related, and, in a perfectly competitive market, they become equal and reflect the marginal productivity of capital. In this study, as in numerous related studies[12,14,16,32,36,37], the focus is on the consumption rate of discount, which is the rate of decline in the marginal contribution of consumption to social welfare. In other words, the corresponding discount factor specifies how many units of consumption added at present would have the same effect on social welfare as a single unit added at time $t$. In turn when the welfare depends on multiple goods, the discount may depend on the particular good that the policymaker considers[31,36,37]. (This simply reflects the relative price changes of the goods.) Therefore, to define discount in our system, we consider a small, marginal perturbation to both $c$ and $f$. Specifically, we consider a given currency unit, a dollar, that allows the consumption of exactly $\mu\varepsilon$ units of the natural resource and $(1 - \mu)\varepsilon$ units of the other goods, where $0 \leq \mu \leq 1$ and $\varepsilon \ll c(0), f(0)$. Accordingly, we define the discount factor at time $t$ as the number of dollars needed at present to compensate for a lack of one dollar at time $t$. This implies that the discount rate, $\delta(t)$, is given by (Supplementary Note 1)

$$\delta(t) = \frac{d\Delta(t)}{dt} = \frac{(1-\mu)u_{cc}\frac{dc}{dt} + (1-\mu)u_{cf}\frac{df}{dt} + \mu u_{fc}\frac{dc}{dt} + \mu u_{ff}\frac{df}{dt}}{(1-\mu)u_c + \mu u_f} + \rho,\qquad(5)$$

where subscripts in this equation denote partial derivatives and the discount factor is given by $\exp(-\Delta)$. The right side of Eq. 5, without the term $\rho$, is due to the change in the marginal contribution of $c$ and $f$ to social welfare. (Note that, if $\mu = 0$ and $dc/dt = cg_c$, then Eq. 5 becomes the Ramsey's discount formula[14,16], $\delta(t) = \eta g_c + \rho$, where $\eta \equiv cu_c/u_{cc}$.) In turn, if $\mu$ reflects the portion in society's basket of goods allocated to consumption of the natural resource, then our definition is consistent with the way the marginal productivity of capital is measured, and the total product (e.g., GDP) is proportional to the total value of all the goods (Supplementary Note 1, Eq. A11). Alternatively, if we are interested in discounting some climate damage, then we can chose $\mu$ to be proportional to the cost that is due to the damage to the natural resource. Note, however, that the choice of $\mu$ only determines the units given to future goods and does not affect the value given to future goods. Specifically, if the proportion of damages to the natural resource differs from $\mu$, then one should consider the changes in relative prices in addition to discounting[36,37]. For example, several authors[37] considered a dual discounting framework in which the natural resource is discounted with $\mu = 0$ and the manufactured goods with $\mu = 1$, where the change in the relative price accounts for the difference; this approach is equivalent to the one presented here.

**Model of the dynamics and management of the natural resource.** Next, we specify how the harvest methods of the renewable natural resource at the global scale determine the dynamics of $c(t)$ and $f(t)$ (Fig. 1). Note that the aggregate harvest functions at the global scale are determined by the various harvest methods used at the local scale. In turn, at the local scale, a non-sustainable harvest in a given area during a given year yields $\beta$ units of the natural resource per unit area, but the ecosystem in that area becomes degraded and ceases to yield resources thereafter. In turn, sustainable harvest in a given area yields less resource ($\alpha\beta$ units, where $0 < \alpha < 1$ is a constant), but the area remains fully functional for future use. For example, non-sustainable harvest may include aggressive fishing methods that inflict irreversible damage on fish populations and their habitats, while sustainable harvest implies sustaining fish populations and harvest at the fish growth rate, while also using methods that preserve the habitat and the age and size structures of the fish[38]. In turn, the productivity of the natural resource per unit area, $\beta$, may increase due to technological developments but may also decrease due to other environmental changes, such as climate change. Accordingly, we assume that $\beta(t) = \beta_0 \exp(g_f t)$, where $\beta_0 = \beta(0)$ and $g_f$ is the rate of change in productivity. It follows that the total amount of the natural resource harvested globally at time $t$ is given by

$$f(t) = (\alpha H_s(t) + H_n(t))\beta_0 e^{g_f t},\qquad(6)$$

where $H_n$ is the area that is non-sustainably harvested in year $t$ (becomes degraded and cannot be harvested thereafter), and $H_s(t)$ is the area that is being sustainably harvested and remain non-degraded in year $t$ ($H_s(t) = H(t) - H_n(t)$).

In turn, we distinguish two types of regions: those that have a single manager, and those that are shared such that all managers are free to harvest. Ultimately, the harvest methods used by all managers determine the total areas that become degraded at time $t$ in the managed and in the shared regions at the global scale,

$H_n^1(t)$ and $H_n^2(t)$, respectively $\left(H_n = H_n^1 + H_n^2\right)$. Accordingly, the total non-degraded areas in all managed regions, $x_1$, and in all shared regions, $x_2$, decrease due to non-sustainable harvest as follows:

$$\frac{dx_1}{dt} = -H_n^1(t), \tag{7a}$$

$$\frac{dx_2}{dt} = -H_n^2(t). \tag{7b}$$

Moreover, the harvest functions are constrained by the non-degraded areas:

$$H_s^1(t) + H_n^1(t) \leq x_1(t), \quad H_s^2(t) + H_n^2(t) \leq x_2(t) \tag{8}$$

for all $t$, where $H_s = H_s^1 + H_s^2$.

In turn, we assume that harvest comes with a direct cost as more labor and resources are directed toward harvesting. We incorporate this direct cost as a reduction in $c(t)$, which would otherwise grow exponentially at an exogenous rate $g_c$ due to technological developments. Specifically, we assume that $c(t)$ is given by

$$c(t) = c_0 e^{g_c t} - C_1(x_1) \cdot \left(H_n^1 + \lambda H_s^1\right) - C_2(x_2) \cdot \left(H_n^2 + \lambda H_s^2\right), \tag{9}$$

where $C_1$ and $C_2$ are the direct costs of harvesting (in units of $c$), and $\lambda$ is the ratio between the direct costs of non-sustainable and sustainable harvest.

**Model of the competitive market**. It remains to specify how the harvest strategies of the managers at the local scale are determined, and how these strategies determine the harvest functions at the global scale, $H_s^1(t)$, $H_s^2(t)$, $H_n^1(t)$, and $H_n^2(t)$. We are interested in comparing two types of solutions: The optimal solution that maximizes the social welfare, and the market solution that emerges in a competitive market. The optimal solution is found via the maximization of the social welfare (Eq. 4) subject to the constraints given in Eqs. 6–9. In turn, to define the market solution, we consider a competitive market in which each manager aims to maximize her/his own utility. Specifically, we consider a well-established framework in which the market is perfectly competitive, such that, if property rights are defined everywhere and there are no externalities, the market solution coincides with the optimal solution[12,14,31,40–42]. In turn, the market solution depends on the form of the externalities for the various managers, namely, it depends on how non-sustainable harvest by a given manager affects the ecosystem in regions managed by other managers.

To define the externalities, we distinguish between managed regions and shared regions (Fig. 1). Each managed region is managed by a single manager who determines the harvest method, which may vary anywhere between using only sustainable methods and using only non-sustainable methods. In turn, the harvest method in a given region determines the portion of the region that is harvested and the rate at which the region becomes degraded (Fig. 1). We assume that the management in a given managed region has no externalities as it affects only the degradation level in that region. In turn, the shared regions are managed by a very large number of managers, each of whom is free to harvest without restrictions there. Specifically, we assume that each manager ignores the effect of her/his actions on the future provision of the resource in the shared regions and considers only her/his instantaneous benefit and cost from the harvest. Consequently, the managers have the incentive to increase non-sustainable harvest in the shared regions until the price of the natural resource equals the direct cost of the harvest. These considerations enable us to find the market solution that is given by the unique Nash equilibrium (see the section Numerical methods). In particular, the perfectly competitive market assumption implies that the management in the managed regions is socially optimal under the constraint given by the management in the shared regions. Note that, without shared regions ($x_2 = 0$), there are no externalities and the market solution coincides with the optimal solution.

**Numerical methods**. The numerical results showing the optimal and market solutions are demonstrated in Figs. 2 and 3, system 1. The optimal solution is given by the unique set of non-negative aggregate harvest functions, $H_s^1(t)$, $H_s^2(t)$, $H_n^1$, and $H_n^2(t)$, that maximize social welfare: $\max U^T$ (Eq. 4) in the limit $T \to \infty$, where $c(t)$ and $f(t)$ are given by Eqs. 6 and 9, subject to the constraint given in Eqs. 7 and 8. (Note that using the social welfare function given in Eq. 4 with a constant $\rho$, and considering deterministic dynamics of $c$ and $f$, guarantee that the optimization problem is time consistent and has a unique solution[12,37].) In turn, the market solution is determined by a perfectly competitive market where each manager maximizes her/his own profit. Specifically, consider the set of non-negative harvest functions that maximize utility, $\max U^T$ (Eq. 4) as $T \to \infty$, subject to the constraint given by Eqs. 7a and 8 and the constraint $dx_2/dt = X(t)$. Then, the market harvest is given by the unique solution that satisfies $X(t) = H_n^2(t)$ (consistency criterion).

We used algorithms that find the exact solutions provided that the resolutions are sufficiently fine. Specifically, to find the optimal solution numerically, our algorithm uses Stochastic Programming with backward induction (Supplementary Note 4)[43,44]. (Note that the model's dynamics are deterministic but the general method is still called stochastic.) To find the market solution, our algorithm also uses Stochastic Programming to solve for a given value of $X$. But it finds a solution multiple times, each time for a different value of $X$, until it finds the solution that

satisfies the consistency criterion. These algorithms are coded in C/C++ and are described in detail in Supplementary Note 4.

In turn, in the results shown in Fig. 3, system 2, as well as in Figs. 4 and 5 and in the graphical tool, we assume that the dynamics of $c$ and $f$ follow Eqs. 6–9, but we consider harvest functions that are not given by either the optimal solution or the market solution. In Fig. 3, system 2, we consider harvest functions that follow the market solution until $t = t_1$ and after $t = t_1 + 10$, but between these times, the non-sustainable harvest decreases gradually from its maximal level to zero. In Fig. 4, we calculate $\Delta_{sus}$, which is the cumulative discount that emerges if the harvest is entirely sustainable, namely, $H_n = 0$ and $H_s = x_1 + x_2$ if $t > 0$. Also, in Fig. 4a, we consider three scenarios in which the non-sustainable harvest is higher in the beginning but eventually approaches zero, while $H_n + H_s = x_1 + x_2$.

After we determine the harvest functions, the functions $c(t)$ and $f(t)$ are calculated according to Eqs. 6 and 9. In turn, we calculate the discount rate and the cumulative discount according to Eq. 5 (where the cumulative discount is the integral over time of the discount rate). Specifically, for the case in which only sustainable harvest is used ($\Delta_{sus}$ in Fig. 4), the discount rates are calculated in Supplementary Note 2 and are given by Eqs. B5 and B12. The prices are given by Eq. A10, and the total product is given by Eq. A11. All of these equations are derived in Supplementary Notes 1, 2.

**Choice of parameters**. The parameter values used for all of the numerical simulations, which are given in the Source Data file, are within their realistic ranges. The rate of technological growth is around 1.5–2.0% year$^{-1}$ in developed countries and is higher in some developing countries[16,45]. In turn, the rate of growth in the yield per unit of sustainable harvest, $g_f$, depends on the specific natural resource, where values that were considered in the literature vary from $g_c$ down to much lower (even negative) values[32,37]. Next, the value of $0 \leq a \leq 1$ (unitless) also depends on the particular system. In a fishery, for example, if non-sustainable harvest would imply catching all the fish and sustainable harvest would imply keeping the fish population size fixed, then $a$ would be the growth rate of the fish (i.e., 2% year$^{-1}$ for large fish and higher rates for smaller fish)[38]; In agriculture, sustainable management implies the use of environmentally friendly pest control methods and effective water management, which may result in a comparable crop yield ($\alpha \lesssim 1$), but may be more expensive ($\lambda > 1$)[33,34]. In turn, the ratio between $c(t)$ and the direct costs, $C_1$ and $C_2$ (Eq. 9), determines the relative portion of $c$ that is needed per unit of harvest. Specifically, $c$ (and thus the ratio) is initially small but increases due to technological changes. Also, $C_1$ and $C_2$ may vary with $x_1$ and $x_2$ if the cost varies among regions (e.g., if near-shore regions are depleted, the average direct cost of harvest may increase). Next, note that $0 \leq \mu \leq 1$ (unitless) can be chosen arbitrarily by the policymaker, as it does not affect the harvest strategy and the future value of the natural resource; rather, it determines the currency unit, which, in turn, determines the relative role of the discount and the price in determining the future value of the natural resource. A reasonable choice would be the portion in the basket of goods of the natural resource (e.g., the portion of agricultural products in consumption is ~5% in the United States and is higher in various developing countries), but $\mu$ may be higher if non-market goods are incorporated. Finally, a variety of utility functions that incorporate both $c$ and $f$ were suggested in the literature[12,32,37], including the two that are used here (Eqs. B2, B10)[12], where estimates of $\eta$ vary between 1 and 3 (unitless)[16,41,45], and suggested values for $\rho$ varies between 0 and 3% (year$^{-1}$)[6,16,19,45].

**Analytical and theoretical analysis**. The general discount formula (Eq. 5) is derived in Supplementary Note 1. The discount formulas for the special cases presented in the figures are derived in Supplementary Note 2. The proof of the theorem is given in Supplementary Note 3.

## Data availability
No datasets were generated or analyzed during the current study. All the data needed to reproduce the results is given in the paper. In particular, the parameter values used for each figure are given in the Source Data file. These parameter values are taken from the references that are cited in the Methods section.

## Code availability
The algorithm that we used for finding the optimal and market solutions (Figs. 2 and 3) is described in Supplementary Note 4. The C/C++ code used for generating Figs. 2 and 3 as well as the Matlab code used for generating Figs. 4 and 5 are available as a supplementary code.

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

## Acknowledgements
The author sincerely thanks Charles Perrings for his valuable comments on the paper. The author thanks SAL MCMSC, CLAS and SHESC, ASU, for funding (no. DN5-1057).

## Author contributions
A.L. designed the research, composed the model, conducted the research, and wrote the paper.

## Additional information

**Competing interests:** The author declares no competing interests.

