## [Peer Review File · Nature Communications]

Reviewers' comments:

Reviewer #1 (Remarks to the Author):

General comments

The article highlights that although valuing natural systems and has been a major focus of debate in recent decades, it is still unclear how the environment and its management affect the discount factor. The article presents a model to show how large changes in the harvest of natural resources affect the discount factor via non-linear effects on social welfare. Specifically, it is shown that over-harvesting temporarily keeps the discount rate higher, but is followed by a period of lower discount rates when society adjusts to sustainable harvest. It is concluded that hence when discounting the far-future, it should be used the derived adjusted discount formula implying significantly higher net costs due to future environmental damages.

In my view, the article is well-written and introduces a relevant improvement when endogenizing harvest of natural resources to affect the discount factor. In that sense, if considered that discounting should be applied (there is literature referenced on the optimal/acceptable... levels of discount factor, but there are also advocates of the zero discount factor, based on inter-generational equity, for whom these type of discussion do not even make sense) probably it should be examined and discussed why this improvement or change in assumptions (i.e. endogenizing harvest) is the only or the most necessary one, considering that one could have also related to many other aspects or variables (values and shape of the discounting, the role of technology, of growth, of how goods interrelate, etc.). Related to this, I wonder about the importance of finding a mathematic relationship (of harvest with a model) to a factor such as the discount factor, which in its estimation/assumption involves a great deal of uncertainty but also arbitrariness depending on the philosophical/political/environmental... views.

The other major suggestion I would have is that in my view the article could benefit from improving a bit the structure, coherence and clarity of the assumptions, the computations performed (e.g. making use in the text of summary tables of variables, parameters, values assumed, etc. and complementary results on the sensitivity, robustness...in the Supplementary Material or even with additional resources), and by distinguishing the terms of what is done here versus what it is usually found in the literature, not only in the mainstream discussion of the discount factor.

All in all, in my opinion the article is good and may be published, but some changes need to be introduced in line with the questions raised and arguments presented, to meet the high standards of Nature Communications. The specific comments that I think could help the article to be better understood for the reader are shown below in detail.

Specific comments

Abstract) “current estimates of the discount factor, which are based on extrapolations from its present changes, are biased and are invalid in the long-run”. I follow from the general comment above. If there was a clear cut and widely acceptable way of estimating the discount factor, I would understand that the contribution improves the estimates, adding a forgotten component in relation to harvest, etc. However its estimates, or I should rather said guesses/assumed values, etc. involves a great deal of arbitrariness depending on the philosophical/political/environmental... views of the modellers/policy makers, etc. How important is to account for this relationship between harvest and the discount, compared to many other aspects which seem playing also a role (and may be the lion’ share of discounting), such as the shape assumed over time (whether discounting at all, or having a hyperbolic shape, or a gamma discount –Weitzman, 2001, etc.), the role of technological growth (which can also be related to discounting, and as several other variables involve unaccounted feedbacks), the assumptions regarding utility, its shape, and why base it purely on consumption, etc.? I miss some reflecting regarding the contextualization of this improvement with respect to the wider picture of discounting.

p. 2, lines 43-45) “For example, consider no price changes and a constant annual discount rate, $\equiv \Delta/$. Then, the value of a good that worth \$1M today would be worth ~50K\$ for the same good in 100 years if $= 3\%$, and only ~2.5K\$ if $= 6\%$ ”. I think that these type of examples are very useful. It would be good to have some similar sentences to highlight the differences of the question of the article, i.e. how biased or invalid in the long-run are the discount factors when there is over-harvested natural resources (extending in p. 8, pp. 271 - 273).

p. 4, pp. 121-122) “investment in the harvest results in a decline in production and consumption of the second good (...)” I understand the logic of switching labor and/or resources are directed toward harvest. Notwithstanding, in a typical production process, such as that reflected in the national accounts, typically most goods act as inputs of other goods, so typically the good “harvested” would also enter the production of the second good or combination of goods (e.g., manufactured goods) further in the supply chain, and hence not being that clear this relationship.

p. 4. 139-140) “which implies that, in a perfectly competitive market, social welfare is a function of the total consumption, $c(t)$ and $f(t)$.” It must seem obvious, but “ c ” and “ f ” are not defined (as the consumption of 2 goods, which are exemplified later). More importantly, the results keep on being mostly referenced to “perfectly competitive market” which I think it can never exist in reality (and less when it is assumed that there are shared areas) and always it is derived providing to it some

heroic and unrealistic assumptions. Related to this, these type of definitions of “utility” might be considered highly narrow-minded, also lacking considerations on other values of natural resources (non-consumptive, option values, aesthetic, etc.).

p. 5-6) Results. “Our results show that, following optimal harvest, two phases emerge along the time axis (Fig. 2A,B)...”

I miss some more guidance on the results. Figures 2 and 3 are not easy to interpret, with some abrupt changes and recoveries (and also changes in the discount rate with zig zag and sharp changes) which I think should be further explained, especially since some values fall out of the figures. Also I see the parametrization in the figures’ notes, but then I have no intuition about how different the results would be with different parametrization. Perhaps comprehensive results of the numerical solutions, providing clear sensitivity analyses, discussion of robustness, etc. could be provided in the Supplementary material. Indeed, ideally if it was possible I think it would be great to have/link to some website/applet/GUI...where one can modify/play with the parameters and see their effect.

Also I guess a comment that might be more for the editors to consider is to what extent it is common practice or interesting to have the results in Nature Communications providing basically hypothetical numerical solutions and a theorem, and not on the explanations of key insights for a wider audience.

p. 7, 230-232) “Our study shows that the transition to sustainable harvest after a period of over-harvesting leads to a decline in welfare, economic growth, and in the discount rate (Fig. 2)”.

I am not sure that the relation of over-harvesting to economic growth is shown in this work (although it might have been shown in other articles), if so, what is the mathematical expression and where is it discussed?

p. 7, 243-246) “other studies suggest that a slow in technological development due to environmental degradation may affect the discount (23, 24); and some studies suggest that a decline in harvest would ultimately result in less production, which, in turn, affects economic growth and discount (8)”. It seems that (8) is one work to which compare the results, since it refers to the relation of harvest and discount (although apparently with a quite different view). Is it also the only previous study which considers this relationship?

p.8-9) Methods. I would provide this together with the “Model” section, so that the reader knows that one can find referenced the Appendices.

Reviewer #2 (Remarks to the Author):

Referee report for the article "Over-exploitation of natural resources is followed by an inevitable decline in economic growth and discount rate"

This manuscript sets out to show that the variables "growth" and "discount rate" vary with exploitation of the natural resources in the economy. Basically the story is that first natural resources are over-exploited unsustainably. This gives rise to a high growth rate and high discount rate. After this the depletion of ecosystem resources forces natural resource exploitation to be reduced which leads to a fall in GDP and in discount rate. The "true" – or in this case "long-run" discount rate is thus lower than what it seems in the boom period of unsustainable resource exploitation. The nice point about this is that the discount rate itself is used to motivate a high rate of exploitation. Thus in the initial period of high resource use this "over-exploitation" receives some support in a simple cost benefit analysis carried out with a discount rate that is "too high" – but actually it is the discount rate of that period. Thus as long as we are in this period we appear to be right to exploit the resource but as soon as we snap out of it and fall to a lower rate of growth, we see that we were wrong to have over-exploited the resource since such exploitation is no longer profitable with the new lower discount rate. This would seem to be a puzzle but the author focuses mainly on the longrun rate as the "right" rate.

This particular line of thinking is in my opinion important and – although not completely unknown, still novel enough to be potentially publishable with a good theoretical underpinning, nice examples or case studies and good policy discussion.

I am thus positively disposed, probably more so than most economists. Still I think the current manuscript has a number of significant problems that exclude publication. If one wants to be generous, one could say that it is possible to fix but it would entail a very considerable rewrite. I am thus torn between rejection and major revision.

I have given my comments below in relation to a number of different lines. They may look like details but they are all symptoms of the same main fundamental feature. The article is written in a style that is so far from the traditional way of analysing macroeconomic growth models or optimal resource management that this becomes an obstacle for communication. The potential readers of an article about discounting and growth are of course mainly economists - likely resource economists, environmental economists and similar. I fear that few of them would get through the article and most of them would think it was flawed.

It is possible that the model is in some sense "correct" in its own terms. In fact this is even probable since the main result looks quite similar to what is described elsewhere in the cited literature. The

trouble is that the abstractions and assumptions and other model choices made are far from intuitive and in many cases look wrong. Two such features I find strange are that there are just two ways of managing a natural resource 1. Sustainably 2 Non-Sustainably and that these give fixed payoffs with a simple parameter difference. This is far from the dynamic stock flow models. Secondly that open access areas can be managed “sustainably”. This goes against every thing we know from both Ostrom and many others who work on resource management regimes.

The author also has a very naïve relationship to the discount rate, as if it were something the decision makers of an economy easily could measure and agree on. This is far from the case which makes his main claims somewhat less dramatic, but as mentioned I still find them interesting.

My main recommendation would be thus rejection but I could also consider a very serious major revision where the author would ditch most of the current model and show his main result in a more conventional model.

Detailed comments

Line 1-2. A major challenge in environmental policymaking is determining whether, where and how fast we should adopt sustainable management.

This is quite a strange sentence. Who is “we” – a wise and environmentally conscious manager or an average businessman in say the oil sector? Or the government? If it is an environmentalist or in fact any reasonable social planner it would be strange to answer the question above by deciding to do unsustainable management. However, if it is an entrepreneur in the oil sector they may not even think of this as a choice let alone a challenging one.

Lines 8-14 this key sentence in the abstract shows both the niceness and the naivety of this paper.

It is a nice observation but it seems to suggest a belief that the policy maker believes in the Ramsey rule (or something similar) and actually believes in frequent updates of this or some similar rule to set the social discount rate and so forth. In reality little of this necessarily applies.

Lines 14- 16 Therefore, when discounting the far-future, we should use an adjusted discount formula that I derive here and that implies significantly higher net costs due to future environmental damages. The author needs to decide if he is using discount rates normatively or positively. (See also below)

Lines 31-32 relationship between CBA and optimal management are more complicated...

Line 44-45 Then, the value of a good that worth \$1M today would be worth ~50K\$ for the same good in 100 years if $r = 3\%$, and only ~2.5K\$ if $r = 6\%$.

The author managed to phrase this the wrong way round. It should be “the value of a good worth \$1M in a hundred years would be worth ~50K\$” today.

55-56 Footnote 11 refers to a working paper at RFF. I believe it was published in REEP

Arrow, K., M. L. Cropper, C. Gollier, B. Groom, G. M. Heal, R. G. Newell, W. D. Nordhaus, R. S. Pindyck, W. A. Pizer, P. Portney, T. Sterner, R. Tol and M. L. Weitzman (2014) "Should Governments Use a Declining Discount Rate in Project Analysis?" *Review of Environmental Economics and Policy*, volume 8, issue 2, pp. 145–163 doi:10.1093/reep/reu008. But you should check because there are several similar papers.

82- Model selection. It is somewhat strange ecology and strange economy with these four separate regions (they are typically not so separate) and these three choices of harvest that is "sustainable" unsustainable or 0.

L 91-96. Somewhat strange, non-standard way of presenting the natural resource model. There are just two states yielding $\alpha\beta$ or β units but leading in the former case to a share of the ecosystem κ becoming unproductive..etc.

It is more common to work with continuous and more general functions.

On line 97 α is used for the growth rate of fish stocks – so that I think is confusing – maybe the author intended to use a separate variable here. Otherwise it looks like α is given two different meanings.

105 formula (2) I find it very strange that the harvest just depends on the areas that are harvested sustainably or unsustainably. There are many levels and ways of being unsustainable. Harvest depends on how many people harvest unsustainably at a certain time. There are important stock-flow interactions. The reason people harvest unsustainably is notably because they get gigantic harvests – not for a trivial time but for decades --- then the stock may crash. I don't find the basic production structure specified to be believable or informative and this makes me more than worried for the "derivations".

108-110 sees the introduction of private / open access land. But again this happens in a strange way. It seems the agents can choose to harvest either sustainably or unsustainably on private land. That is fine but it seems they can also do this in open access areas. In the whole literature on Open Access, this is normally excluded. Even if one agent would try to behave sustainably in an open access area – the whole meaning of "open access" is that an undefined number of other agents are taking decisions on the same resource and therefore sustainable use is excluded and not possible.

See for instance the following sentence: Consequently, each agent has four possible actions as he/she can decide how much to harvest sustainably and non-sustainably both on his/her private land and on the shared land. The last part is not generally thought to be possible in open-access.

118-125. some kind of cost structure is discussed but I find the sentences very confusing. Investment and costs of production appear mixed. The cost of production in the resource sector seems to depend on what happens in the sector for production of other goods but I do not see how.

126 to 141 and formula 6 describe welfare which is a function of consumption and of natural resources. I fail to see why natural resources enter into welfare. If it had been scenic beauty then I could understand it but from earlier discussion it is clear that f refers to things like fish and timber

and in that case they would surely already be included in c? Maybe people consume fish and computers and possibly this approach could be defended but I do not think it has been explained or motivated.

The author is very clearly not an economist. He uses quite a few words, investment, inflation, general equilibrium, perfect market economy, open access but in somewhat strange ways that sometimes at least are definitely wrong, and his model is very unorthodox. Of course in principle we might learn something from new approaches and I do not want to come across as thinking that only economists can write about economics but when no effort is made to connect to standard models then ultimately there is little learning or communication. I find the article bewildering and I fear that all other economists will feel the same way. The article is about welfare, production, resources and in particular discounting and growth – so if the article is not intended for economists then one wonders who the readers will be.

231-232 “Our study shows that the transition to sustainable harvest after a period of over-harvesting leads to a decline in welfare, economic growth, and in the discount rate (Fig. 2).

While I find many details of the model confusing I recognize the conclusion at least for the welfare. You can imagine a period of overharvesting of natural resources that leads to an eventual decline in welfare. This is correct although not very surprising. The question of what happens to growth and discount rate is complicated by questions of definition. It may be true that growth as generally measured by conventional measures such as GDP decrease. However if income were properly measured then it would not have been so high in the period of “unsustainable growth”. All the damage being done to stocks of ecosystem resources should have been properly valued. In this case income and growth would not be so high in the first period and thus the “decline” in economic growth may not happen. I suspect it would be the same for the discount rate although again it depends somewhat on how this is defined. It is not common to discuss it as if it were an independent parameter of the system. Many economists describe it in very different ways.

I think this touches on a central point for the author who writes:

Consequently, we should consider the expected future discount rates to be lower than today's discount rate, which implies that environmental damages will incur higher net costs. Ignoring the harvest-induced decline in cumulative discount not only falsify cost-benefit analyses, it also creates a bias: Over-harvesting increases the discount rate in the short-run, which unjustifiably biases our expectations to anticipate higher future discount rates, which, in turn, is used to justify further exploitation.

I sort of like this sentence and the sentiment it conveys. It is cute and rings true but one has to know that the discussion of discount rates is much complicated by question of “whose” discount rates we are discussing. The author's discussion is so unusual that it is a little hard to judge but I would think he is discussing the optimal discount rate in a normative sense for an omnipotent social planner. The trouble is that social planners hardly exist, at least they are rarely omnipotent and they don't really make the decisions about forestry and fishing... When there are dictators who do make such decisions it seems they often maximize other functions.

The authors who discuss discount rates generally fall into one of two groups. Those who think it is a normative ethical parameter that cannot be observed and that the social planner should choose to guide her decisions. In this case it follows something like the Ramsey Rule. The other group thinks it is endogenous to the economy and can be observed. This group however does not believe in any Ramsey rule or in social planning, welfare maximisation in this sense. They typically think the discount rate should match the observed rate of interest on secure bonds.

The author of this paper seems generally closer to the first group but then speaks of the discount rates movements as something highly observable in spite of the fact that it is the sum of a number of second derivatives of very special production and consumption functions he has chosen. I think the paper in its current form would be much criticized or maybe ignored. I think however it is possible (in fact likely) that the main insight could survive a major transformation. I think a conventional economic model could be formulated with an ecosystem with stock and flow properties. The effects on growth of harvest decisions could be discussed and could be quite similar. I think one could also discuss the “cute effect” on the optimal normative social discount rate as it is usually defined.

I believe the intuition about the role of discount rates for motivating investments that are relevant for our future is partly true. But the author must signal more clearly that he understands this only refers to an ideological battle in arenas like the Stern Review, the IPCC and similar. Actual resource decisions are generally made with little recourse to discounting – or based on horrendously high rates motivated by the cost of actually procuring capital or the profitability “demands” of companies.

I have the feeling most of my economics colleagues would put a simple reject on this paper. I am sympathetic to its main direction. However, if it is going to convince anybody then it has to be written in a more accessible style. It is after all mainly directed to economists and must therefore either be written as economists analyse natural resource and growth issues or at least refer to such models and choose and motivate any necessary departures. As it currently stands much of the material will be an obstacle to communication. This applies for instance to the whole discussion of open access and private resources used in the early part of model construction and in figure 1. The whole idea of only having two levels of production one for sustainable management and one for unsustainable management is hard to accept. As I mentioned there is a lot of dynamics there. The role of the two goods must be clarified better, how they interact and contribute to welfare. How production and costs are decided. I am sorry to sound like a boring conventional economist but if you write an article about growth and discounting then that is large part of your audience – and if you build a model that is completely different then the burden of proof is on you. I think your vital points could be made more effectively in quite a conventional model.

Figures 2-3 give some very precise impression but clearly they are some very special case of your model and you hardly comment on them in the text. I am not sure what their function is.

Reviewer #3 (Remarks to the Author):

Please see the attached review.

Reviewer 1

I thank the Reviewer for carefully reading the Manuscript and for the helpful comments. Below please find each of the Reviewer's comments (black) followed by my response (blue).

General comments

The article highlights that although valuing natural systems and has been a major focus of debate in recent decades, it is still unclear how the environment and its management affect the discount factor. The article presents a model to show how large changes in the harvest of natural resources affect the discount factor via non-linear effects on social welfare. Specifically, it is shown that over-harvesting temporarily keeps the discount rate higher, but is followed by a period of lower discount rates when society adjusts to sustainable harvest. It is concluded that hence when discounting the far-future, it should be used the derived adjusted discount formula implying significantly higher net costs due to future environmental damages.

In my view, the article is well-written and introduces a relevant improvement when endogenizing harvest of natural resources to affect the discount factor. In that sense, if considered that discounting should be applied (there is literature referenced on the optimal/acceptable... levels of discount factor, but there are also advocates of the zero discount factor, based on inter-generational equity, for whom these type of discussion do not even make sense) probably it should be examined and discussed why this improvement or change in assumptions (i.e. endogenizing harvest) is the only or the most necessary one, considering that one could have also related to many other aspects or variables (values and shape of the discounting, the role of technology, of growth, of how goods interrelate, etc.). Related to this, I wonder about the importance of finding a mathematic relationship (of harvest with a model) to a factor such as the discount factor, which in its estimation/assumption involves a great deal of uncertainty but also arbitrariness depending on the philosophical/political/environmental... views.

I thank the Reviewer for stating that the study is important, relevant and well-written. The comment of the Reviewer that follows includes two parts. In the first part, the Reviewer mentions that “there are also advocates of the zero discount factor, based on inter-generational equity.” Note that, many authors (e.g., Stern in his report (6)) are advocates of zero pure time preference ($\rho = 0$ in my notations). However, the zero pure time preference dictates that the discount rate should be lower, but still not zero. In my model, I refer to the pure time preference, ρ , as a constant parameter, and my main results do not depend on the choice of ρ . To make this point clear in the Manuscript, I have provided a better explanation of the meaning of the symbols in the Model (see Table 1, which is a new table that defines the symbols). Also, in the Discussion, I explain that “the effect of harvest on discounting is additive and should be considered in addition to (not instead of) changes dictated by other considerations. For example, the value of the pure time preference, ρ (Eqs. 2,3), is controversial; some authors argue that its market-based value of $\sim 3\%$ should be used, but

others argue that $\rho \approx 0$ should be used based on considerations of intergenerational equity (6, 16, 19, 42). The value of ϕ_τ , however, does not depend on ρ and should be subtracted from Δ regardless of the value of ρ ." (lines 300-305).

In the second part of the comment, the Reviewer notes that many factors could affect the discount rate. In light of that, the Reviewer wonders why studying the effect of harvest if there are already so many uncertainties about the value of the discount rate. The answer is that it is important to study how harvest patterns affect the discount because it imposes significant changes on the calculation of the discount rate, on top of those imposed by other mechanisms. In turn, this implies a significant adjustment to the future value of natural resources.

In the revised Manuscript, I have added analysis to examine the magnitude of the change in the discount due to changes in harvest (Figs. 4 and 5 and lines 306-323). This analysis shows that the adjustment to the value of future goods due to endogenizing harvest is significant. Specifically, my results show that the value of future goods should be higher by a factor of $\exp(\phi_\tau)$ compared to the results in studies that suggested that we should adjust the discount rate gradually (e.g., within τ years) from its current value, δ_{today} , to its sustainable rate, δ_{sus} (lines 288-302). The magnitude of this factor is shown in Fig. 5 for various parameter values. It shows that $\exp(\phi_\tau) \geq 2$ (and could be significantly higher) for various reasonable parameter ranges. This factor is significant and implies the need for considerable adjustment to current environmental policies.

Note that my study does not intend to resolve the existing controversies around discounting; rather, the intent is to suggest a correction that would apply on top of the existing paradigms and that should be accepted on all sides. Namely, the adjustment that I suggest to the discount factor is additive and should be considered in addition to (not instead of) other adjustments suggested in the literature due to other mechanisms, as I explain in lines 300-307. I elaborate more on this issue below when I respond to the specific points by the Reviewer.

The other major suggestion I would have is that in my view the article could benefit from improving a bit the structure, coherence and clarity of the assumptions, the computations performed (e.g. making use in the text of summary tables of variables, parameters, values assumed, etc. and complementary results on the sensitivity, robustness...in the Supplementary Material or even with additional resources), and by distinguishing the terms of what is done here versus what it is usually found in the literature, not only in the mainstream discussion of the discount factor.

The Reviewer suggests to improve the clarity and structure in the following three aspects.

Clarity of the model

I have made numerous changes to improve the clarity of the Model section. First, I rearranged the section and I begin with the description of the social welfare and the discount rate (as the

Reviewer suggested in the specific comments below). This makes a good connection with the Introduction and justifies why we would like to endogenize harvest (which is to determine the terms dc/dt and df/dt in the discount formula). Next, I present the dynamics of $f(t)$ and $c(t)$ and how they depend on the aggregate harvest functions. Subsequently, I specify how the managers determine the harvest functions (e.g., optimal solution versus market solution). I also replaced Fig. 1 with a new figure that clearly presents the model. In addition, I added a paragraph describing how I define the discount rate (lines 87-102), and I also added a paragraph that better explains the considerations associated with the dynamics of f (lines 135-153).

Complementary results on sensitivity and robustness

What proves the robustness of the main results is, above all, the theorem. This theorem implies that the cumulative discount, Δ , eventually must decline and become lower than Δ_{sus} (where the discount factor is given by $\exp(-\Delta)$). To further show the robustness of the results, I extended the theorem to make it independent of some specific assumptions (see details in a respond to another comment of the Reviewer below). Accordingly, I have added two new figures that examine the magnitude of the decline in Δ predicted by the model. This decline depends on several parameters, and Figs. 4 and 5 show how it varies with these parameters.

What is done here versus what is known from the literature

In the revised Manuscript, I begin the Model section by describing how my model differs from previous models: “We begin with describing a well-established framework (25, 35, 36) that specifies how social welfare and the discount rate depend on the provision of the natural resource over time, $f(t)$, and on the consumption of other goods over time, $c(t)$. Next, we specify how harvest at the global scale determines the dynamics of $f(t)$ and $c(t)$. This builds on and generalizes previous studies that considered the dynamics of $f(t)$ and $c(t)$ as exogenous (e.g., growing exponentially) (25, 35, 36).” (Lines 81-85).

In addition, I elaborate more on what was done and I clarify that “several mechanisms underlie the preference for present consumption (15, 16). One mechanism, which we focus on in this study, is that the marginal contribution to welfare of a given unit of consumed goods will be lower if our society consumes more goods in the future. Other mechanisms are that we may prioritize our happiness today over our happiness tomorrow (pure time preference) and that we need to be averse to uncertainties about future consumption.” (Lines 97-102.)

Next, in the Discussion section, I explain that previous studies have derived expressions similar to δ_{sus} , but they assumed that the transition to δ_{sus} will be gradual. In my study, I show that, in addition to the decline from δ_{today} to δ_{sus} , there will be a negative “shock” that imposes an additional decline in the cumulative discount (Eq. 8). Thus, I added Fig. 5, which shows the magnitude of the change in the discount factor due to the shock, $\exp(\phi_\tau)$. Specifically, $\exp(\phi_\tau)$ is the adjustment to the discount factor that is needed in order to correct a policy that

would use δ_{today} for the next τ years and switch to δ_{sus} afterward. These issues are discussed further in lines 287-324.

All in all, in my opinion the article is good and may be published, but some changes need to be introduced in line with the questions raised and arguments presented, to meet the high standards of Nature Communications. The specific comments that I think could help the article to be better understood for the reader are shown below in detail.

Specific comments

Abstract) “current estimates of the discount factor, which are based on extrapolations from its present changes, are biased and are invalid in the long-run”. I follow from the general comment above. If there was a clear cut and widely acceptable way of estimating the discount factor, I would understand that the contribution improves the estimates, adding a forgotten component in relation to harvest, etc. However its estimates, or I should rather said guesses/assumed values, etc. involves a great deal of arbitrariness depending on the philosophical/political/environmental views of the modellers/policy makers, etc. How important is to account for this relationship between harvest and the discount, compared to many other aspects which seem playing also a role (and may be the lion’ share of discounting), such as the shape assumed over time (whether discounting at all, or having a hyperbolic shape, or a gamma discount –Weitzman, 2001, etc.), the role of technological growth (which can also be related to discounting, and as several other variables involve unaccounted feedbacks), the assumptions regarding utility, its shape, and why base it purely on consumption, etc.? I miss some reflecting regarding the contextualization of this improvement with respect to the wider picture of discounting.

The Reviewer follows from the main comment and (1) asks that I clarify that there is no agreement on how the discount should be measured and (2) inquires whether the adjustment to the discount that I suggest is significant compared to other mechanisms that have been suggested (“How important is to account for this relationship between harvest and the discount, compared to many other aspects which seem playing also a role.”)

To address point (1), I have added a new paragraph (lines 87-102) in which I explain that “there is no consensus on the best measure of the social rate of discount ...” (lines 88-89), and I note that, in this study, “In this study, we focus on the social rate of time preference, which we refer to as the discount rate” (lines 93-94). Also, note that I discuss the various mechanisms that could affect the discount rate in the Introduction (lines 47-64) and discussion (lines 266-271 and lines 300-308). Specifically, going back to the Reviewer’s comment, I discuss the fact that the “philosophical/political views” of the policymakers may affect their choice of the pure time preference, ρ (lines 302-305). I also discuss the idea that uncertainties regarding technological growth may affect the “shape assumed over time” (lines 54-56, 267-268, and 305-307.) I also examine how the shape of the utility function affects the adjustment to the discount by comparing the two cases of non-substitutable goods and partially substitutable goods (Figs. 2,

3, and 5). Note that I deleted from the Abstract the specific sentence mentioned by the Reviewer from the abstract.

To address point (2), I added the analysis that is shown in Figs. 4 and 5 and discussed in lines 309-326. These figures show how the correction to the discount factor that the study suggests depends on various parameters. I show that the results suggest a significant modification to the discount, irrespective of modifications suggested due to other mechanisms. See my reply to the main comment that includes more details.

p. 2, lines 43-45) “For example, consider no price changes and a constant annual discount rate, $\equiv \Delta/$. Then, the value of a good that worth \$1M today would be worth ~50K\$ for the same good in 100 years if $= 3\%$, and only ~2.5K\$ if $= 6\%$ ”. I think that these type of examples are very useful. It would be good to have some similar sentences to highlight the differences of the question of the article, i.e. how biased or invalid in the long-run are the discount factors when there is over-harvested natural resources (extending in p. 8, pp. 271 - 273).

The Reviewer restates her/his will to see some more quantitative results. As I mentioned in the response to the major comments, there are added in Figs. 4 and 5 and lines 308-324.

p. 4, pp. 121-122) “investment in the harvest results in a decline in production and consumption of the second good (...)” I understand the logic of switching labor and/or resources are directed toward harvest. Notwithstanding, in a typical production process, such as that reflected in the national accounts, typically most goods act as inputs of other goods, so typically the good “harvested” would also enter the production of the second good or combination of goods (e.g., manufactured goods) further in the supply chain, and hence not being that clear this relationship.

The Reviewer asks why the harvest results in a cost that reduces the consumption of the manufactured goods. In this study, I focus on the harvest of natural goods that are used for consumption by households (e.g., fish harvest, crop yield). I do not incorporate in the model the idea that these goods are potentially being used for the production of manufactured goods. In other words, I follow the frameworks discussed in Refs. 25, 35 and 36 (unlike the framework mentioned by the Reviewer, which is discussed in reference 12). Including these relationships (e.g., f affects g_c) could be an interesting topic for future studies. In the revised Manuscript, I clarify this issue, e.g., “I assume that technological development is exogenous and consider an exponential increase in the efficiency of producing manufactured goods and in the yield of natural resources per unit of non-degraded area. (In particular, the production of manufactured goods does not depend on natural resources.)” (Lines 73-76.)

p. 4. 139-140) “which implies that, in a perfectly competitive market, social welfare is a function of the total consumption, $c(t)$ and $f(t)$.” It must seem obvious, but “ c ” and “ f ” are not defined (as the consumption of 2 goods, which are exemplified later). More importantly, the results keep on being mostly referenced to “perfectly competitive market” which I think it can never exist in

reality (and less when it is assumed that there are shared areas) and always it is derived providing to it some heroic and unrealistic assumptions. Related to this, these type of definitions of “utility” might be considered highly narrow-minded, also lacking considerations on other values of natural resources (non-consumptive, option values, aesthetic, etc.).

Here, the Reviewer comments that: (1) I use c and f in the text to refer to the goods, but they are actually defined as the provision/consumption of the goods; and (2) I assume a “perfectly competitive market,” whereas real markets are not such.

To address point (1), I went through the Model and Results sections as well as the Appendixes and fixed that semantical issue.

Regarding point (2), in the revised Manuscript, I extend the theorem and I show that the main result holds without the assumption of a perfectly competitive market. Specifically, I explain that “For the theorem, we consider a general case in which the market solution implies a higher rate of non-sustainable harvest. For our simulations, however, we specify here a special case in which we distinguish between the externalities in the managed and the shared regions.” (Lines, 189-191). Then, in the next paragraph, I explain the assumptions that I used for the simulation, which include a variety of specific assumptions, including the assumption that the market is perfectly competitive. In turn, I explain that the theorem applies more generally and does not necessitate specific assumptions about how H_n is determined (one only needs to assume that H_n may be positive, regardless of what leads to that). In turn, I extended the theorem to include the more general case: “... for any Δ that emerges if non-sustainable harvest ($H_n(t) > 0$) occurs between times t_0 and t_1 , there exists $t_c > t_1$ such that $\Delta(t_c) \leq \Delta_{sus}(t_c)$.” (Lines 252-253.) (The extension to the proof is given in Appendix D, Lemma 5 and proof of Theorem 1.)

p. 5-6) Results. “Our results show that, following optimal harvest, two phases emerge along the time axis (Fig. 2A,B)...” I miss some more guidance on the results. Figures 2 and 3 are not easy to interpret, with some abrupt changes and recoveries (and also changes in the discount rate with zig zag and sharp changes) which I think should be further explained, especially since some values fall out of the figures. Also I see the parametrization in the figures’ notes, but then I have no intuition about how different the results would be with different parametrization. Perhaps comprehensive results of the numerical solutions, providing clear sensitivity analyses, discussion of robustness, etc. could be provided in the Supplementary material. Indeed, ideally if it was possible I think it would be great to have/link to some website/applet/GUI...where one can modify/play with the parameters and see their effect.

Also I guess a comment that might be more for the editors to consider is to what extent it is common practice or interesting to have the results in Nature Communications providing basically hypothetical numerical solutions and a theorem, and not on the explanations of key insights for a wider audience.

The Reviewer raises here three different points: (1) There is a need for some more guidance for interpreting Figs. 2 and 3, (2) there is a need for showing the robustness of the results (e.g., sensitivity analysis), and (3) the results are basically hypothetical numerical solutions and a theorem, not explanations of key insights for a wider audience.

To address point (1), I have added a label, “ t_1 ”, at the time the resources in the open-access regions are being exhausted and the discount declines sharply. There is another label, t_0 , which represents the time after which the entire system is being harvested. Throughout the Results section and in the captions of Figs. 2 and 3, I refer to these two times to clarify to which parts of the figure I refer. (Note that the label t_1 in Fig. 2 and in Fig. 3 refer to the same time.)

Regarding point (2), the idea is that Fig. 2, indeed, demonstrates a special case. The robustness of the results is implied by the theorem, which proves that the main result must hold under some general assumptions. Accordingly, I believe that showing more panels similar to Fig. 2 would be less valuable. Instead, I added Figs. 4 and 5, which show the lower bound of the magnitude of the change in the discount factor due to over-harvesting, as implied by the theorem, for various parameter values. I focused on analyzing the effect of the most significant parameters, such as the growth in the production of the natural resource per unit area (g_f) and the substitutability of the natural resource and the other goods (determined by u). In Fig. 5, I also examine the role of τ in order to compare the adjustment I recommend to those recommended by studies in which a gradual transition to δ_{sus} over $\sim\tau$ years was suggested (see more on that below).

Regarding point (3), I believe that Figs. 4 and 5 also address this point as they present key insights for a wider audience. Namely, one could understand the policy implications of the results presented in these figures without getting into specific details.

p. 7, 230-232) “Our study shows that the transition to sustainable harvest after a period of over-harvesting leads to a decline in welfare, economic growth, and in the discount rate (Fig. 2)”. I am not sure that the relation of over-harvesting to economic growth is shown in this work (although it might have been shown in other articles), if so, what is the mathematical expression and where is it discussed?

The Reviewer comments that the relationship between harvest and economic growth was not shown explicitly in the original Manuscript. Accordingly, in the revised Manuscript, I have added graphs showing the economic growth (and its decline during the transition to sustainable harvest) to panels C and D in Fig. 2. Specifically, if μ represents the portion of the consumption of the natural resource in the basket of goods, then, by definition, the total production at time t (which is proportional to GDP if we are focusing on a specific country) is proportional to $c(t)P_c(t) + f(t)P_f(t)$, where P_c and P_f are the prices of the manufactured goods and the natural resource, respectively (see line 124-126 and Eq. A11 in Appendix A in the revised Manuscript).

p. 7, 243-246) “other studies suggest that a slow in technological development due to environmental degradation may affect the discount (23, 24); and some studies suggest that a decline in harvest would ultimately result in less production, which, in turn, affects economic growth and discount (8)”. It seems that (8) is one work to which compare the results, since it refers to the relation of harvest and discount (although apparently with a quite different view). Is it also the only previous study which considers this relationship?

The Reviewer suggests that I include a better comparison of my results with those found in the literature. In the revised Manuscript, I explain that the framework that specifies how social welfare and discounting depend on $f(t)$ and $c(t)$ is well-established. Previous studies, however, considered the dynamics of $f(t)$ and $c(t)$ as exogenous (e.g., growing exponentially), whereas I endogenize these dynamics to examine how the dynamics of harvest affect social welfare and discounting (e.g., lines 81-85). In turn, in the Discussion, I note that “Expressions for other forms of the utility function are given in Appendix B and in the literature (25, 35).” (Lines 314-315.) I emphasize that what is novel in my study is the calculation of the expected decline in the cumulative discount, ϕ_τ . This is demonstrated in Figs. 4 and 5 and discussed in lines 280-326. To my knowledge, this idea is novel, and the term ϕ_τ has not been considered in previous studies.

p.8-9) Methods. I would provide this together with the “Model” section, so that the reader knows that one can find referenced the Appendices.

In the Model section in the revised Manuscript, I describe the methods that are needed to understand the model (lines 183-191). Also, I give references to the appendices when needed. However, I kept the Methods section at the end because it provides a summary for readers who will not read the entire Model section.

Reviewer 2

I thank the Reviewer for carefully reading the Manuscript and for the helpful comments. I also thank the Reviewer for her/his positive attitude and her/his ability to read and understand the main ideas despite the various difficulties that he/she finds. Below please find each of the Reviewer's comments (black) followed by my response (blue).

This manuscript sets out to show that the variables “growth” and “discount rate” vary with exploitation of the natural resources in the economy. Basically the story is that first natural resources are over-exploited unsustainably. This gives rise to a high growth rate and high discount rate. After this the depletion of ecosystem resources forces natural resource exploitation to be reduced which leads to a fall in GDP and in discount rate. The “true” – or in this case “long-run” discount rate is thus lower than what it seems in the boom period of unsustainable resource exploitation. The nice point about this is that the discount rate itself is used to motivate a high rate of exploitation. Thus in the initial period of high resource use this “over-exploitation” receives some support in a simple cost benefit analysis carried out with a discount rate that is “too high” – but actually it is the discount rate of that period. Thus as long as we are in this period we appear to be right to exploit the resource but as soon as we snap out of it and fall to a lower rate of growth, we see that we were wrong to have over-exploited the resource since such exploitation is no longer profitable with the new lower discount rate. This would seem to be a puzzle but the author focuses mainly on the longrun rate as the “right” rate.

This particular line of thinking is in my opinion important and – although not completely unknown, still novel enough to be potentially publishable with a good theoretical underpinning, nice examples or case studies and good policy discussion. I am thus positively disposed, probably more so than most economists. Still I think the current manuscript has a number of significant problems that exclude publication. If one wants to be generous, one could say that it is possible to fix but it would entail a very considerable rewrite. I am thus torn between rejection and major revision. I have given my comments below in relation to a number of different lines. They may look like details but they are all symptoms of the same main fundamental feature. The article is written in a style that is so far from the traditional way of analysing macroeconomic growth models or optimal resource management that this becomes an obstacle for communication. The potential readers of an article about discounting and growth are of course mainly economists - likely resource economists, environmental economists and similar. I fear that few of them would get through the article and most of them would think it was flawed.

It is possible that the model is in some sense “correct” in its own terms. In fact this is even probable since the main result looks quite similar to what is described elsewhere in the cited literature. The trouble is that the abstractions and assumptions and other model choices made are far from intuitive and in many cases look wrong. Two such features I find strange are that there are just two ways of managing a natural resource 1. Sustainably 2 Non-Sustainably and that these give fixed payoffs with a simple parameter difference. This is far from the dynamic stock

flow models. Secondly that open access areas can be managed “sustainably”. This goes against every thing we know from both Ostrom and many others who work on resource management regimes. The author also has a very naïve relationship to the discount rate, as if it were something the decision makers of an economy easily could measure and agree on. This is far from the case which makes his main claims somewhat less dramatic, but as mentioned I still find them interesting. My main recommendation would be thus rejection but I could also consider a very serious major revision where the author would ditch most of the current model and show his main result in a more conventional model.

The Reviewer finds the Manuscript interesting, but he/she raises a major concern that the model is presented in a non-traditional manner, which may be an “obstacle for communication”. Accordingly, the Reviewer suggests major revisions to better justify and explain the various assumptions used in the model. To address this issue, I have completed a major rewrite of several parts of the Manuscript, particularly in the Model and Discussion sections. Below, I first describe the major changes that I have made in the Model section, and, then, I explain how the more specific issues raised by the Reviewer were addressed.

How I revised the Manuscript to address the concerns of the Reviewer.

In the revised Manuscript, I divide the Model section into three parts. In the first part, I describe how social welfare and the discount rate depend on the provision of the natural resource over time, $f(t)$, and on the consumption of the other goods over time, $c(t)$. In the second part, I describe a model that specifies how harvest at the global scale determines the dynamics of $f(t)$ and $c(t)$. In the third part, I describe how the market determines harvest strategies. I also emphasize that the framework that specifies how social welfare and discounting depend on $f(t)$ and $c(t)$ is well-established (Refs. 25, 35, 36). However, previous studies considered the dynamics of $f(t)$ and $c(t)$ as exogenous (e.g., simply growing exponentially), whereas, in the present study, I endogenize these dynamics to examine how harvest affects social welfare and discounting. (see Lines 80-84.) Namely, what I describe in the first part of the Model section is not the novel part of the study. Rather, the novel part is that I consider how c and f depend on the harvest (endogenizing the harvest), in contrast to previous studies (e.g., Refs. 25, 35, 36) that simply assumed that $c(t) = c_0 \exp(g_c t)$ and $f(t) = f_0 \exp(g_f t)$.

To clarify the first part further, I added several justifications and comments in the Manuscript (highlighted in yellow). This includes an explanation that “the distinction between c and f is necessary because, if the natural resource and the other goods are not entirely substitutable and the ratio between them varies over time, then social welfare depends on the ratio of c to f over time and cannot be written as a function of a single variable (30).” (Lines 105-108). I also added an explanation about the various measures of the discount rate, and I clarify that I consider the social rate of time preference (lines 87-97; see more on this below).

In the second part, I specify how $f(t)$ and $c(t)$ depend on the harvest (still without specifying how the managers chose how to harvest). I explain that “ c and f characterize the total provision of the two goods at the global scale, and accordingly, we consider a large ecosystem that comprises a large number of distinct regions (Fig. 1).” (Lines 135-137). I also explain that “we are interested in long-lasting effects of the harvesting on the provision of the natural resource, and therefore, we focus on irreversible degradations of the ecosystem, rather than on temporary fluctuations of the resource stock. These degradations may occur, for example, if some ecosystem services are permanently lost (5) or if the ecosystem providing the renewable resource collapses or undergoes an irreversible regime shift in some regions, such as occurs in eutrophication and deforestation (3, 4, 10).” (Lines 141-146). (Note that the dynamical variables, x_1 and x_2 , characterize the portion of the ecosystem that has not yet collapsed.) In turn, I explain that “... some portion of the ecosystem, $H_n(t)$, becomes degraded during that year (Fig. 1). These portions are determined by the various harvest methods that are being used worldwide. For example, $H_n(t)$ may characterize the portion of the global fish or timber stock that is no longer available due to the collapse of fisheries or the irreversible degradation of forests worldwide (40)” (lines 148-151). In other words, I do not assume that there are only two harvest methods. All we need to assume is that, at the global scale (e.g., worldwide) there are some degradations that occur at a rate H_n , which is determined by the various harvest methods that are used by the various managers worldwide. For example, some managers may use a method that does not degrade their fishery, while some others may choose a method that causes the eutrophication of some large areas of another fishery; ultimately, H_n is the portion of the system that becomes degraded in year t .

Finally, in the third part, I specify how the aggregate harvest strategy is determined by the managers. I explain that “We are interested in comparing two types of solutions: (1) The optimal solution that maximizes the social welfare and (2) the market solution that emerges in a competitive market. The optimal solution is found via the maximization of social welfare (Eq. 2) subject to the constraints given in Eqs. 4-7 (Methods). In turn, the market solution depends on the form of the externalities for the various managers, namely, it depends on how non-sustainable harvest by a given manager affects the ecosystem in regions managed by other managers. For the theorem, we consider a general case in which the market solution implies a higher rate of non-sustainable harvest. For our simulations, however, we specify here a special case in which we distinguish between the externalities in the managed and the shared regions.” (Lines 184-191.) Namely, I explain that the theorem applies more generally and does not necessitate specific assumptions about how H_n is determined. (One only needs to assume that, in the market solution, H_n is positive during some time period, irrespective of the reason). In turn, I extended the theorem to include the more general case: “... for any Δ that emerges if non-sustainable harvest ($H_n(t) > 0$) occurs between times t_0 and t_1 , there exists $t_c > t_1$ such that $\Delta(t_c) \leq \Delta_{sus}(t_c)$.” (Lines 256-258.) For the simulations, however, I still consider a specific case, and I distinguish between the open-access regions and the managed regions.

In addition, I made a new version of Fig. 1, and I believe that the new figure does a better job of explaining the model and the meaning of the harvest functions.

The Reviewer inquires: Why did I assume that the managers can manage the ecosystem only sustainably or non-sustainably?

As explained above, I do not have this assumption in the revised Manuscript. I only assume that the portion of the ecosystem that becomes degraded in a given year, H_n , is determined by the collective management of the managers (each in her/his own region). If more managers choose to use less sustainable methods, then H_n will increase, but each manager can use methods that range from the most sustainable ones to the least sustainable ones: "... manager determines a harvest method, which may vary anywhere between using only sustainable methods to using only non-sustainable methods." (Lines 192-194.) See also Fig. 1.

The Reviewer inquires: Why did I allow open-access regions to be managed sustainably?

Note that my results demonstrate that the shared regions are not being harvested sustainably. (The dashed red lines in Fig. 2C,D coincide with the zero line, and I emphasize in the revised Manuscript that the managers harvest the resource in the shared regions non-sustainably in line 446). This is in accord with the argument by the Reviewer. The fact that I consider the possibility that the open-access regions could be managed sustainably is legitimate. The fact that this does not happen only emphasizes the consistency of my results with what is known from the literature. In other words, there is no need to assume a priori that a sustainable harvest is impossible in open-access regions; instead, I allow this property to emerge naturally. Of course, I could assume that a sustainable harvest is not allowed in the open-access regions, but this would have made no difference in the results shown in Figs. 2 and 3 (since this option is not being used).

The Reviewer comments that I assume that the decision makers could measure and agree on the discount rate.

I believe that this inquiry refers more generally to how policymakers choose the discount rate and to the fact that there are numerous disagreements about its value. There are even disagreements concerning how this rate should be measured. Note that I do not argue that policymakers agree on the value of the discount rate or on how it should be measured. This is why I avoid claims of such as "the value of the discount factor is X." Instead, I focus on the role of over-harvest and how it would affect the value of future goods.

I have made several changes in the Manuscript to clarify this issue. In particular, in the beginning of the Model section, I added a paragraph that explains the different ways to measure the discount rate (lines 87-97), e.g., "There is no consensus on the best measure of the social rate of discount, but the common measures are the social rate of time preference,

the social and private rates of return to investment, or some combination of the three (15, 37)...” (Lines 87-89.) In turn, I note that, in this study, the focus is on the social rate of time preference.

In addition, in the revised Manuscript, I discuss further the various debates concerning the value of the discount rate. For example, one of the most controversial issues is whether policymakers need to include the pure time preference, ρ , when discounting the far future. As I mention in the revised Manuscript, “some authors argue that its market-based value of $\sim 3\%$ should be used, but others argue that $\rho \approx 0$ should be used based on considerations of intergenerational equity (6, 16, 19, 42).” (Lines 302-304.) Nevertheless, the adjustment that I suggest to the discount factor, $\exp(\phi_\tau)$, does not depend on the value of ρ : “The value of ϕ_τ , however, does not depend on ρ and should be subtracted from Δ regardless of the value of ρ .” (Lines 304-305.) For example, advocates of $\rho = 2\%$ may suggest a value of, say, 10\$ per unit of goods 100 years from now, whereas advocated of $\rho = 0$ may suggest a value that is ~ 3 -times higher for the same good (30\$ per unit). However, my analysis shows that this value should be multiplied by $\exp(\phi_\tau)$, regardless of the choice of ρ . Namely, if $\exp(\phi_\tau) = 2$ and $\rho = 2\%$ then the study suggests that the value of the good would be 20\$ instead of 10\$, whereas if $\rho = 0$, then the value would be 60\$ instead of 30\$). Resolving the existing controversies around discounting is not the objective of my analysis; rather, its objective is to suggest a correction that would apply on top of the existing paradigms and should be accepted on both “sides”. I elaborate more on that issue below in response to a related comment by the Reviewer.

In the revised Manuscript, I also added an analysis showing the magnitude of the adjustment to the discount factor suggested by my results for various parameter values (Figs. 4 and 5 and lines 310-329).

Why I did not use a “traditional” form of a resource stock model (and how my model is a special case of such a model)

The Reviewer inquires why I did not use a more traditional resource stock dynamic model. Such models examine the dynamics of the resource as a continuous variable (e.g., $df/dt = g(f) - H$). The first reason is that I consider the resource stock at the global scale, and therefore, considering a non-spatial model would clearly not be a good choice. I cannot assume that the global fish stock is represented by a single variable characterizing some average fish density. There is a critical difference between (1) a state in which half of the fisheries has collapsed and the other half have a fish density of X fish per unit area and (2) a state in which all fisheries have a fish density of $X/2$ per unit area.

Of course, this has been recognized by many other researchers, and models of renewable resources on large-scale often include multiple patches, each of which has its own resource stock. For example, some models use N coupled differential equations, where N is the number

of patches, e.g., Sanchirico, J. N., & Wilen, J. E. (1999). *J. Environ. Econ. Manag.*, 37(2), 129-150, and Sanchirico, J. N., & Wilen, J. E. (2005). *J. Environ. Econ. Manag.*, 50(1), 23-46.

There are two major reasons, however, why I do not use this spatial formalism either. First, as I explain in the revised Manuscript, “We are interested in the long-lasting effects of the harvesting on the provision of the natural resource, and therefore, we focus on irreversible degradations of the ecosystem, rather than on temporary fluctuations of the resource stock” (Lines 141-143.) Then, I explain that these degradations may occur due to a collapse of the ecosystem providing the renewable resource in certain regions (lines 143-146). These collapses are important ingredients of the dynamics, and they have been incorporated extensively in dynamic population models (3, 4, 10). One way to incorporate the idea that the population in each patch can collapse is to assume that, in each patch, if the population goes below a certain threshold, it will not recover naturally or will be very slow to recover (hysteresis). Another way to incorporate collapses is to consider both the resource stock and the state of the ecosystem as two distinct dynamical variables in each patch. In turn, if the ecosystem becomes degraded, it does not provide the renewable resource (e.g., a fish population cannot recover if the system undergoes eutrophication). In a sense, my model is a simplified special case of this kind of spatial models with hysteresis in which I consider only two states in each patch: degraded (collapsed) and non-degraded (functional). This is a reasonable simplification because I consider long time-scales. Then, the variables x_i simply characterize the total portion of the fisheries that have not collapsed, and the function $H_n(t)$ simply characterizes the portion of the patches that collapse during year t .

The other reason why I do not use an explicit spatial model is that solving and presenting the market dynamics (where each manager manages a single patch and there are externalities) would be highly complex and would deviate from the main point of the paper. Most of the other authors that used space-explicit models solved their models only from a social planner’s perspective. In summary, it makes sense to use my model because (1) it captures the essence of what I am trying to model and (2) it is a model that can be solved and its results can be presented. Presenting a more complex model where the resource stock varies continuously in each location in space would deviate from the main point of the paper, and it would be impractical to solve.

Detailed comments

Line 1-2. A major challenge in environmental policymaking is determining whether, where and how fast we should adopt sustainable management. This is quite a strange sentence. Who is “we” – a wise and environmentally conscious manager or an average businessman in say the oil sector? Or the government? If it is an environmentalist or in fact any reasonable social planner it would be strange to answer the question above by deciding to do unsustainable management. However, if it is an entrepreneur in the oil sector they may not even think of this as a choice let alone a challenging one.

The study is about maximizing social welfare. Accordingly, I changed “we” to “our society” in line 1.

Lines 8-14 this key sentence in the abstract shows both the niceness and the naivety of this paper. It is a nice observation but it seems to suggest a belief that the policy maker believes in the Ramsey rule (or something similar) and actually believes in frequent updates of this or some similar rule to set the social discount rate and so forth. In reality little of this necessarily applies. Lines 14- 16 Therefore, when discounting the far-future, we should use an adjusted discount formula that I derive here and that implies significantly higher net costs due to future environmental damages. The author needs to decide if he is using discount rates normatively or positively. (See also below)

The Reviewer mentions that I do not specify in the Abstract whether I use discount rates normatively or positively. The short answer is that the correction that I suggest should be accepted both on those that consider discounting normatively and positively. Basically, my model predictions can be tested empirically, namely, one may be able to measure whether the discount rate has declined and to prove (disprove) my theory in the future. In that sense, the approach in the model is positive. Nevertheless, the adjustment that I suggest to the discount can be considered in addition to (not instead of) corrections suggested by previous studies that have used “normative” considerations, such as the correction to the pure time preference (see my response to the main comment above). Therefore, the corrections that I propose should be accepted on both “sides”. I elaborate more on this issue below in my reply to another major comment by the Reviewer.

Lines 31-32 relationship between CBA and optimal management are more complicated...

I changed the sentence to “one approach is to consider a social planner whose objective is to maximize social welfare...” (Lines 32-33.)

Line 44-45 Then, the value of a good that worth \$1M today would be worth ~50K\$ for the same good in 100 years if $\delta = 3\%$, and only ~2.5K\$ if $\delta = 6\%$. The author managed to phrase this the wrong way round. It should be “the value of a good worth \$1M in a hundred years would be worth ~50K\$” today.

I revised the above-mentioned sentence (lines 42-43).

55-56 Footnote 11 refers to a working paper at RFF. I believe it was published in REEP Arrow, K., M L. Cropper, C Gollier, B Groom, G M. Heal, R G. Newell, W D. Nordhaus, R S. Pindyck, W A. Pizer, P Portney, T Sterner, R Tol and M, L. Weitzman (2014) “Should Governments Use a Declining Discount Rate in Project Analysis?” Review of Environmental Economics and Policy,

volume 8, issue 2, pp. 145–163 doi:10.1093/reep/reu008. But you should check because there are several similar papers.

I corrected the reference (Ref. 16).

82- Model selection. It is somewhat strange ecology and strange economy with these four separate regions (they are typically not so separate) and these three choices of harvest that is “sustainable” unsustainable or 0.

The Reviewer restates a concern from the main comment. See my detailed reply above. In short, I do not limit the variety of harvest methods used in each site. I do assume, however, that the ecosystem is spatially-extended, and that the ecosystem may be either degraded or non-degraded in each location. Also, I assume that each region is either managed by a single manager or is subject open-access, where this specific assumption is used for the simulations but is not needed for the theorem.

L 91-96. Somewhat strange, non-standard way of presenting the natural resource model. There are just two states yielding $\alpha\beta$ or β units but leading in the former case to a share of the ecosystem κ becoming unproductive..etc. It is more common to work with continuous and more general functions. On line 97 α is used for the growth rate of fish stocks – so that I think is confusing – maybe the author intended to use a separate variable here. Otherwise it looks like alfa is given two different meanings.

The Reviewer restates a concern from the main comment. See my detailed reply above. I consider that a given manager can use any method, but he/she can increase the yield in a given year by using less sustainable methods: “we assume that, in each managed region, a single manager determines a harvest method, which may vary anywhere between using only sustainable methods to using only non-sustainable methods. In turn, the harvest method determines the portion of the region that is being harvested and the rate at which the region becomes degraded (Fig. 1).” (Lines 192-195.) Specifically, the strategy used by a manager (in a managed region) determines the portion of the region that becomes degraded, i.e., $0 \leq y \leq 1$, where $y = 0$ occurs when the most sustainable methods are used and $y = 1$ occurs when the most non-sustainable methods are used. In turn, the yield per unit area is given by $\beta(y + (1 - y)\alpha)$, where $\alpha < 1$. Also, I removed the symbol α from the example to avoid confusion (lines 158-161).

105 formula (2) I find it very strange that the harvest just depends on the areas that are harvested sustainably or unsustainably. There are many levels and ways of being unsustainable. Harvest depends on how many people harvest unsustainably at a certain time. There are important stock-flow interactions. The reason people harvest unsustainably is notably because they get gigantic harvests – not for a trivial time but for decades --- then the stock may crash. I don’t find the basic

production structure specified to be believable or informative and this makes me more than worried for the “derivations”.

The Reviewer restates a concern from the main comment. See my detailed reply above. For simplicity, I focus on the portion of the ecosystem or the ecosystem services that have been degraded. I give the following examples: “For example, $H_n(t)$ may characterize the portion of the global fish or timber stock that is no longer being provided due to the collapse of fisheries or the irreversible degradation of forests worldwide (40). For another example, $H_n(t)$ may characterize the persistent reduction in crop yield per unit area caused by the degradation of vital ecosystem services and the increase in the persistence of pests due to pesticides (33, 34).”

Also, the Reviewer mentions that: “The reason people harvest unsustainably is notably because they get gigantic harvests – not for a trivial time but for decades.” Indeed, this is the case in my model as well. In Fig. 2, for example, $\alpha = 0.02$, which means that the non-sustainable harvest yields 50 times more resource than the sustainable harvest in a given year. Also, non-sustainable harvest takes place over ~ 40 years (Fig. 2C,D).

108-110 sees the introduction of private / open access land. But again this happens in a strange way. It seems the agents can choose to harvest either sustainably or unsustainably on private land. That is fine but it seems they can also do this in open access areas. In the whole literature on Open Access, this is normally excluded. Even if one agent would try to behave sustainably in an open access area – the whole meaning of “open access” is that an undefined number of other agents are taking decisions on the same resource and therefore sustainable use is excluded and not possible. See for instance the following sentence: Consequently, each agent has four possible actions as he/she can decide how much to harvest sustainably and non-sustainably both on his/her private land and on the shared land. The last part is not generally thought to be possible in open-access.

The Reviewer restates a concern from the main comment. See my detailed reply above. There is no need to assume a priori that sustainable harvest is impossible in open-access regions. Instead, I this property emerge naturally, and the results show that the sustainable harvest of open-access regions does not happen.

118-125. some kind of cost structure is discussed but I find the sentences very confusing. Investment and costs of production appear mixed. The cost of production in the resource sector seems to depend on what happens in the sector for production of other goods but I do not see how.

The reviewer asks for a better explanation of Eq. 7. In the revised Manuscript, I explain that “we assume that the harvest at time t comes with a cost as more labor and resources are directed toward harvesting. We incorporate this cost in the model as a reduction in the consumption of the other goods, which would otherwise grow exponentially at an exogenous

rate g_c due to technological developments” (Lines 176-179). Note that the cost terms are relevant for the early stages of the economy (essentially $t < t_1$). In the later stages, the cost terms become negligible compared to the term $c_0 \exp(g_c t)$.

126 to 141 and formula 6 describe welfare which is a function of consumption and of natural resources. I fail to see why natural resources enter into welfare. If it had been scenic beauty then I could understand it but from earlier discussion it is clear that f refers to things like fish and timber and in that case they would surely already be included in c ? Maybe people consume fish and computers and possibly this approach could be defended but I do not think it has been explained or motivated.

The Reviewer asks why we need to consider c and f separately in the utility function. In the revised Manuscript, I explain that: “The distinction between c and f is necessary because, if the natural resource and the other goods are not entirely substitutable and the ratio between them varies over time, then social welfare depends on the ratio between c and f over time and cannot be written as a function of a single variable (30).” This is a very common and a well-established assumption, where many authors have used the welfare function given by Eq. 2 (e.g., Refs. 12, 25, 35, 36).

The author is very clearly not an economist. He uses quite a few words, investment, inflation, general equilibrium, perfect market economy, open access but in somewhat strange ways that sometimes at least are definitely wrong, and his model is very unorthodox. Of course in principle we might learn something from new approaches and I do not want to come across as thinking that only economists can write about economics but when no effort is made to connect to standard models then ultimately there is little learning or communication. I find the article bewildering and I fear that all other economists will feel the same way. The article is about welfare, production, resources and in particular discounting and growth – so if the article is not intended for economists then one wonders who the readers will be.

The Reviewer comments that some terminology and jargon are used inaccurately. According, I revised the Manuscript to make sure that the terms are defined correctly. Among the terms mentioned by the Reviewer, only open-access and perfectly competitive market are used in the Manuscript. The open-access area is used as an example to a shared region where all or multiple managers are free to harvest. In turn, I use the term “perfectly competitive market with perfect foresight” to describe the condition under which, if there are no externalities, social welfare is maximized in equilibrium (lines 203-204). This same terminology is used, for example, in Xepapadeas A. (2005) (Ref. 12), who argues that: “It is known that in the absence of externalities there is an equivalence between the outcome of the social planner’s problem and the outcome of the competitive equilibrium with perfect foresight (Becker and Boyd 1997).” (End of page 17 and beginning of page 18 there.)

231-232 “Our study shows that the transition to sustainable harvest after a period of over-harvesting leads to a decline in welfare, economic growth, and in the discount rate (Fig. 2). While I find many details of the model confusing I recognize the conclusion at least for the welfare. You can imagine a period of overharvesting of natural resources that leads to an eventual decline in welfare. This is correct although not very surprising. The question of what happens to growth and discount rate is complicated by questions of definition. It may be true that growth as generally measured by conventional measures such as GDP decrease. However if income were properly measured then it would not have been so high in the period of “unsustainable growth”. All the damage being done to stocks of ecosystem resources should have been properly valued. In this case income and growth would not be so high in the first period and thus the “decline” in economic growth may not happen. I suspect it would be the same for the discount rate although again it depends somewhat on how this is defined. It is not common to discuss it as if it were an independent parameter of the system. Many economists describe it in very different ways.

The Reviewer comments that “if income were properly measured then it would not have been so high in the period of unsustainable growth.” Then: “All the damage being done to stocks of ecosystem resources should have been properly valued.” This would imply, according to the Reviewer, that “... income and growth would not be so high in the first period and thus the “decline” in economic growth may not happen.”

I like this nice explanation that the Reviewer provides. Indeed, this is why the optimal solution shows no sign of any decline in economic growth or discount (Fig. 2A,B). The social planner is getting prepared for the forthcoming limit on the harvest by avoiding over-harvesting already in the early stages. However, the decline in f can be fully avoided only if there are no externalities. The presence of externalities (e.g., open-access regions) leads to over-harvesting even though the managers are aware of the forthcoming decline in the resource (e.g., Fig. 2C,D). The predictability of the decline mitigates its magnitude but does not eliminate it. This is because there is no way to “save” the resource in the open-access regions (only to harvest even less in the managed regions, but this may not be enough). Note that, in my model, the managers are fully aware of the future decline, but they still over-harvesting because of the externalities (if they did not harvest, some other manager would).

In the revised Manuscript, I explain that “the optimal solution exhibits no such decline in economic growth or in Δ because the social planner plans for the forthcoming constraints on the harvest by avoiding over-harvesting in the early stages ($t < t_0$); in the market solution, managers also take into account the forthcoming decline in f and avoid non-sustainable harvesting in the managed regions prior to time $t = t_1$, but they still over-harvest in the shared regions.” (Lines 231-235.)

I think this touches on a central point for the author who writes: Consequently, we should consider the expected future discount rates to be lower than today’s discount rate, which implies that environmental damages will incur higher net costs. Ignoring the harvest-induced decline in

cumulative discount not only falsify cost-benefit analyses, it also creates a bias: Over-harvesting increases the discount rate in the short-run, which unjustifiably biases our expectations to anticipate higher future discount rates, which, in turn, is used to justify further exploitation. I sort of like this sentence and the sentiment it conveys. It is cute and rings true but one has to know that the discussion of discount rates is much complicated by question of “whose” discount rates we are discussing. The author’s discussion is so unusual that it is a little hard to judge but I would think he is discussing the optimal discount rate in a normative sense for an omnipotent social planner. The trouble is that social planners hardly exist, at least they are rarely omnipotent and they don’t really make the decisions about forestry and fishing... When there are dictators who do make such decisions it seems they often maximize other functions. The authors who discuss discount rates generally fall into one of two groups. Those who think it is a normative ethical parameter that cannot be observed and that the social planner should choose to guide her decisions. In this case it follows something like the Ramsey Rule. The other group thinks it is endogenous to the economy and can be observed. This group however does not believe in any Ramsey rule or in social planning, welfare maximisation in this sense. They typically think the discount rate should match the observed rate of interest on secure bonds. The author of this paper seems generally closer to the first group but then speaks of the discount rates movements as something highly observable in spite of the fact that it is the sum of a number of second derivatives of very special production and consumption functions he has chosen. I think the paper in its current form would be much criticized or maybe ignored. I think however it is possible (in fact likely) that the main insight could survive a major transformation. I think a conventional economic model could be formulated with an ecosystem with stock and flow properties. The effects on growth of harvest decisions could be discussed and could be quite similar. I think one could also discuss the “cute effect” on the optimal normative social discount rate as it is usually defined.

The Reviewer inquires about two related issues. First, the Reviewer inquires whether the change in the discount rate that I propose is only to be used by an omnipotent social planner, or, is it “real”, in the sense that it should also be used by, say, selfish managers who would like to maximize their own profit. Second, the Reviewer inquires whether I consider the discount from a normative or from a positive point of views.

First, I emphasize that the discount that I predict is indeed “real”. It should be used, in principle, not only by an omnipotent social planner. My results predict that the discount rate will actually be lower in the real market in the future. Nevertheless, it can take decades before this decline will occur. Therefore, the study is relevant only when considering policies that will affect the far future. This also answers the core of the second part of the comment. Namely, my results can be tested empirically, at least in principle, by observing the changes in the market discount rate in the far future.

Next, to complete the answer to the second part of the comment, note that a few economists might take an extremely “positive” approach (e.g., argue that we should never consider a

decline in discount), and a few might take an extremely “normative” approach (e.g., argue that we should not discount at all), but most economists take some intermediate approaches. Specifically, the core of the debate around discounting encompasses the following three issues.

1. Should we use a declining schedule of discount starting today?

Some economists (e.g., Weizmann, Gollier, Refs. 16, 20-23) have shown that, due to uncertainty about future economic growth, social planners should aim to maximize the expected NPV, which is equivalent to considering a declining schedule of discount (normative approach). On the other hand, the market does not show evidence for such a decline during the next 30-40 years (Ref. 15), so a positive approach would imply considering the market discount, at least for the next 30-40 years, in contrast to the suggestion imposed by normative considerations.

My study, however, raises no such contradiction. In the revised Manuscript, I explain that: “This result suggests that, in the long run, the discount factor is not sensitive to the rate of discount in the short run. Specifically, the result does not suggest that the discount rate is given by δ_{sus} starting today, but what it does suggest is that, if the discount rate remained δ_{today} for the next τ years, then the shock, ϕ_{τ} , would be greater than ϕ_0 (Fig. 4)...” (Lines 288-291.) Then, I explain that “This result bridges the gap between advocates of declining schedules of discount rates and advocates of considering the discount rate that is predicted by the market, which does not show evidence of any decline during the next 30-40 years (15). Specifically, the discount rate could continue at its present rate for several decades and decline only afterward, which would have a similar effect on the discount factor in the long-run (Fig. 4A).” (Lines 297-301.)

2. Should we determine the value of the pure time preference based on the market or based on ethical considerations?

I explain in the revised Manuscript that “the value of the pure time preference, ρ (Eqs. 2,3), is controversial; some authors argue that its market-based value of $\sim 3\%$ should be used, but others argue that $\rho \approx 0$ should be used based on considerations of intergenerational equity (6, 16, 19, 42).” (Lines 303-305.) I also explain that this debate has no effect on the adjustment to the discount that we suggest: “The value of ϕ_{τ} , however, does not depend on ρ and should be subtracted from Δ regardless of the value of ρ .” (Lines 305-306.) See more on that in my response to the main comment.

3. Should we consider the social rate of time preference or the social rate of return to investment when we discount the future?

The interesting debate around this issue (e.g., Refs. 15, 37) is different from the debate between the positive and the normative economists. But I mention it here because it is another

major debate that is related to how we should discount the future. In my model, I consider the social rate of time preference, but since the measures are strongly correlated, the prediction that one measure is declining also suggests that the other measures are declining. I discuss this issue in lines 87-97.

I believe the intuition about the role of discount rates for motivating investments that are relevant for our future is partly true. But the author must signal more clearly that he understands this only refers to an ideological battle in arenas like the Stern Review, the IPCC and similar. Actual resource decisions are generally made with little recourse to discounting – or based on horrendously high rates motivated by the cost of actually procuring capital or the profitability “demands” of companies.

The Reviewer comments that, in practice, resource decisions are made with little attention to changes in the discount rate. I agree that the study is relevant for policies that affect the far future, such as climate policies. For example, the value of the discount in the far future may have a significant effect on the calculations of the social cost of carbon. In the revised Manuscript I explain that “Accurate discounting is particularly important for environmental policies in which the resultant damages are long-term, such as policies concerning climate change and provision of natural resources (6, 17, 18).” (Lines 39-40.) I also emphasize throughout the Discussion that the results are relevant only for policies with long-term environmental consequences.

I have the feeling most of my economics colleagues would put a simple reject on this paper. I am sympathetic to its main direction. However, if it is going to convince anybody then it has to be written in a more accessible style. It is after all mainly directed to economists and must therefore either be written as economists analyse natural resource and growth issues or at least refer to such models and choose and motivate any necessary departures. As it currently stands much of the material will be an obstacle to communication. This applies for instance to the whole discussion of open access and private resources used in the early part of model construction and in figure 1. The whole idea of only having two levels of production one for sustainable management and one for unsustainable management is hard to accept. As I mentioned there is a lot of dynamics there. The role of the two goods must be clarified better, how they interact and contribute to welfare. How production and costs are decided. I am sorry to sound like a boring conventional economist but if you write an article about growth and discounting then that is large part of your audience – and if you build a model that is completely different then the burden of proof is on you. I think your vital points could be made more effectively in quite a conventional model.

The Reviewer restates the same concerns that he/she raises in the main comment and in the detailed comments. See my detailed reply above.

Figures 2-3 give some very precise impression but clearly they are some very special case of your model and you hardly comment on them in the text. I am not sure what their function is.

As commented by the Reviewer, Figs. 2 and 3 present some special cases. Indeed, this is their purpose. The more general result is given in the theorem. In the revised Manuscript, I have added Figs. 4 and 5, which demonstrate the lower bound on how much the discount must decline based on the theorem.

Reviewer 3

I thank the Reviewer for carefully reading the Manuscript and for the helpful comments. Below please find each of the Reviewer's comments (black) followed by my response (blue).

Thank you for offering the opportunity to review the manuscript entitled, "Over-exploitation of natural resources is followed by an inevitable decline in economic growth and discount rate." I found the submission to be fairly well written and motivated. However, I offer some questions and comments below that could possibly improve the submission over its current form.

Major Comments

1. The submitting author is not the first person to recognize the importance of the rate of discount on the sustainable allocation of scarce resources. There is a long and rich history (and discussion), pertaining to the optimal discount rate, in the resource economics literature. The author's model does a somewhat adequate job of explaining how the discount factor is affected by changes to the harvest rate. However, the author's arguments are less clear in terms of the effect of the discount rate on social welfare.

A. I would encourage the author to read the following text, from which I lean on heavily in this comment:

Bergstrom, J.C., and A. Randall. 2010. Resource economics: An economic approach to natural resource and environmental policy, 3rd edition. Edward Elgar: Northampton, MA.

Specifically, I refer to page 379 of the book, which explains that discounting is motivated by the productivity of capital – that is, its purpose is to provide optimal incentives for saving and for allocating scarce capital efficiently. Bergstrom and Randall (2010) go on to argue that: "...[discounting] of utility itself makes little sense but, with productive capital, consumption will be discounted to reflect the reward for saving and investment." Thus, the author needs to clarify what exactly is being discounted.

B. Where I get confused in the author's model is the difference between the parameter (ρ) in equation (6) (which presumably represents the time value of money) and (δ) in equation (7) (which represents the discount factor). Are the time value of money and the discount factor related? If so, how? I have sometimes seen the assumption that ρ and the discount factor are related as $\rho = \frac{1}{(1+\delta)}$.

C. The author should also discuss the difference between private and social discount rates. In regards to the social discount rate, Bergstrom and Randall (2010) question whether should reflect the productivity of capital at the margin or for society as a whole. They go on to argue that the social discount rate should reflect the size and timescale of

the resource under investigation. For small projects with modest time horizons, the discount rate should reflect the marginal efficiency of capital. For large projects with very long time horizons, the discount rate should reflect the real growth rate of the economy. Is this the difference in the near- and long-term discount rate that the current author finds in his own work? Regardless, this needs to be discussed with greater thoroughness and clarity in the submitting author's manuscript.

The Reviewer refers to the long discussion in the literature about discounting and is asking for better clarifications of the relationship between the discount rate and social welfare and for a better discussion of the related literature. The Reviewer also specifies three detailed sub-comments, points A-C. Here I first refer to the main comment more generally, and then I address points A-C specifically.

The relationship between the discount rate and social welfare and the connection of my model to the discounting literature

In the revised Manuscript, I describe the relationship between the discount rate and social welfare in the first part of the Model section, lines 87-134 (see also Appendices A and B). First, I emphasize that my derivations in that part are well-established in the literature (lines 81-85). Namely, the novel part of my study is not the equations for social welfare and discount (Eqs. 2 and 3). Rather, the study is novel because previous studies that considered a utility that depends on both goods assumed that f and c simply increase exponentially (e.g., Refs. 25, 35, 36). In contrast, I study how harvest affects the dynamics of c and f (Eqs. 4-7), which, in turn, affect the discount rate.

Next, in a new paragraph that I added to address this comment, I give an overview of the various ways to define the discount and the relationships between them (lines 87-102). (See more on this below when I address points B and C.) Subsequently, I note that I consider the utility function given by Eq. 2, which has been used extensively in the literature (e.g., Refs. 12, 25, 35, 36), and I provide justifications and details in lines 104-113. In turn, Eq. 3, which provides the formula for the discount rate, simply follows from the definition of the discount rate (given in lines 93-97) when social welfare is given by Eq. 2, as I show in Appendix A.

Point A

The Reviewer asks that I clarify whether I consider discounting of the utility itself or discounting that reflects the reward for saving and investment. The simple answer is that, in this study, I consider how the "reward for saving and investment" is affected by harvest, thereby affecting the discount. Consider, for example, the Ramsey discount formula (using the same notations used in the Manuscript):

$$\delta = \eta g + \rho.$$

Using Bergstrom's terminology (and my notations), δ is the discount rate, the term ηg reflects the "reward for saving and investment", and the term ρ is due to "discounting of utility itself". The argument by Bergstrom *et al.* that "discounting of utility itself makes little sense" simply

implies that ρ should be zero. (Note that there is no agreement about that and some economists argue that ρ should be determined by the market, e.g., $\rho = 2\%$ is used in the DICE model, but in any case, my results do not depend on the value of ρ .)

Back to my model, instead of Ramsey's discount formula, I consider Eq. 3:

$$\delta = X + \rho,$$

where “ X ” is a complex term that replaces ηg and includes the derivatives of c and f . (The term ηg in Ramsey's formula incorporates the consumption of a single good that grow at a rate g , whereas the term in Eq. 3 incorporates two goods that grow at rates that will be determined later by the harvest.) In other words, in my model, I endogenize the first term in the right-hand-side of Eq. 3 and I examine how it depends on the dynamics of harvest.

In the revised Manuscript, I explain that “The right side of Eq. 3, without the term ρ , is due to the change in the marginal contribution of c and f to welfare. (Note that, if $\mu = 0$ and $dc/dt = cg_c$, then Eq.3 becomes the Ramsey's discount formula (14, 16), $\delta(t) = \eta g_c + \rho$, where $\eta \equiv cu_c/u_{cc}$.)” (Lines 121-123.) Furthermore, in the Discussion, I show that the correction that I suggest for the discount factor, $\exp(\phi_\tau)$, is independent of the value of ρ (lines 303-306).

Point B

The Reviewer asks that I clarify (1) the relationship between the discount rate, δ , and the pure time preference, ρ , and (2) the relationship between the time value of money and the discount factor.

I believe that the explanation for point A also addresses issue (1). In my notations, δ is the discount rate, and ρ is due to “discounting of utility itself” (which I denote by the more common convention “pure time preference”). Back to the comment, the formula that the Reviewer describes represents the relationship between the discount factor and the discount rate in discrete-time representation for a constant discount rate. In continuous-time representation, the discount factor is given by $\exp(-\delta t)$ if δ is constant. More generally, for $\delta(t)$ that varies over time, the discount factor is given by $\exp(-\Delta)$ where $\delta = d\Delta/dt$, as I explain in the revised Manuscript (line 121). To avoid any misinterpretation of the symbols δ and ρ , I clarify their definitions in the text, and I also added a table of symbols (Table 1). Also, note that, in the original submission, there was a typo in which the term ρ did not appear in the right-hand-side of Eq. 3. In the revised Manuscript I corrected the typo. Also, as a side comment, note that different authors use different notations. Bergstrom *et al.* use r to denote the discount rate, some other sources use δ (like myself), and some even use ρ (and then use a different symbol for the pure time preference). In the Manuscript, I use the same notations and the same terminology used in the review by Groom *et al.* (Ref. 15).

To address issue (2), I added a paragraph that specifies the relationship between the discount rate and the marginal productivity of capital (e.g., the time value of money) (lines 87-102). Specifically, I explain that “There is no consensus on the best measure of the social rate of

discount, but the common measures are the social rate of time preference, the social and private rates of return to investment, or some combination of the three (15, 37).” (Lines 88-90.) In turn, I explain that “these three measures are strongly correlated, and in a market with no distortions (e.g., a perfectly competitive market), all three measures become equal and reflect the marginal productivity of capital (15, 38, 39).” (Lines 91-93.) I also note that, in this study, I focus on the social rate of time preference (line 93), and I give its definition in lines 94-97.

Point C

The Reviewer asks that I better discuss two issues: (1) the difference between private and social discount rates, and (2) the difference between marginal changes and large changes to the economy and how it affects the discount.

Issue (1) is addressed in the new paragraph that explains the various measures for the social rate of discount (lines 87-97). Specifically, I mention the social rate of time preference and the social and private rates of return to investment as three common measures. I note that “these three measures are strongly correlated, and in a market with no distortions (e.g., a perfectly competitive market), all three measures become equal and reflect the marginal productivity of capital (15, 38, 39). (Lines 91-93.)

Regarding issue (2), the Reviewer is correct that the mechanism described by Bergstrom *et al.*, in which “for large projects with very long time horizons, the discount rate should reflect the real growth rate of the economy,” is precisely the mechanism for the change in the discount shown in the Manuscript. Specifically, in the context of the Manuscript, the large changes are due to changes in the provision of the natural resource, which impose a non-marginal change in welfare. In the revised Manuscript, I explain that “the discount rate may be affected by large perturbations that significantly affect social welfare, such as environmental degradations that occur due to climate change and resource harvest (24, 25). Particularly, the utilization of natural resources may affect the future provision of food and ecosystem services and have a significant, non-marginal effect on social welfare (24, 26-30). Several authors showed that these global changes in the provision of non-substitutable natural resources might affect their relative price (31, 32) and the discount rate (24, 25).” (Lines 56-61.) Specifically, in the context of the present paper, I explain that: “In this paper, I examine how the discount rate is affected by large changes in the provision of a natural resource, which could be imposed by large changes in harvest patterns at the global scale.” (Lines 64-65.)

2. The submitting author’s model is quite long and requires several equations. I am curious why the author chose not to solve this problem using optimal control theory. To me, it would seem easier for the reader to follow if the model was set up as maximization problem with the social welfare equation (6) representing the objective function, subject to a series of constraints (equations 2 - 5). The model is fairly hard to follow in its current form. Of course, the author chose the current specification for a reason, so I am not implying that he has to solve it using optimal control theory; however, it would help to explain why it is solved piece-meal (its current form) over a more structural approach.

The Reviewer inquires why I did not use optimal control to solve the model where Eq. 2 is the objective function under the constraints given by Eqs. 4-7. (Note that the equation numbers have been changed in the revised Manuscript.)

I consider two types of solutions. The first is the optimal solution (by a social planner). This solution is given by the maximization of Eq. 2 under the constraints given in Eqs. 4-7, as suggested by the Reviewer. The second solution, however, is the market solution that incorporates externalities. Over-harvesting may occur because managers will care about their own profits and do not care about the benefits and costs incurred by the other managers. Then, the solution does not maximize social welfare and is sub-optimal from a social planner's perspective. Specifically, the market solution is described in the Methods section lines 330-334.

In the revised Manuscript, I have added a better explanation of this issue in a new paragraph, lines 184-191. In addition, as I noted above, I rearranged the Model section and made numerous changes to improve its clarity.

Minor Comments

1. The current manuscript contains a fair number of grammatical mistakes including the omission of definite and indefinite articles and verbs. The author should thoroughly revise the entire submission.

I went over the Manuscript to remove the grammatical errors.

2. Why is the model solved from a social planner's perspective as opposed to starting from say microeconomic fundamentals (representative household's utility or representative firm's profit maximization)?

The Reviewer inquires about the model used for the market dynamics vs. the optimal solution. The idea of the analysis is to present two types of solutions and to compare them. (This is done in the numerical solution, Fig. 2A,B vs. Fig. 2C,D as well as in the theorem.) The first type of solution is the optimal one (from a social planner's perspective). The second solution, however, which I called the market solution, is solved not from a social planner's perspective. This is the solution that emerges when each manager maximizes her/his profit in a competitive market. I clarify this issue in lines 184-191. In turn, the assumptions used for the simulations of the market solution are given in lines 192-210.

3. I believe that the author should refer to $B(t)$, in equation (1), as the *net* benefit function. It is referenced later in the manuscript as representing net benefits, but this is not stated explicitly in lines 34-39 on page 2.

The description in the text is correct and $B(t)$ in Eq. 1 is the benefit minus the cost at time t . Accordingly, the NPV is given by the net benefit minus the net cost (the integral over $B(t)$).

The terms “net benefit” and “net cost” refer to the integral over time of the benefits and costs (which is why the value of the discount matters). I made a few minor changes throughout the text to avoid the use of the term “net cost” when it might have been unclear.

4. Page two, lines 40-42 – the author should clarify that the benefits to be gained will depend on whether the underlying resource is renewable or non-renewable.

I rewrote this sentence (lines 38-39).

5. Page two, lines 44-47 – the term “worth” is ambiguous (see argument about private versus social discount rate in comment one above). I would encourage the author to change this phrasing to “is valued at...”

I changed the sentence according to the suggestion (lines 42-43).

6. I am not sure if it is necessary, but the submission omits any discussion of steady-state levels – e.g., a steady-state harvest rate.

In the revised Manuscript, I note that “...the asymptotic rates of non-sustainable harvest decreases exponentially (Eq. D28, (41)).” (Lines 215-216.) I do not elaborate more on this issue in the main text as this analysis deviates from the main point of the paper, and there are several similar analyses of the asymptotic rates of harvest in the literature (e.g., Ref. 41).

REVIEWERS' COMMENTS:

Reviewer #2 (Remarks to the Author):

The manuscript has been improved substantially after several rounds of comments and editing.
Reviewers' comments:

Reviewer #1 (Remarks to the Author):

In my view the author has addressed quite satisfactorily my concerns and I believe several of other reviewers. To me a few aspects – probably minor compared to the purposes and mathematical complexity of the article - remain regarding the clarity of economic concepts. I think although I was not able to fully value the completeness of proofs, etc. (I do not have a background in mathematics, but in economics), I think I understood the additional value of the article, and consider that it could be publishable and further discussed among the scientific community.

Comments

“by definition, the total production at time t (which is proportional to GDP if we are focusing on a specific country)”. I think I would not agree with claiming that proportionality given how empirically it is typically accounted for, but I understand the simplifications made in the modelling. I think something similar occurred with Reviewer 2 comment on “if income were properly measured then it would not have been so high in the period of “unsustainable growth”, since perhaps if GDP was measured differently or our (as society) reference variables were other measures incorporating ecological damage (even getting rid of the concept of “externality” itself) we could be talking about different results or less importance of the discount factor (or in this case how harvest of natural resources affects it). Also, with concepts and associations such as “welfare” to “utility”, just basically associated to consumption, and the discount rate in an important thread of literature and here. I would consider interesting recognizing the limitations of such approaches and acknowledging the existence of other economic paradigms and conceptual frameworks.

As highlighted as well by other reviewers, different readers will have different capabilities to go into the equations and proofs, but the logic of the argument and simulations might be easier to grasp for many economists. Somehow following my previous comment on the usefulness of providing complementary results on the sensitivity, robustness, etc. in the Supplementary Material, I consider that providing a kind of tool/application for modelling simulations (e.g. for a chooser to modify at least a few parameters; it could be with a limited choice of them, based on the provided example, etc.) to very intuitively observe the effects, and notably on harvest of natural resources on the discount factor.

Reviewer #2 (Remarks to the Author):

I think there is movement in the right direction when it comes to the theory of dual discounting where I believe that the authors are simply using earlier work by references (35 and 36). This is fine and could be said clearer so that this complex paper becomes a little more readable. Simply starting by saying that you are building on earlier work with dual discounting and they endogenising the

growth rates of the manufactured and environmental goods would remove a first hurdle to understand what your paper is about. Then however you must also focus on explaining how you do this. The part about which areas are sustainably/unsust. managed and how the productivities evolve etc is the crucial addition that might make this paper very interesting – but again I think you write in a confusing manner. If this can be clarified we might have an important contribution. I am afraid I cannot judge this yet.

Detailed thoughts and reactions

I find the paragraph from lines 30-38 including formula 1 quite confused.

You say in the text that you are going to maximize welfare but then the formula has an integral of B.

The most common interpretation would be that B is concrete benefits – like consumption goods (minus costs – since you write net benefits). In that case the formula is wrong because you said you wanted to model welfare which would be based on the Utility of benefits ie $U(B)$ or $W(B)$. Another interpretation of your text would be that you use the symbol B to symbolize the utility of consumption. This would be a somewhat unusual way of wrting. Either way formula 1 just confuses.

You write “The rationale behind discounting is that the objective of our society is to maximize welfare rather than net output” but then your formula does just the opposite and maximizes net output..

Formula 2 does the job in the usual way so I think it might be better to use that straight away. Furthermore B is not in your list of variables but – never mind maybe better to just skip (1).

The new text 87-102 tries to sort things out but verges on confusion..

There are conceptually different measures such as social and private discount rates. You cant say they are “correlated” – that is a statistical term. These rates cannot be measured and we have no idea if they would be correlated. You could say that they are closely related – but that is conceptual not statistical.

Altogether it worries me deeply that the authors seem to think we can somehow go in and measure or discover what the discount rate “actually is”. This is conceptually impossible. It is a theoretical and abstract construct – Look at your own formula (3) which consists of lots of second derivatives of the utility function. How do you imagine you are going to measure this??

This confusion is also reflected in some of their answers to my questions about whether they believe in normative or descriptive approach – and they say both... The only way you can “measure” the discount rate is if you decide it is by definition equal to the (riskless) rate of return on capital. The latter can be measured – but then with that approach you don’t need articles like the one we are discussing nor is their much point in elaborating on equations like (3) since it is anyway an observable rate of return that we are talking about.

To try to be a bit positive and constructive I think the part of the text from (2) to (3) or from lines 103 to 134 is correct and simply restates what has been published earlier as (35) and (36). As long as the

formula (3) says the same as these earlier references then we can feel safe that the text is correct and it might be easiest to stay closer to these texts and as mentioned skip some of the text I criticized above.

The interesting and innovative part starts afterwards in deciding how much c and f will be produced and that is where sustainable and unsustainable harvesting comes in.

I do however have a hard time trying to understand the dynamics described for which areas are managed sustainably or not.

It seems this could be simple enough if you assumed that open access leads to non-sustainable use and a single owner leads to sustainable use but the text is very convoluted and poorly written – I don't know but I think they do NOT make this simple and natural assumption. However I cannot figure out what they do assume. This would seem to be one of the central points so therefore I am lost how they get their results. I think the authors need to make a greater effort to clearly state how the harvesting methods and the areas in which they are practiced evolve.

Do they depend on ownership patterns and degree of open access and exactly how?

Do they depend on the discount rate perhaps that would seem to be interesting...

Fig 1 is visually nice but I am afraid it does not help me understand...

I find the description is verbacious and unclear. I want to know what H means....

In lines 166-167 it sounds like the H variables are areas managed sustainably or not sustainably.

In lines 151-152 it sounds like the H variables are a measure of how much the productivity is lowered in one of these areas.

Later it sounds like x_1 and x_2 are the areas. But it seems that open access areas can be sustainably managed... which is highly confusing.

Several times I find assumptions strange. There is some technical progress β which seems to apply equally to the productivity of degraded and sustainably managed areas.

I think a good and natural way to develop your model would have been to describe the logic of how the areas sustainably and non-sustainably managed develop (as a function of open access). This should be enough. Normally the welfare cost would be a fall in f . I am confused that you introduce an additional cost term c in equation 7 and don't quite understand how this is to be motivated. Is this intended to introduce a varied degree of substitutability between c and f ? (there would be other ways to do that).

Any way I am not sure quite how much of this is retained in your part 3 when you start to compare the "optimal" and "market", solutions. Here I think you should be careful in using the word "market" since it has many ideological connotations and you describe one special case here. If you had no

externalities and well defined ownership everywhere (ie no open access) would the two then be identical?

On page 6 you start describing “results” and refer to figures – but these are not the theoretical results as far as I can see but some very detailed simulation. Such simulations depend on lots of parameter values and I am not sure when and how these were introduced or discussed? As far as I can see, they are just mentioned at the end of the figure captions but not motivated or discussed. the reader is left worrying if the results would be the same with our parameter values. The paper does seem to claim some theoretical results too – for general functional forms but I am not sure of which part of the results and conclusions are based on the general theory results and which part comes from a simulation where I would need to know whether or not these have any general validity...

As for the figures with detailed simulations, I am unsure .. Could you somehow, make us believe that these particular simulations based on these particular numerical assumptions are at all plausible. What would have happened with other values?

Reviewer #3 (Remarks to the Author):

Dear editor,

I believe that the author has satisfactorily addressed all of the comments and concerns that I raised after the initial submission. The author seems to have addressed the other reviewers' comments as well, although I cannot speak on behalf of the other reviewers.

Thank you

Reviewer 1

I sincerely thank the Reviewer for carefully reading the Manuscript once again and for the helpful additional comments. Below please find each of the Reviewer's comments (black) followed by my response (blue).

General comments

In my view the author has addressed quite satisfactorily my concerns and I believe several of other reviewers. To me a few aspects – probably minor compared to the purposes and mathematical complexity of the article - remain regarding the clarity of economic concepts. I think although I was not able to fully value the completeness of proofs, etc. (I do not have a background in mathematics, but in economics), I think I understood the additional value of the article, and consider that it could be publishable and further discussed among the scientific community.

I thank the Reviewer for acknowledging that my responses are quite satisfactory and that only relatively minor issues remain. Below please find my response to the remaining comments.

Comments

“by definition, the total production at time t (which is proportional to GDP if we are focusing on a specific country)”. I think I would not agree with claiming that proportionality given how empirically it is typically accounted for, but I understand the simplifications made in the modelling. I think something similar occurred with Reviewer 2 comment on “if income were properly measured then it would not have been so high in the period of “unsustainable growth”, since perhaps if GDP was measured differently or our (as society) reference variables were other measures incorporating ecological damage (even getting rid of the concept of “externality” itself) we could be talking about different results or less importance of the discount factor (or in this case how harvest of natural resources affects it). Also, with concepts and associations such as “welfare” to “utility”, just basically associated to consumption, and the discount rate in an important thread of literature and here. I would consider interesting recognizing the limitations of such approaches and acknowledging the existence of other economic paradigms and conceptual frameworks.

The Reviewer raises several related issues here, where the main comment is that the discount may depend on the “weight” given to the natural resource when accounting for the total product or economic growth. Specifically, in the model, this is determined by the parameter μ , which specifies the proportion given to the consumption of the natural good in the currency unit. To try and paraphrase this comment in my own words, if the policymakers incorporate only the market goods, then they may define μ as the portion in the basket of goods of the natural resource, whereas if they incorporate non-market goods in the currency unit, then they would consider a higher value of μ . In turn, the value of μ may affect the definition of the discount rate, thereby affecting its value. In light of this, the Reviewer asks that I clarify how

the discount depends on that weight, and that I accordingly acknowledge the existence of paradigms and conceptual frameworks other than discounting.

Let me first respond to this main issue, and then I will address the more subtle issues that the Reviewer raises in the comment. The Reviewer is correct that the discount and the relative prices may depend on μ , and that this is an interesting point to emphasize and discuss. Note, however, that the future value of the natural resource does not depend on the choice of μ . Specifically, the factor of change in the value of the natural resource at time t (compared to the present) is given by $X \times Y$, where X is the factor of change in the price (the price at time t divided by the present price), and Y is the discount factor. In turn, the value of μ may affect each X and Y separately, but it cannot affect the value of $X \times Y$. Specifically, a higher value of μ would imply that the discount would be more sensitive to changes in harvest patterns, while the relative price would be less sensitive to these changes. Similarly, if μ is low, then the discount would hardly respond to changes in harvest patterns, but the price would respond more dramatically to these changes. Overall, the valuation of the future natural resource would not depend on μ .

To address this issue in the Manuscript, I added the following sentences to the Discussion section (lines 312-328): “note that the future values of natural resources do not depend on the proportion given to their consumption in the currency unit, μ . Specifically, their future values do not depend on whether they are accounted for as market or as non-market goods. Nevertheless, μ does affect the relative weights given to the discount factor and to the prices of natural resources in determining the resources’ future values (25, 35, 36). Specifically, ignoring the role of non-market natural resources in economic growth (considering a lower value of μ) would imply that a change in the provision of these resources has a larger effect on their prices but a smaller effect on the discount factor (Appendix A).” In turn, I added the following explanation to Appendix A: “note that the changes in the values of the goods at a given time, which are given by $\delta + v_f$ for the natural resource and by $\delta + v_f$ for the manufactured good, do not depend on μ (Eqs. A13, A17).” (Also, note that the transition to sustainable harvest incorporates not only the decline in the cumulative discount, but also an increase in the price of the natural resource, as pointed out in lines 220-221 and demonstrated in Fig. 2.)

Furthermore, this shows that a complementary approach to discounting would look at the changes in relative prices that would follow the transition to sustainable harvest. This leads naturally to the acknowledgment that the Reviewer asks for. I explain this in the Manuscript in the sentence that follows (lines 318-320): “Therefore, focusing on the inevitable increase in the price of natural resources following their over-harvesting would result in the same conclusion and present an alternative approach to the one presented here.”

Finally, let me address a few other issues that follow from the comment. First, the Reviewer comments about the inaccuracy in claiming that the production is proportional to the GDP. However, note that this sentence is taken from the response letter, not from the Manuscript. In the Manuscript, I simply refer to the total production as being proportional to the total value

of all the goods (line 118). This is the simplest way to define it in the model (which is, of course, simplified).

Second, the Reviewer suggests that “perhaps if GDP was measured differently ... we could be talking about different results or less importance of the discount factor.” As I commented above, I agree that the currency would affect the definition of the discount (and, accordingly, its value). However, it is important to emphasize that the dynamics of the harvest functions and of the provision of the resources would not be affected at all by a change in μ . In the revised Manuscript, I emphasize this issue: “Note that, in both the optimal and the market solutions, the harvest functions, as well as $c(t)$ and $f(t)$, do not depend on μ (only the discount and the prices do).” (Lines 228-229.) This is clear, for example, from the revised Methods section. The methods for finding $c(t)$ and $f(t)$ in both the optimal and the market solutions do not use Eq. 3, and therefore, do not depend on μ (e.g., lines 327-334). The discount rate is calculated (using Eq. 3, in which μ appears) only after $c(t)$ and $f(t)$ are determined (lines 350-352).

As highlighted as well by other reviewers, different readers will have different capabilities to go into the equations and proofs, but the logic of the argument and simulations might be easier to grasp for many economists. Somehow following my previous comment on the usefulness of providing complementary results on the sensitivity, robustness, etc. in the Supplementary Material, I consider that providing a kind of tool/application for modelling simulations (e.g. for a chooser to modify at least a few parameters; it could be with a limited choice of them, based on the provided example, etc.) to very intuitively observe the effects, and notably on harvest of natural resources on the discount factor.

The Reviewer suggests that I provide a tool that allows the user to change the parameters of the model and observe the results.

According to this interesting suggestion of the Reviewer, I created a graphical tool that allows the user to examine how Fig. 4 would look like for other parameter values, and to create different scenarios for the actual Δ that corresponds to a given pattern of harvest, Δ_{scenario} (as in Fig. 4A). Specifically, in the scenarios considered in the tool, the non-sustainable harvest function, H_n , is such that the discount rate remains today’s discount rate for the next τ_1 years. Then, H_n decreases gradually over the next τ_2 years. Finally, after $\tau_1 + \tau_2$ years, $H_n = 0$. (Note, however, that the non-sustainable harvest stops earlier if $x = x_1 + x_2$ approaches 0.1 of its initial value.) The user can chose the value of the parameters τ_1 and τ_2 , as well as the parameters g_c, g_f, μ, η and α . The user shall observe that Δ_{scenario} always goes below Δ_{sus} in the long run.

The installation package is attached, as well as a “Readme.pdf” file with instructions and a “demo.pdf” file with a few snapshots of expected outputs. See the screenshot of the tool below, where the yellow line shows Δ_{scenario} and the blue line shows Δ_{sus} .

Screenshot of the graphical tool

As of Fig. 2, note that the simulations that determine the harvest functions and the functions $c(t)$ and $f(t)$ are coded with C/C++ and are heavy and require about a week to run. It could take months per simulation in Matlab or on some online script, not to mention the immense amount of work that it would take to adjust the code. But note that I also included with the submission the C/C++ code that was used for generating Figs. 2, 3 as well as the Matlab code that was used for generating Figs. 4, 5 and the graphical tool.

In turn, in the text, I extended the Methods section to further explain how Figs. 4, 5 as well as the plots in the graphical tool were generated (lines 342-349). I draw the reader's attention to the graphical tool in the Results section (line 234), in the Discussion section (line 273), in the legend of Fig. 4 (line 449-450), and in the Methods section (lines 342 and 383).

Reviewer 2

I sincerely thank the Reviewer for carefully reading the Manuscript once again, for the helpful additional comments, and for the patience in going through the details of the model. Below please find each of the Reviewer's comments (black) followed by my response (blue).

I think there is movement in the right direction when it comes to the theory of dual discounting where I believe that the authors are simply using earlier work by references (35 and 36). This is fine and could be said clearer so that this complex paper becomes a little more readable. Simply starting by saying that you are building on earlier work with dual discounting and they endogenising the growth rates of the manufactured and environmental goods would remove a first hurdle to understand what your paper is about. Then however you must also focus on explaining how you do this. The part about which areas are sustainably/unsust. managed and how the productivities evolve etc is the crucial addition that might make this paper very interesting – but again I think you write in a confusing manner. If this can be clarified we might have an important contribution. I am afraid I cannot judge this yet.

The Reviewer suggests that (1) it would be helpful to explain right from the beginning that I build on earlier work with dual discounting and endogenize the growth rates of the manufactured and environmental goods. Then, the Reviewer mentions that (2) the paper can become an important contribution, but it would still require clarifications about the model. (The Reviewer further specifies in how to clarify the model below.)

Regarding issue (1), I added a sentence to the abstract explaining that “I build on earlier works with dual discounting and endogenize the growth rates of the manufactured and environmental goods.” (Lines 8-9). Also I explain in the revised last paragraph of the Introduction that “we examine how the discount rate is affected by large changes in harvest patterns at the global scale, such as the transition from over-harvesting to harvesting sustainably, which affect the provision of natural resources.” (Lines 63-65.) In addition, I explain this issue in detail right at the beginning of the Model section, in a paragraph that I revised for clarity (lines 72-78): “We begin with describing a well-established framework (32, 35, 36) that specifies how social welfare and the discount rate depend on the provision of the natural resource over time, $f(t)$, and on the consumption of other goods over time, $c(t)$. Next, we specify how harvest at the global scale affects the dynamics of $f(t)$ and $c(t)$ (which would grow exponentially if the harvest functions are fixed). This builds on and generalizes previous studies that considered the dynamics of $f(t)$ and $c(t)$ as exogenous (e.g., growing exponentially irrespective of the harvest) (32, 35, 36). We complete the model by describing how the harvest strategies are determined by the various managers in a competitive market.”

Regarding issue (2), I thank the Reviewer for acknowledging the potential importance of the paper. I made various clarifications in the Model and Results sections as well as the legend of

Fig. 1 to address the issues raised by the Reviewer. Please see my response to the comments below.

Detailed thoughts and reactions

I find the paragraph from lines 30-38 including formula 1 quite confused. You say in the text that you are going to maximize welfare but then the formula has an integral of B. The most common interpretation would be that B is concrete benefits – like consumption goods (minus costs – since you write net benefits). In that case the formula is wrong because you said you wanted to model welfare which would be based on the Utility of benefits ie $U(B)$ or $W(B)$. Another interpretation of your text would be that you use the symbol B to symbolize the utility of consumption. This would be a somewhat unusual way of wrting. Either way formula 1 just confuses.

You write “The rationale behind discounting is that the objective of our society is to maximize welfare rather than net output” but then your formula does just the opposite and maximizes net output..

Formula 2 does the job in the usual way so I think it might be better to use that straight away. Furthermore B is not in your list of variables but – never mind maybe better to just skip (1).

The Reviewer mentions that the interpretation given to Eq. 1 is confusing and inaccurate. I believe that the Reviewer has missed something here, but the text has been a bit confusing as well. The symbol $B(t)$, indeed, characterizes the benefit minus the cost in units of consumption (as suggested by the Reviewer, and I revised line 34 to clarify this). However, note that the integral is not over $B(t)$; rather, it is over $B(t)\exp(-\Delta(t))$, where $\exp(-\Delta(t))$ is the discount factor. Therefore, $B(t)\exp(-\Delta(t))$ is the addition to welfare due to the added benefits and costs (or due to the change in consumption). Furthermore, note that Eq. 1 is not part of the Model. (The Model starts from Eq. 2.) Eq. 1 is used for the introductory purpose of explaining what the discount rate is and to motivate the paper.

The new text 87-102 tries to sort things out but verges on confusion.. There are conceptually different measures such as social and private discount rates. You cant say they are “correlated” – that is a statistical term. These rates cannot be measured and we have no idea if they would be correlated. You could say that they are closely related – but that is conceptual not statistical. Altogether it worries me deeply that the authors seem to think we can somehow go in and measure or discover what the discount rate “actually is”. This is conceptually impossible. It is a theoretical and abstract construct – Look at your own formula (3) which consists of lots of second derivatives of the utility function. How do you imagine you are going to measure this?? This confusion is also reflected in some of their answers to my questions about whether they believe in normative or descriptive approach – and they say both... The only way you can “measure” the discount rate is if you decide it is by definition equal to the (riskless) rate of return on capital. The latter can be measured – but then with that approach you don’t need articles like

the one we are discussing nor is their much point in elaborating on equations like (3) since it is anyway an observable rate of return that we are talking about.

To try to be a bit positive and constructive I think the part of the text from (2) to (3) or from lines 103 to 134 is correct and simply restates what has been published earlier as (35) and (36). As long as the formula (3) says the same as these earlier references then we can feel safe that the text is correct and it might be easiest to stay closer to these texts and as mentioned skip some of the text I criticized above.

This comment of the Reviewer refers to the paragraph in lines 79-94 (in the revised version) and comprises the following points:

- (1) The word “correlated” was inappropriate.
- (2) The Reviewer is worried that I think that “we can somehow go in and measure or discover what the discount rate actually is.” Also, the Reviewer mentions that one could measure the riskless rate of return on capital.
- (3) This leads the Reviewer to the next question: If we were to decide that the discount is given by the riskless rate of return on capital, then it is “anyway an observable rate of return that we are talking about”, so why do we need to analyze its value in the first place?
- (4) The Reviewer acknowledges that the part of the model that relates the welfare and the goods to the discount rate (lines 95-126 in the revised version) is correct, and that the “problem” only appeared in the paragraph that provides the background (lines 81-96).

Regarding point (1): I agree with the Reviewer that the word “correlated” was a poor choice and I changed it to “closely-related”, as suggested by the Reviewer (line 83).

Regarding point (2): I agree that the discount cannot be measured “directly” (only indirectly via, e.g., estimates of the rate of return on capital). In that context, let me emphasize that I do not discuss the empirical measures of the discount in the paragraph mentioned above (lines 79-94), nor do I argue there that it could be measured. After I deleted the word “correlated”, the paragraph simply discusses the various approaches to defining the discount. Specifically, in lines 79-83, I simply mention the various alternative ways to define or consider the discount. Then, in lines 83-84, I simply mention that, under the assumption of a perfectly competitive market, these measures become equal (of course, this does not hold in real markets, but this is only to be more specific about what “closely-related” means in this context). This sentence is backed-up, for example, by Groom *et al.* (Ref. 15), that writes that “in the perfectly competitive paradigm, all rates are equal and hence it does not matter which rate: i , r , or δ , is used ...” (Page 452, lines 3 and 4 from the top.) Then, in lines 85-90, I give a more detailed definition of the social rate of time preference, and finally, in lines 90-94, I briefly introduce the mechanisms that could affect the social rate of time preference. In light of this, I believe that the confusion might have arisen from the use of the word “measure” to describe the alternative definitions of the discount rate. I did not mean to imply that these are empirical

measures, but I agree that the use of the word “measure” was inappropriate and could be confusing. I revised the sentence, and I simply write that “there is no consensus on which quantity policymakers should consider as the social rate of discount, but the most widely recognized candidates are the social rate of time preference, the social and private rates of return to investment, or some combination of the three (15, 37).” (Lines 80-83.) This does not imply that anyone could measure these quantities. In addition, I went over the rest of the Manuscript to make sure that any phrasing that could be confusing in this context is revised accordingly.

Regarding point (3): I focus in this study on the social rate of time preference, as I explain in line 85. But regardless, using theory to study the future rate of return on capital is still important. The reason is simply that to determine climate policy, for example, we need to estimate/predict what that rate would be hundreds of years from now, whereas the financial markets only give us a glance of the next 30-40 years. As Groom *et al.* mention: “The difficulty in the long run is the absence of financial assets whose maturity extends to the horizon associated with the new types of projects and policies that the government is faced with, e.g. global warming. Government bonds, for example, do not extend beyond 40 years in general. In the absence of a measure of the long run discount rate determined by financial markets, Gollier (2002a, b, 2004b) turns to economic theory to provide some answers.” (Last paragraph on page 465.) (It is worth mentioning that Gollier examined the effect of uncertainty in technological development.)

The interesting and innovative part starts afterwards in deciding how much c and f will be produced and that is where sustainable and unsustainable harvesting comes in. I do however have a hard time trying to understand the dynamics described for which areas are managed sustainably or not. It seems this could be simple enough if you assumed that open access leads to non-sustainable use and a single owner leads to sustainable use but the text is very convoluted and poorly written – I don’t know but I think they do NOT make this simple and natural assumption. However I cannot figure out what they do assume. This would seem to be one of the central points so therefore I am lost how they get their results.

I think the authors need to make a greater effort to clearly state how the harvesting methods and the areas in which they are practiced evolve. Do they depend on ownership patterns and degree of open access and exactly how? Do they depend on the discount rate perhaps that would seem to be interesting...

[Note that the comment of the Reviewer continues, but I split it and I give the rest of the comment after my response to this part.]

In this main comment, the Reviewer asks that I clarify the part of the Model and the Results sections that explain how the harvest functions are determined by the managers. Here I address the comment and, by doing so, I also explain how I revised the Manuscript accordingly.

(Note that the citations from the Manuscript that I give below are of sentences that I added or revised.)

How the harvest functions are determined – optimal solution

First of all, note that I describe two types of solutions: (1) the optimal solution that maximizes social welfare, and (2) the market solution that emerges in a competitive market. The definition of the optimal solution is straight-forward. The social planner decides how much to harvest sustainably and non-sustainably over time such that social welfare is maximized. There is no distinction between the way that the managed and the shared regions are being managed in that case because the social planner determines the harvest functions for the entire system. I explain that “The optimal solution is found via the maximization of the social welfare (Eq. 2) subject to the constraints given in Eqs. 4-7 (Methods).” (Lines 175-177.)

In turn, the results show that, following the optimal solution, the harvest is mostly sustainable, although some degree of non-sustainable harvest is present. In the revised Manuscript, I revised the paragraph that explains the results of the optimal solution throughout (lines 200-212). Among other changes, I explain that “the optimal solution comprises non-sustainable harvest... because an increase in f at a given time has a greater effect on welfare than the same increase at a later time; the lower the discount rate, the lower the rate of non-sustainable harvest.” (Lines 209-212.)

How the harvest functions are determined by the managers – competitive market solution

In the market solution, the idea is to consider or “simulate” the harvest dynamics that emerge in some simple model of a competitive market. First of all, I note that: “For the theorem, we consider a general case in which the market solution implies a higher rate of non-sustainable harvest.” (Lines 179-180.) Namely, the theorem applies generally to cases where the market dictates higher levels of non-sustainable harvest (e.g., see lines 252-253 in the theorem). The reason why I go further and describe a more detailed model is that I would like to demonstrate a simulation of some special case, which would complement and visualize the general theoretical results given in the theorem. Specifically, I explain that “for our simulations, we specify here a special case in which we distinguish between managed regions and shared regions that are prone to higher levels of non-sustainable harvest.” (Lines 180-182.)

A possible modeling approach could be, as suggested by the Reviewer, to assume that the management in the managed regions is sustainable and the management in the shared regions is non-sustainable. This would yield almost the same results as those presented in the Manuscript. However, I felt that such a model would be superficial for the following reasons. First, even the optimal solution exhibits some degree of non-sustainable harvest. The idea is that, without externalities (if there are no shared regions), the market solution will be identical to the optimal solution. Second, I expect that the management would depend on the welfare.

After all, why are we examining the discount if the discount does not affect the management? The relationship between the harvest and the discount is double-sided, where higher discount rates are expected to result in higher rates of non-sustainable harvest. Third, if I am so confident that the market would dictate these solutions, why would I “assume” them? Why not letting these solutions emerge naturally as the results?

Therefore, in the Model, I simply assume a perfectly competitive market with perfect forecast, such that, without externalities, the market solution and the optimal solution coincide. See, for example, Xepapadeas A. (2005) (Ref. 12), explaining that “It is known that in the absence of externalities there is an equivalence between the outcome of the social planner’s problem and the outcome of the competitive equilibrium with perfect foresight” (end of page 17 and beginning of page 18 there). I also emphasize in the revised Manuscript that “there are no externalities if there are no shared regions ($x_2 = 0$).” (Lines 190-191.) This guarantees that “without externalities ($x_2 = 0$) ... the market solution coincides with the optimal solution (12, 38, 39).” (Lines 193-195.)

In turn, to better explain how managers chose the harvest strategy in the shared regions, I explain in the revised Manuscript that “we assume that the number of managers in each shared region is very large, such that the non-sustainable harvest by a given manager negligibly affects the degradation level in that region. Therefore, each manager is incentivized to harvest non-sustainably as he/she ignores the effects of her/his own actions on the dynamics of x_2 . These considerations allow us to find the market solution, as described in Methods.” (Lines 197-199.)

The results show almost exactly what was suggested by the Reviewer. The shared regions are being harvested only non-sustainably, whereas the managed regions are being harvested almost only sustainably. The difference, however, is that these are now results and not model assumptions. I revised the second paragraph of the Results section, where the market solution is described, and I begin by explaining that “following the market solution in the case where some regions are shared, the rate of non-sustainable harvest is higher than the socially-optimal rate (over-harvesting, Fig. 2C,D). Specifically, the managers in the shared regions harvest non-sustainably, whereas the harvest in the managed regions is primarily sustainable.” (Lines 213-216.)

Going back to the comment, the Reviewer asks (1) how the harvest functions depend on the degree of open access and (2) whether the harvest functions depend on the discount rate. The answer to (1) is that the (initial) portion of shared regions is given by x_2 at time 0. The regions do not change ownership, but at some point, the shared regions become fully degraded. Therefore, a higher $x_2(0)$ implies that higher levels of over-harvesting would occur. I explain that: “Eventually, however, at time $t = t_1$ (Fig. 2), the shared regions become degraded ($x_2 = 0$) and the total rate of non-sustainable harvest, H_n , declines.” (Lines 218-219.) I also explain that the decline in the discount rate is greater if more regions are shared (lines 222-223).

In turn, the answer to (2) is, yes. For example, consider the optimal solution. Although the harvest functions do not depend directly on the discount rate (Eq. 3 is not used for finding the optimal and market solutions), the managers would be more willing to harvest non-sustainably if the marginal effect of the resource on social welfare would be lower in the future (namely, if the discount rate is higher). Specifically, the rate of non-sustainable harvest increases with g_c and g_f . I explain that “the optimal solution comprises non-sustainable harvest... because an increase in f at a given time has a greater effect on welfare than the same increase at a later time; the lower the discount rate, the lower the rate of non-sustainable harvest.” (Lines 209-212.)

In addition to the revisions mentioned above, I extended and reorganized the Methods section, which describes the more technical aspects of what I did to generate the results in each figure. Furthermore, I made several additional changes for clarity in the Model and Results sections and in the legend of Fig. 1, which I specify below as I refer to the specific sub-comments of the Reviewer.

Fig 1 is visually nice but I am afraid it does not help me understand... I find the description is verbacious and unclear. I want to know what H means....

The Reviewer comments that the legend of Fig. 1 is not sufficiently clear and that I need to clarify what H would mean according to the figure. I revised the legend of Fig. 1 accordingly. I explain that H_n is the total dark gray area (the area that becomes degraded starting next year), and that H_s is the total green area (the area that is harvested but does not become degraded). I also added a sentence explaining that “the total area under harvest, H , is given by the green and the light gray areas combined, $H = H_n + H_s$.” Also, I added an explanation that “The variables x_1 and x_2 (Eqs. 5, 6) characterize the total non-degraded areas (blue, green and light gray) in the managed and in the shared regions, respectively.”

In lines 166-167 it sounds like the H variables are areas managed sustainably or not sustainably. In lines 151-152 it sounds like the H variables are a measure of how much the productivity is lowered in one of these areas.

The Reviewer comments that the meaning of H_n following lines 166-167 (lines 158-159 in the revised version) might seem different from its meaning following lines 151-152 (lines 141-143 in the revised version). Note that the first interpretation of the Reviewer, which follows from lines 158-159, is the correct one. $H(t)$ is the total area under harvest in year t , while $H_n(t)$ is the area that becomes degraded during year t (and remain degraded afterward). This is also explained in lines 138-140: “we assume that a given portion of the global ecosystem, $H(t)$, is being harvested in year t , while some portion of the ecosystem, $H_n(t)$, becomes degraded during that year, and cannot be used for harvest thereafter (Fig. 1).”

I revised the sentence in the example in lines 141-143 to clarify that the meaning of H_n there is the same: “For example, $H_n(t)$ may characterize the portion of the global fish or timber stock that becomes unavailable due to the collapse of fisheries or the irreversible degradation of forests worldwide in year t (40).” Furthermore, note that the H_n does not have to represent an “area”; rather, it may represent more generally the portion of the system that becomes degraded in a given year. For example, it could be that 2% of the ecosystem services are lost in a given year (i.e., $H_n = 0.02$). This is what I demonstrate in another example in lines 143-145, which I also revised for clarity.

Later it sounds like x_1 and x_2 are the areas. But it seems that open access areas can be sustainably managed... which is highly confusing.

The Reviewer inquires whether x_1 and x_2 represent the areas, and comments that it is confusing that open access areas can be managed sustainably. In the Manuscript, x_1 represents the total non-degraded areas in the managed regions, and x_2 represents the total non-degraded areas in the open-access regions. As I explain above, the model allow the managers in the open-access areas to harvest sustainably, but they have no incentives to do so, and therefore, they do not use this option. I explain in the revised Manuscript that “we assume that the number of managers in each shared region is very large, such that the non-sustainable harvest by a given manager negligibly affects the degradation level in that region. Therefore, each manager is incentivized to harvest non-sustainably as he/she ignores the effects of her/his own actions on the dynamics of x_2 .” (Lines 195-198.) I also write that “the managers in the shared regions harvest non-sustainably, whereas the harvest in the managed regions is primarily sustainable.” (Lines 214-216.)

Several times I find assumptions strange. There is some technical progress beta which seems to apply equally to the productivity of degraded and sustainably managed areas.

The Reviewer comments that the increase in the yield of a given area under harvest, g_f , seems to apply equally to areas that are managed sustainably and to degraded areas. Note, however, that the degraded areas cannot be harvested, and therefore, the increase in β , g_f , does not affect these areas. To clarify this issue, I explain that “... some portion of the ecosystem, $H_n(t)$, becomes degraded during that year, and cannot be used for harvest thereafter” (Lines 139-140.) (Also note that x_1 and x_2 are the non-degraded areas, and that harvest is constrained by Eq. 6, which implies that only these areas can be harvested.) I believe that, perhaps, the Reviewer intended to comment that, in reality, there may be a different rate for growth in the yield of sustainable and of non-sustainable harvest. I agree that I could use some g_f^s for sustainable harvest and g_f^n for non-sustainable harvest, but naturally, in the kind of model that I propose, simplifying assumption are necessary.

I think a good and natural way to develop your model would have been to describe the logic of how the areas sustainably and non-sustainably managed develop (as a function of open access). This should be enough.

The Reviewer suggests that a good strategy to describe the model would be to describe the dynamics of the harvest functions and how they depend on the open-access areas. I believe that my response to the comments above explains this issue as well. In the third part of the Model section, I describe the incentives of the managers in the different regions. In the Results section, I describe the resulting dynamics of the harvest functions. Specifically, I explain that “the managers in the shared regions harvest non-sustainably, whereas the harvest in the managed regions is primarily sustainable. The total area under (non-sustainable) harvest in the shared regions increases over time, and consequently, $f(t)$ continues to increase over an extended period of time, which postpones the decline in the discount rate and in the cumulative discount. Eventually, however, at time $t = t_1$ (Fig. 2), the shared regions become degraded ($x_2 = 0$) and the total rate of non-sustainable harvest, H_n , declines...” (Lines 214-219.)

Normally the welfare cost would be a fall in f . I am confused that you introduce an additional cost term c in equation 7 and don't quite understand how this is to be motivated. Is this intended to introduce a varied degree of substitutability between c and f ? (there would be other ways to do that).

The Reviewer asks why the cost term in Eq. 7 is needed, because the cost of non-sustainable harvest would be the future reduction in f . The answer is that the cost term in Eq. 7 represents the direct cost of performing the harvest. Consider, for example, the harvest of fish. In most bioeconomic models, the objective is to maximize the net present value, which incorporates the net benefits from selling the fish and the net costs of operating the vessels. In other words, there are two costs to harvest: the direct cost of operating the vessels (labor, fuel, vessel maintenance, etc.), and the “indirect” cost due to having fewer fish to harvest in the days that follow. The cost term in Eq. 7 is analogous to the direct cost of operating the vessels.

In turn, this direct cost term plays the following role in the dynamics of c and f . Think about the early stages of the economy, where c is still low. Only a small portion of the ocean is harvested and degradation of fisheries is not an issue. However, there are still regions of the ocean that are not under harvest, simply because it would be expensive to build and operate so many ships. In other words, one has to decrease c to harvest more f . Specifically, in the model, without the cost terms in Eq. 7, there would be no reason for the managers in the shared regions not to over-harvest these regions right away. In turn, as the economy grows, c increases and the direct cost becomes less and less important. Instead, a greater portion of the ocean is harvested, and the over-harvesting becomes the main issue and the main impediment to further increasing the harvest.

In lines 168-169, just before Eq. 7, I explain that “we assume that harvest comes with a direct cost as more labor and resources are directed toward harvesting.” Also, I revised the first paragraph of the Results section to better describe how the direct cost affects the dynamics of harvesting: “Our numerical results show that, following the optimal solution (Fig. 2A,B), two phases emerge along the time axis. In the first phase ($t < t_0$), $c(t)$ is initially small, and the harvest rates are limited due to the direct cost of harvesting (Eq. 7). Over time, as $c(t)$ increases, the direct cost plays a less significant role, and the society increases the harvest rates.” (Lines 200-203.)

Any way I am not sure quite how much of this is retained in your part 3 when you start to compare the “optimal” and “market”, solutions. Here I think you should be careful in using the word “market” since it has many ideological connotations and you describe one special case here. If you had no externalities and well defined ownership everywhere (ie no open access) would the two then be identical?

The Reviewer suggests that I should be careful with the word “market” and asks whether the “market” and the “optimal” solutions will be the same in the case where there are no externalities and ownership is well-defined everywhere (i.e., no open-access regions). First, note that the Reviewer is correct that, if there are no open-access regions ($x_2 = 0$), then there are no externalities and ownership is defined everywhere. In turn, the answer to the Reviewer’s question is, yes. I explain that “without externalities ($x_2 = 0$), ... the market solution coincides with the optimal solution” (Lines 193-195.)

On page 6 you start describing “results” and refer to figures – but these are not the theoretical results as far as I can see but some very detailed simulation. Such simulations depend on lots of parameter values and I am not sure when and how these were introduced or discussed? As far as I can see, they are just mentioned at the end of the figure captions but not motivated or discussed. the reader is left worrying if the results would be the same with our parameter values. The paper does seem to claim some theoretical results too – for general functional forms but I am not sure of which part of the results and conclusions are based on the general theory results and which part comes from a simulation where I would need to know whether or not these have any general validity...

The Reviewer raises the following issues regarding the numerical results: (1) The parameter values are given in the figure legends but are not discussed; (2) “the reader is left worrying if the results would be the same with our parameter values”; and (3) the Results section start with a description of the numerical results, while the theorem is presented afterward, but it is not clear where I switch from describing the specific simulation results to describing the theorem’s results.

Regarding issue (1), the Reviewer is correct that the discussion about the parameter values was scattered and incomplete in the previous submission. Accordingly, I added a paragraph to the

Methods section (lines 356-378) where I discuss, for each parameter, what would be its reasonable choices and what kind of system the various choices could represent, including all the necessary citations from the literature. In turn, note that the parameters used in the figures are within those ranges. Also note that, clearly, the parameter values depend on the specific system and there are wide ranges from which it would be reasonable to choose the parameters.

Regarding issue (2), note that some of the numerical results are specific and some of them are more general. (Namely, although the details may be quantitatively different, some general properties of the dynamics are more robust.) Throughout the text, I emphasize those results that are more general. First, as pointed out by the Reviewer, the main result of the paper is backed-up by the theorem, which is much more general and does not depend on specific choices of parameters. Second, in the Results section, I explain the mechanisms underlying some of the more general results. For example, harvest in the shared regions is non-sustainable because the managers have no incentives to harvest sustainably there; over-harvesting occurs in the market solution due to the externalities in the shared regions; etc.

At the same time, I emphasize that the main result is robust, even though some of the numerical results are not. Specifically, the transition from over-harvesting to sustainable management in Fig. 2 (at time t_1) is very sharp, but this is only because of the unrealistic assumption that there are only two types of regions. In reality, I would expect to see a more gradual transition. Therefore, I show that the main results are more robust and general, and they hold in cases where the non-sustainable harvest decreases gradually. Specifically, in Theorem 1, there is no specific assumption about the externalities of the managers and no assumption about the specific shape of the harvest functions. The Theorem shows that, regardless of the duration of the transition to sustainable harvest, the cumulative discount must ultimately decline to a lower level than it would have had if managers follow optimal or sustainable harvest from the beginning: "...for any Δ that emerges if non-sustainable harvest occurs ($H_n(t) > 0$) between times t_0 and t_1 , there exists $t_c > t_1$ such that $\Delta(t_c) \leq \Delta_{sus}(t_c)$." (Lines 252-253.) I also explain that "a more gradual transition to sustainable harvest may result in a more gradual decline in Δ , but the ultimate magnitude of the decline must exceed that of the decline in Δ that occurred formerly due to the over-harvesting (Figs. 3, 4A)." (Lines 234-236.) In turn, I demonstrate this result in Figs. 3 and 4A. (This might not be necessary since it is already being shown in the theorem, but further visualization is helpful for some readers.) Specifically, Fig. 3 demonstrates that a more gradual transition to sustainable harvest will not change the main result and the cumulative discount will ultimately decline. Also, the scenarios in Fig. 4A demonstrate the same idea: In all of these scenarios, Δ that corresponds to the scenario always goes below Δ_{sus} in the long run (and this must be the case, according to Theorem 1). Note that, for the scenarios in Fig. 4A, I assume that the dynamics are given by Eqs. 4-7, but I chose the harvest functions "by hand" instead of simulating the specific market that is considered in Fig. 2 (e.g., lines 342-349).

Regarding issue (3), note that the first two paragraphs of the Results section describe the numerical results, while the last two paragraphs describe the more general results (Theorem 1). I revised the Manuscript to make this clear. I begin the Results section with “Our numerical results show that...” (line 200). Then, in the third paragraph, I clarify that I switch to describing the theoretical results of the theorem: “More generally, Theorem 1 below shows that the main result is robust and does not depend on specific assumptions and parameters. Specifically, the theorem shows that over-harvesting may increase Δ in the short-run, but ultimately, Δ would return to a lower value than it would have had if managers used optimal harvesting or only sustainable harvesting (see proof in Appendix D and demonstration in Figs. 3, 4 and Supporting Graphical Tool).” (Lines 230-234.)

As for the figures with detailed simulations, I am unsure .. Could you somehow, make us believe that these particular simulations based on these particular numerical assumptions are at all plausible. What would have happened with other values?

The Reviewer asks whether the results demonstrated in Fig. 2 are plausible and whether they would hold for other parameter values. I believe that my response to the previous comment addresses this comment as well. Some of the numerical results are, indeed, specific. But some results are more general, as I show in the theorem and explain the text. Specifically, the theorem shows that the main result (the decline in the discount is inevitable) is general. See my response to issues (1) and (2) in the previous comment.

Reviewers' comments:

Reviewer #1 (Remarks to the Author):

In my view the author has addressed quite satisfactorily my concerns and I do not feel capable of providing further specific review. Perhaps there was some misplacement or error within the system, but I did not find though in the files what is described as:

"The installation package is attached, as well as a "Readme.pdf" file with instructions and a "demo.pdf" file with a few snapshots of expected outputs."

Reviewer #2 (Remarks to the Author):

I believe this paper makes one interesting point by emphasizing that not only do discount rates affect economic production (and for instance nature resource extraction) decisions but also the opposite applies. Resource extraction decisions can affect the sustainability of resource sectors and thus the long run growth rate - and thereby, in turn, the optimal, long run discount rate.

Although this point might not be entirely new to economic thinking, it is unusual to put any emphasis on it and I think it is a point that deserves to be made and emphasized.

Having said this, I have found work with this manuscript very heavy going. The text is hard to read and although my questions, comments and suggestions have actually led to quite a lot of improvement, I feel a lot more remains to be done to give this paper the clarity one would expect from a Nature publication.

In the last round the reply letter was 16 pages long and although many errors have been put right and, as mentioned, real progress made, I still feel nervous about accepting the paper.

In one of the newly written sections, which contains corrections of earlier language including "correlation" between the measures below which the authors agreed was wrong, the authors now write:

There is no consensus on which quantity policymakers

should consider as the social rate of discount, but the most widely recognized candidates are the social

rate of time preference, the social and private rates of return to investment, or some combination of the

three (15, 37). Note, however, that these three quantities are closely-related, and, in a perfectly

competitive market, all three become equal and reflect the marginal productivity of capital (15, 38, 39).

This is STILL confusing at best if not just wrong.

the social rate of time preference is applied to the discounting of UTILITY. It is a component of the social rate of discount (for goods). It is not a candidate nor is it "closely related" to the social rate of discount for goods. It is a completely separate ethical component concerning how much we value the utility or welfare of future people - not how we discount the value of a certain cost or benefit to them.

Unfortunately I feel there are too many instances of such confusing statements still in the text.

Reviewer 2

I thank the Reviewer for reading the Manuscript once again and for the additional comments. Below please find each of the Reviewer's comments (black) followed by my response (blue).

I believe this paper makes one interesting point by emphasizing that not only do discount rates affect economic production (and for instance nature resource extraction) decisions but also the opposite applies. Resource extraction decisions can affect the sustainability of resource sectors and thus the long run growth rate - and thereby, in turn, the optimal, long run discount rate. Although this point might not be entirely new to economic thinking, it is unusual to put any emphasis on it and I think it is a point that deserves to be made and emphasized.

Having said this, I have found work with this manuscript very heavy going. The text is hard to read and although my questions, comments and suggestions have actually led to quite a lot of improvement, I feel a lot more remains to be done to give this paper the clarity one would expect from a Nature publication. In the last round the reply letter was 16 pages long and although many errors have been put right and, as mentioned, real progress made, I still feel nervous about accepting the paper.

I thank the reviewer for acknowledging again the importance of the paper, as well as mentioning that there was "quite a lot of improvement" and that "real progress made" due to the revisions. At the same time, however, the Reviewer mentions that more needs to be done and that he/she still "feels nervous about accepting the paper" and raises the comment that follows. Below, following the comment, I explain how I addressed it.

In one of the newly written sections, which contains corrections of earlier language including "correlation" between the measures below which the authors agreed was wrong, the authors now write: "There is no consensus on which quantity policymakers should consider as the social rate of discount, but the most widely recognized candidates are the social rate of time preference, the social and private rates of return to investment, or some combination of the three (15, 37). Note, however, that these three quantities are closely-related, and, in a perfectly competitive market, all three become equal and reflect the marginal productivity of capital (15, 38, 39)."

This is STILL confusing at best if not just wrong. The social rate of time preference is applied to the discounting of UTILITY. It is a component of the social rate of discount (for goods). It is not a candidate nor is it "closely related" to the social rate of discount for goods. It is a completely separate ethical component concerning how much we value the utility or welfare of future people - not how we discount the value of a certain cost or benefit to them.

Unfortunately I feel there are too many instances of such confusing statements still in the text.

The Reviewer argues in the comment that the “social rate of time preference” is applied to the discounting of utility. According to the Reviewer, the “social rate of time preference” is only a component of the social rate of discount (and not a candidate to be the social rate of discount).

However, the Reviewer has clearly confused between the “social rate of time preference” and the “pure rate of time preference”. The pure rate of time preference (also known as the “utility discount rate” or “pure impatience”) is not considered a candidate for being the social discount rate and, as the Reviewer said, it is only a component of that discount. In turn, I correctly used the term “social rate of time preference” (also known as the consumption rate of discount), which is a standard term used for the social rate of discount in studies focusing on consumption. (Although many papers simply use the term social rate of discount without further discussion.)

Here are a few citations from the well-known review by Ben Groom *et al.* (Ref. 15) that clearly show that my use of terminology is correct (note that their notations, δ and ρ , are similar to mine): (1) “economists and others have argued at length over which of several potential discount rates should be used... Several candidates exist, the most widely recognised of which are the social rate of return on investment (r) and the rate at which society values consumption at different points of time, the Social Rate of Time Preference (δ)” (page 446 lines 6-12); (2) “The term δ is defined as the social rate of time preference, which reflects the change in relative value that society places on units of consumption at adjacent periods of time.” (bottom of page 448); (3) “The Ramsey rule in equation (3) shows why it is valid to consider the social rate of time preference, δ , and the rate of return on capital, hereafter r , as candidates for the socially efficient discount rate for projects or policies whose costs and benefits are measured in consumption equivalents” (page 451 last paragraph); (4) “in the perfectly competitive paradigm, all rates are equal and hence it does not matter which rate: i , r , or δ , is used ...” (page 452 lines 3-4); (5) “In a deterministic world we noted that there are two underlying characteristics of individual preferences’ which determine the social rate of time preference, δ ; (i) pure impatience, ρ , and (ii) the desire to smooth growing wealth over time, θ ” (page 465 section 4.3).

I believe that it is very clear from the Manuscript that I do not consider the pure time preference as the social rate of discount, and I also revised the Manuscript to further clarify this. First, I write explicitly that the “social rate of time preference” should not be confused with the “pure rate of time preference.” Then, I give a clear definition of the social rate of time preference that makes it very clear that what I consider is simply the social rate of discount in the “usual” sense that it is used in the related theoretical papers (lines 89-92). Also, I emphasize that the mechanism for discounting that I focus on is that “the marginal contribution to welfare of a given unit of consumed goods will be lower if our society becomes wealthier and consumes more goods in the future.” (Lines 93-95.) And, I explain that the pure time preference is only one mechanism (and therefore, one component) for discounting (lines 95-96). Note also that, in the paragraph that follows, I define ρ as the rate of pure time preference and δ as the social rate of discount (or the social rate of time preference), and in Eq. 3 the term ρ is only one

component in the equation for δ . In light of all that, I believe that the discussion is very clear, particularly in the revised Manuscript.

Finally, the Reviewer mentions that there are other instances of “such confusing statements” in the text. Unfortunately, this comment is not particularly constructive as it does not provide any information about what these statements that confuse the Reviewer are. Not to mention that I feel like this comment is not particularly fair as a comment that comes after my very serious effort in the last two revisions and addressing all the comments of the Reviewers. Nevertheless, I can see how having doubts about what I consider to be the discount rate could be very confusing, and many parts of the text would not make sense if I were considering the pure time preference. Therefore, I hope that my revision that followed the comment above has also removed the confusion in other places throughout the text. In addition, I also went through the text once more to make sure that my use of jargon is correct and clearly explained. I thank the Reviewer again for the very helpful comments on the various versions of the Manuscript.

Reviewer #2 (Remarks to the Author):

I still have mixed feelings about your manuscript.

I like the idea that you can see a discount rate as endogenous but instead of explaining it well as a complex and original idea you dive in and talk about it as if it were affecting behavior in a way that sometimes throws me off and I worry will confuse other readers too – more about this below. I also find numerous instances of text that I find hard to read.

Your handling of the various concepts such as optimal discount rate, market rates, “social rate of time preference” and the “pure rate of time preference” has in various versions been sometimes wrong and sometimes confusing. In your rebuttal of my last comments you point out that the last two concepts are different and you are right I confused them but I think that this is partly a sign of how confusing your text is. It is true that “social rate of time preference” is used for “socially optimal discount rate”. However in a context where we are trying to avoid confusion between discounting of goods and of utility I would have chosen a different wording. I am after all a discounting enthusiast and expert and if I misread your text there is a risk others will do the same. So that particular paragraph can be defended and is “not wrong” – but it is still not well written either.

I raise a few other issues I found when rereading your manuscript but I give these as a few examples, I am not sure they are a complete list.

I see a problem in the comment:

One major debate followed the publication of the Stern report, which used a discount rate that is smaller than the one dictated by the market, and consequently, argued for radical emission cuts

Firstly The market does not “dictate” anything. There are ideologies and simplified theories that under certain conditions suggest the use of market rates – but these are theories or ideologies not market dictates. Secondly the market rate has actually for a long time been about 0 so the statement is not true in any simple sense.

The authors say:

In this paper, we examine how the discount rate is affected by large changes in harvest patterns at the global scale, such as the transition from over-harvesting to harvesting sustainably, which affect the provision of natural resources.

I think the whole novelty of your article hinges on the idea of endogenizing the discount rate. Before you get into details like “We show that over-harvesting temporarily keeps the discount rate higher, but is followed by a period of lower discount rates when society adjusts to sustainable harvesting.” You need to discuss the fundamental and complex idea of a changing discount rate.

I would suggest you make this clearer: roughly like this

We believe that the differences between sustainable and unsustainable management of ecosystems will be so large in the future that the whole economy will be affected. Since discount rates depend on growth, this means that the discount rate itself could be affected. This is problematic since the discount rate is supposed to be a (constant) parameter in an optimization that itself decides future production and thus growth. Then you can continue with your text “In this paper, we examine how the discount rate is affected by large changes in harvest patterns at the global scale, such as the transition from over-harvesting to harvesting sustainably, which affect the provision of natural resources.”

Ideally you should however discuss the mathematical and philosophical problems of having an endogenous discount rate. If the discount rate is used to decide investment and growth strategies and these themselves determine the discount rate – then can we be sure there is a unique optimum? There might be different development and growth paths – each with their own discount rate path – and they might all look optimal on their own but how do we choose between them if our choice criterion involves a discount rate that is variable.

In your Discussion, you talk about the discount rate going up and down as if it were something that could be observed and might directly affect the behavior of agents. I find this confusing. Decentralized Agents will be using market rates – their cost of borrowing capital. I think of the optimal discount rate as something to be discussed by governments and researchers for instance to put into DICE and other IAMs. Thus I think of rising and falling discount rates in a figurative sense but I worry you think of it as something that can be a) observed and b) has a direct effect in the behavior of agents.

Finally as a separate point I still find it hard to see what drives the harvest decisions (various H) in your model. Is each manager making their own decision based – presumably on market rates of interest. What decides the entry and exit of different types of manager- and thus their numbers?

Reviewer 2

I thank the Reviewer for the additional comments on the Manuscript. Below please find each of the Reviewer's comments (black) followed by my response (blue).

I still have mixed feelings about your manuscript. I like the idea that you can see a discount rate as endogenous but instead of explaining it well as a complex and original idea you dive in and talk about it as if it were affecting behavior in a way that sometimes throws me off and I worry will confuse other readers too – more about this below. I also find numerous instances of text that I find hard to read.

The Reviewer mentions again that he/she liked the main idea of the paper, but what still bothers her/him is that I write about the change in the discount in a way that could be interpreted as if it were affecting behavior. First of all, I would like to emphasize that I do not argue at any point that the change in the discount affects behavior. I do not even discuss this issue. I simply present the change in the discount as something that emerges naturally as a consequence of over-harvesting. I do not argue or assume how it will affect the behavior of individuals. I do not even know if it is going to affect the decisions of climate policymakers. I can only suggest that it should be incorporated in the relevant policies and hope that it will if the scientific community will read my paper. When I emphasize the merit of the paper in the Introduction, I simply write that “Revealing harvest-induced changes in the discount rate will provide policymakers with better evaluations of long-term benefits and costs, thereby enabling them to improve long-term environmental policies.”

At the same time, I understand from the Reviewer's comments below that the wording that I used has been confusing in several places in the text, and therefore, I revised the Manuscript accordingly. I believe that the following revisions that followed the additional comments of the Reviewer have clarified this point and removed the remaining confusions and doubts.

Your handling of the various concepts such as optimal discount rate, market rates, “social rate of time preference” and the “pure rate of time preference” has in various versions been sometimes wrong and sometimes confusing. In your rebuttal of my last comments you point out that the last two concepts are different and you are right I confused them but I think that this is partly a sign of how confusing your text is. It is true that “social rate of time preference” is used for “socially optimal discount rate”. However in a context where we are trying to avoid confusion between discounting of goods and of utility I would have chosen a different wording. I am after all a discounting enthusiast and expert and if I misread your text there is a risk others will do the same. So that particular paragraph can be defended and is “not wrong” – but it is still not well written either.

The Reviewer refers to her/his comment in the previous revision. He/she acknowledges that the terminology that I used there is correct. But to avoid future confusion by other readers, the Reviewer suggests that I simply avoid the terms that could confuse. Accordingly, I revised the two relevant sentences once again and I do not use the term “social rate of discount” that caused the confusion.

I raise a few other issues I found when rereading your manuscript but I give these as a few examples, I am not sure they are a complete list.

I see a problem in the comment: One major debate followed the publication of the Stern report, which used a discount rate that is smaller than the one dictated by the market, and consequently, argued for radical emission cuts. Firstly The market does not “dictate” anything. There are ideologies and simplified theories that under certain conditions suggest the use of market rates – but these are theories or ideologies not market dictates. Secondly the market rate has actually for a long time been about 0 so the statement is not true in any simple sense.

The Reviewer mentions that the term “dictated by the market” that refers to the discount rate might sound like I incorrectly argue that there is a simple way to determine the discount rate based on the market. I revised the sentence and simply write that Stern used a discount rate that is smaller than those used in previous major studies (line 48).

The authors say: In this paper, we examine how the discount rate is affected by large changes in harvest patterns at the global scale, such as the transition from over-harvesting to harvesting sustainably, which affect the provision of natural resources. I think the whole novelty of your article hinges on the idea of endogenizing the discount rate. Before you get into details like “We show that over-harvesting temporarily keeps the discount rate higher, but is followed by a period of lower discount rates when society adjusts to sustainable harvesting.” You need to discuss the fundamental and complex idea of a changing discount rate. I would suggest you make this clearer: roughly like this. We believe that the differences between sustainable and unsustainable management of ecosystems will be so large in the future that the whole economy will be affected. Since discount rates depend on growth, this means that the discount rate itself could be affected. This is problematic since the discount rate is supposed to be a (constant) parameter in an optimization that itself decides future production and thus growth. Then you can continue with your text “In this paper, we examine how the discount rate is affected by large changes in harvest patterns at the global scale, such as the transition from over-harvesting to harvesting sustainably, which affect the provision of natural resources.”

The Reviewer suggests that I add a particular sentence to clarify that the mechanism for the change in the discount is that the change in the provision of the natural resource will be so large that it will affect the entire economy. I added a sentence that is very similar to the one suggested by the Reviewer (lines 60-63).

Ideally you should however discuss the mathematical and philosophical problems of having an endogenous discount rate. If the discount rate is used to decide investment and growth strategies and these themselves determine the discount rate – then can we be sure there is a unique optimum? There might be different development and growth paths – each with their own discount rate path – and they might all look optimal on their own but how do we choose between them if our choice criterion involves a discount rate that is variable.

The Reviewer refers here to the fact that the planner uses the discount to determine policy, but in turn, the policy itself affects the discount. Accordingly, the Reviewer asks for clarification as to whether there is a unique optimum or whether there are multiple discount pathways such that the solution seems to be optimal in each.

The answer is that in my model, there exists a unique optimum, and let me explain why. I believe that the Reviewer refers to the well-known time inconsistency issue that may occur in certain situations (but not in my model). Time inconsistency emerges in cases that necessitate a reassessment of current policy decisions in the future, which can create situations in which policymakers in the future will adopt an “optimal” strategy that differs from the strategy that the present policymakers assumed to be the optimal future policy when they first determined the policy. For example, time inconsistency could occur if the pure time preference, ρ , varies over time, because in that case the planner today will have a certain plan, but the planner tomorrow will have different preferences and might change that plan. For another example, time inconsistency could occur if there is uncertainty with respect to future economic growth and the planner will have more information tomorrow than he/she has today.

All these scenarios are very interesting, but they are irrelevant for the present study. My model is deterministic, and the pure time preference, ρ , is simply a constant. Therefore, there exists a unique social optimum that maximizes the social welfare (Eq. 2). Namely, in my model, the change in the discount rates is due to real changes in the provision of the natural resource, and the planning is time consistent. The fact that there exist a unique solution to this kind of maximization problem is also well-known and is mentioned, for example, by C. Traeger (Ref. 35), who uses the same social welfare function and states that using a constant ρ “assures time consistency of the planning functional” (page 216, three lines below Eq. 2).

To clarify this issue in the Manuscript, I emphasize that “using the social welfare function given in Eq. 2 with a constant ρ , and considering deterministic dynamics of c and f , guarantee that the optimization problem is time consistent and has a unique solution (12,35))” (lines 342-344). (I also mentioned that the solution is unique throughout the model section.)

In your Discussion, you talk about the discount rate going up and down as if it were something that could be observed and might directly affect the behavior of agents. I find this confusing. Decentralized Agents will be using market rates – their cost of borrowing capital. I think of the optimal discount rate as something to be discussed by governments and researchers for instance

to put into DICE and other IAMs. Thus I think of rising and falling discount rates in a figurative sense but I worry you think of it as something that can be a) observed and b) has a direct effect in the behavior of agents.

The Reviewer comments that, in the Discussion section, I use phrases like “the discount rate increases”, which could be misinterpreted as if I were arguing that the discount rate could be observed or affect the behavior of agents.

First of all, let me emphasize that my intention was not to argue that the discount could be measured. Also, I do not mention at all that the behavior of any agent is affected by the discount. When I claim that the “discount rate declines”, for example, I simply mean that the policymaker needs to consider a lower discount rate. Namely, the social rate of discount declines in the sense that a social planner should consider a declining discount if he/she intends to do a good job (i.e., maximize social welfare). Note that many other authors use the same terminology and freely discuss the discount rate as increasing or decreasing.

At the same time, I understand from the Reviewer’s comment that the wording that I used has been confusing in several sentences and I revised these sentences accordingly. I believe that the confusion was mainly in the third paragraph of the Discussion and I rewrote three sentences there. In particular, instead of writing that the discount rate may increase in the short run but decrease in the long run, I explain that: “over-harvesting might continue for a couple of decades, which may keep the provision of natural resources high in the short run, but will ultimately result in an even lower provision of these resources. Therefore, continued over-harvesting may justify considering higher discount rates in the short run, but discounting the long run less” (where it is clear in the context that the policymakers are those that need to consider the lower or the higher discounts.)

In addition, in the previous version, I wrote that there is a controversy over the value of the rate of pure time preference, ρ . In the revised Manuscript, I rephrased this sentence and I clarify that the controversy is over the value of ρ that should be considered by policymakers (lines 317-318).

Finally as a separate point I still find it hard to see what drives the harvest decisions (various H) in your model. Is each manager making their own decision based – presumably on market rates of interest. What decides the entry and exit of different types of manager- and thus their numbers?

The Reviewer inquires as to how the managers decide how to harvest in the shared regions in my numeric model. First of all, note that I do not consider a behavioral model. Rather, I consider the well-known approach of a perfectly competitive market with utility-maximizing agents. Specifically, if ownership is well-defined everywhere and there are no externalities, the market solution coincides with the optimal solution. This is, of course, a standard modeling

approach, particularly in studies of economic growth and discounting (e.g., Refs. 12, 14, 31, 39-41, and many others).

In turn, for the model to be well-defined, one needs to define the externalities in each region. In the managed regions, there are no externalities. In the shared regions, I simply assume that the managers do not care about the future provision of the resource in the region (as the resource will be shared with many other managers). As a result, the managers in the shared regions harvest only non-sustainably, and the aggregate rate at which they harvest is such that the price of the natural resource becomes equal to the direct cost of the harvest. In turn, the harvest in the shared regions stops when these regions become fully degraded.

Note that I do not go into further details as to how agents enter into or exit from the shared regions, because these details are not necessary. The case in which the agents in the shared regions simply do not care about the future provision of the resource there defines the externality and thereby defines a unique solution for the maximization problem. This model could be realized in Stackelberg's setting in which there is no cost for entry and the number of agents approaches infinity. But again, these details are simply not necessary because I already assume that the managers have no incentive to conserve the natural resource in the shared regions.

To clarify this in the revised Manuscript, I begin with emphasizing that, for the numeric simulations, I consider a perfectly competitive market in which the market solution coincides with the optimal solution if there are no externalities (lines 187-191). Bringing this explanation to the beginning of the description makes it clear that I consider a standard macroeconomic model and not an agent-based behavioral model. In addition, I rewrote the sentences that describe the management in the shared regions (lines 197-202). Also, note that I give more details in the numeric methods section where I describe in more detail how I find the optimal and the market solutions (e.g., lines 339-349). I thank the Reviewer again for the very helpful comments on the various versions of the Manuscript.

REVIEWERS' COMMENTS:

Reviewer #2 (Remarks to the Author):

The manuscript has been improved substantially after several rounds of comments and editing.